# UNDERSTANDING THE IMPLICIT BIASES OF DESIGN CHOICES FOR TIME SERIES FOUNDATION MODELS

**Annan Yu,**[1,*]   **Danielle C. Maddix,**[2,†]   **Boran Han,**[2]   **Xiyuan Zhang,**[2]
**Abdul Fatir Ansari,**[2]   **Oleksandr Shchur,**[2]   **Christos Faloutsos,**[3]
**Andrew Gordon Wilson,**[4]   **Michael W. Mahoney,**[4]   **Yuyang Wang**[2]
[1] Center for Applied Mathematics, Cornell University
[2] Amazon Web Services    [3] Amazon Selling Partner Services
[4] Amazon Supply Chain Optimization Technologies

## ABSTRACT

Time series foundation models (TSFMs) are a potential class of powerful, general-purpose tools for forecasting and related temporal tasks, but their behavior is strongly shaped by subtle inductive biases in their design. Rather than developing a new model and claiming that it is better than existing TSFMs, e.g., by winning on existing benchmarks, our objective is to understand how the various "knobs" of the training process affect model quality. Using a mix of theory and controlled empirical evaluation, we identify and show how various design choices (e.g., patch size, embedding choice, training objective, etc.) lead to implicit biases in fundamental model properties (e.g., temporal behavior, geometric structure, how aggressively or not the model regresses to the mean, etc.), and how these biases can be intuitive or counterintuitive, depending on properties of the model and data. We illustrate in a case study on outlier handling how multiple biases interact in complex ways.

## 1 INTRODUCTION

Design decisions associated with modern machine learning models typically involve multiple trial-and-error iterations using one or more of many possible "knobs," including model architectures, loss functions, and parameters and hyperparameters such as learning rate, batch size, patch size, and embedding strategy. A common workflow is to adjust these knobs based on intuitions about the model and then use available data, typically in the form of curated, well-established benchmarks, to train and evaluate the model. For example, in the case of time series, intuitions include that nearby data points are similar and should be mapped close to each other, that low-frequency structure is typically more stable and informative than high-frequency fluctuations, and that regression algorithms should regress to the mean. This process, which is expensive, is iterated upon until one obtains a model that is *quantitatively* "better" on well-established benchmarks.

Unfortunately, model improvements often do not stand the test of time: when the data and compute are *much* larger, or when the data are *qualitatively* different than the well-established benchmarks on which the model was trained, the same knob fiddling can have a different, and often deleterious, effect. This leads to worse results and more model development effort in the longer term. Understanding how various design choices affect downstream model properties is of increasing importance as the community aims to design foundation-like models that are forward-compatible with qualitatively more and different data.

Here, we consider these issues in the context of recently-developed time series foundation models (TSFMs). Our goal is not to develop "yet another" model. Rather, we aim to understand the effects, i.e., the *implicit biases*, encoded by these various design decisions. We are particularly interested in how those effects vary with the properties of the data and the model, in particular for data that are similar to well-established benchmarks versus data that are qualitatively different. In time series forecasting, there are multiple well-known public benchmarks, e.g., M4 (Makridakis et al.,

---

*Work done during an internship at AWS.
†Correspondence to: Danielle C. Maddix <dmmaddix@amazon.com>.

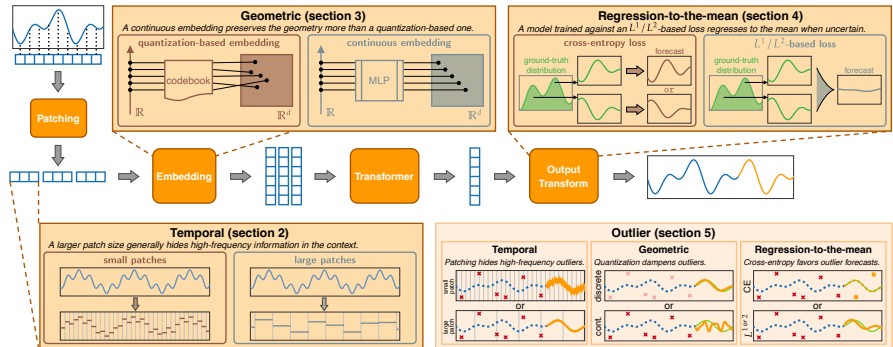

**Figure 1:** An illustration of examples of the three broad classes of biases that we identify and investigate. The *temporal bias* tends to be induced by patching, which transplants the temporal dependencies from the input sequence to a Transformer's hidden space; the *geometric bias* tends to be related to the input embedding, where a discrete versus a continuous embedding preserves the input space's geometry differently; the *regression-to-the-mean bias* tends to be caused by the training loss, where a continuous loss tends to average different outcomes more than a cross-entropy loss does. We also illustrate the interaction of these biases in a case study of outlier handling.

2020), ETT (Zhou et al., 2021), and the large collection from Godahewa et al. (2021) including Kaggle's Tourism (Athanasopoulos et al., 2011), Electricity (Trindade, 2015), Traffic (Lai et al., 2017), and Dominick (James M. Kilts Center, 2020). There has also a recent explosion of TSFMs, e.g., LLMTime (Gruver et al., 2023), Chronos (Ansari et al., 2024a), Moirai (Woo et al., 2024), TimesFM (Das et al., 2024), Chronos-Bolt (Ansari et al., 2024b), WaveToken (Masserano et al., 2025), and TOTEM (Talukder et al., 2024). If a foundation model is to be easily adapted to new forecasting tasks, with minimal task-specific data (Liang et al., 2024), this leads to an important open question: *how does one know the biases of various knobs in a TSFM on data that are qualitatively different from the benchmarks on which the TSFM was trained and evaluated?*

In this paper, we identify three classes of design choices — the "knobs" of patch size, whether the embedding is discrete or continuous, and the choice of loss; and, based on extensive analysis, we characterize three corresponding classes of implicit biases — which we call the "temporal bias," the "geometric bias," and the "regression-to-the-mean bias." We describe in detail how each design choice tends to affect the corresponding bias; and we show how, depending on the model, this effect could be intuitive or very counterintuitive. We also provide an example of how the design choices can interact in complex ways. Our main results, summarized in Figure 1, are as follows.

- **Patch Size → Temporal Bias.** Given a discrete sequence sampled from a time series, many TSFMs group these inputs into patches of length $k$, with $k = 1$ as a "trivial" corner case, and map each patch into a Transformer's hidden space. This design choice of the patch size parameter leads to a *temporal bias*: depending on its value, a TSFM may favor low-frequency information more or less aggressively. A larger patch size, e.g., 16 in Chronos-Bolt, favors low-frequency components in the time series, which can be challenging on chaotic (or even non-chaotic) systems that consist of high frequency and/or mixed-frequency components. Conversely, a smaller patch size, e.g., 1 in Chronos or 2 or 4 or 8 in Chronos-Bolt, can capture parts of these high frequency components. Importantly, many time series are periodic or quasi-periodic,[1] and capturing these two related but distinct phenomena can depend on temporal bias in subtle ways. (See section 2.)

- **Embedding Strategy → Geometric Bias.** In TSFMs, there are two common embedding strategies: discrete; and continuous. A discrete embedding quantizes the co-domain of a time series (e.g., the real line for a univariate time-series) into a finite number of regions, where each is treated as a "word token," as in an LLM. This is used in Chronos. A continuous embedding defines a trainable continuous function, usually an MLP, that maps the time series into a Transformer's hidden space. This is used in, e.g., Chronos-Bolt, Moirai, TimesFM and Time-MoE. This design choice leads to a *geometric bias*: continuous embeddings preserve most of the input's geometric structure, while quantization maps nearby values to different discrete tokens. As a result, quantization-based embeddings introduce several qualitative distinctions: they emphasize local

---

[1] Periodic phenomena repeat at *precise* intervals, e.g., New Year's Day, while quasi-periodic phenomena follow complex, non-repeating patterns of *roughly* regular intervals, e.g., Mother's Day.

information over global context; they treat fine- and large-scale patterns with less discrimination; and they remain invariant to signals that are shifted by an offset. (See section 3.)

- **Training Loss → Regression-to-the-mean Bias.** Discrete and continuous embeddings are two ways to handle the input domain; similarly, the output space can also be treated in a discrete or continuous way. A cross-entropy loss, e.g., in Chronos, computes forecasts as a discrete classification task, and is more aligned with LLMs. A continuous regression-based loss is more aligned with classical statistical methods: the $L^2$-based mean squared loss is a popular choice for models that perform point forecasting, e.g., TimesFM, whereas models that predict quantiles are usually trained on an $L^1$-based quantile loss or negative log-likelihood loss, e.g., Chronos-Bolt. This design choice leads to a *regression-to-the-mean bias*: TSFMs that use a regression-based loss tend to regress to either the mean or median forecast, while TSFMs that use a cross-entropy loss tend to regress much less aggressively and thus, for instance, have the flexibility to hop between basins for chaotic data and to better learn quasi-periodic information. (See section 4.)

In each case, we provide theory and/or quantification through controlled empirical evaluations to corroborate the existence of each bias and to demonstrate how each bias follows from the corresponding design choice. To demonstrate this, we mostly compare Chronos versus Chronos-Bolt because they are natural to compare: they use the same T5 Transformer backbone, they have comparable sizes, and they are pretrained on similar datasets (while most TSFMs are trained on different datasets). We also show in ablations that other TSFMs relying on the same choices of knobs behave (qualitatively) similarly to Chronos-Bolt. Lastly, as an illustration of how to use our insights, we demonstrate how these various biases can interact and mix with each other through an outlier handling analysis. (See section 5.) An important summary conclusion of our investigation is that *design choices, e.g., a large patch size, a continuous embedding, and a continuous training loss, which may induce intuitive implicit biases that work well on well-established time series benchmarks, may actually be deleterious to qualitatively different datasets such as chaotic systems, and for these datasets the opposite design choices may be beneficial.* (See Appendix A.)

**Related Work.** Recent time-series foundation models (TSFMs), e.g., TimesFM (Das et al., 2024), Chronos (Ansari et al., 2024a), Moirai (Woo et al., 2024), PatchTST (Nie et al., 2022), Time-MoE (Shi et al., 2025), Chronos-Bolt (Ansari et al., 2024b), iTransformer (Liu et al., 2023), UniTime (Liu et al., 2024b), TOTEM (Talukder et al., 2024), share the common goal of generalization across domains. Empirical studies compare them (Liang et al., 2025; Zhao et al., 2025) across various benchmarks. Several works also investigated specific

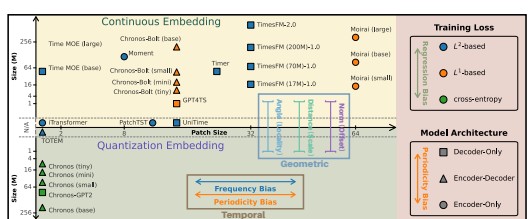

**Figure 2:** Design choices across TSFMs and their induced implicit biases.

design choices of TSFMs, e.g., tokenization strategies (Nie et al., 2022; Talukder et al., 2024), scaling laws for model size and dataset coverage (Woo et al., 2024; Shi et al., 2025), and architectural choices, e.g., state-space models vs. Transformers (Gu & Dao, 2023; Ma et al., 2024). In this work, we systematically analyze the inductive biases of Transformer-based TSFMs. (See Figure 2.)

While our experiments include many TSFMs, we extensively focus our comparisons on Chronos (Ansari et al., 2024a) and Chronos-Bolt (Ansari et al., 2024b) for two reasons. First, they share the T5 backbone of the same size and are trained on similarly mixed corpora of real-world and kernel-synthetic data. Second, together they span all major design choices discussed above: Chronos-Bolt treats the time-series co-domain as continuous, embeds size-16 patches with a two-layer MLP, and is trained with an $L^1$ quantile-regression loss; in contrast, Chronos "learns the language of time series" by quantizing values into 4096 bins (thus using no patching), assigning each bin a trainable embedding vector, and training via cross-entropy classification of real-valued output "tokens."

## 2 TEMPORAL BIAS: HOW DO TSFMS LEARN TIME?

A *temporal bias* occurs when a TSFM represents or learns certain temporal structures (e.g., specific frequencies and periodic patterns) more strongly than others. Time series data are fundamentally

different from discrete language data, in that they are often thought of as evolving over a continuous time domain. As a result, notions such as frequency, periodicity, and seasonality play a central role, and these notions often determine the dominant structures that a model must capture. Here, we distinguish between two related temporal notions (frequency and periodicity), and we describe how they are affected by different design choices. We outline key considerations in this section, and leave additional details to Appendix C.

**FREQUENCY.** The *frequency bias*, which occurs when a model learns low-frequency information better than high-frequency one, was observed and studied in general neural networks (Rahaman et al., 2019; Yang & Salman, 2019; Xu, 2020), and it was later extended to sequential models such as Transformers (Piao et al., 2024) and state-space models (SSMs) (Yu et al., 2025a). Most of these models have been shown, both theoretically and empirically, to be better at learning the low frequencies than high frequencies. One might hypothesize that we generally want to learn low-spectral modes better, because they usually contain less noise and/or describe more dominant "large-scale" structure. However, in many tasks, e.g., forecasting chaotic systems (Zhang & Gilpin, 2025a) or systems with quasi-periodic properties, high-spectral modes contain essential information that steers the trajectories, and their role cannot be downplayed. It is important to understand the cause of the frequency bias and to be able to tune it for the nature of the specific problem.

A TSFM operates on a discrete sample of a continuous time domain, in which a time series is defined. Given a continuous time series that can be sampled, we can hope to control this bias by changing the sampling interval $\Delta t$. More commonly, however, time series are recorded at fixed frequency intervals, and the data is a sampled sequence with a given frequency, and we have no control over $\Delta t$. When we talk about the "spectral modes," we imagine the Fourier modes in a fixed sequence $\mathbf{x} = (x_1, \ldots, x_L)$, indexed by natural numbers from $1$ to $L$. Then, we ask: what determines the spectral content that a TSFM will learn? Perhaps surprisingly, a frequency bias is induced as soon as $\mathbf{x}$ is embedded into the hidden space. The reason is that an embedding function $\phi : \mathbb{R}^k \to \mathbb{R}^d$ maps a patch of size $k$, i.e., $(x_i, \ldots, x_{i+k-1})$, into the hidden space. The following result, whose full statement and proof can be found in Theorem $1'$ in Appendix C.1.1, makes precise a sense in which patches of different frequencies are embedded differently.

**Theorem 1** (See Theorem $1'$ for the full statement)**.** Let $k$ be a patch size. Let $\mathbf{V} = \begin{bmatrix} \mathbf{v}_1 & \cdots & \mathbf{v}_n \end{bmatrix} \in \mathbb{R}^{k \times n}$ be a collection of $n$ patches whose frequencies range in an interval of size $\omega$ and $\mathbf{U} = \begin{bmatrix} \mathbf{u}_1 & \cdots & \mathbf{u}_n \end{bmatrix} \in \mathbb{R}^{k \times n}$ be a collection of $n$ patches whose frequencies are separated by a distance of at least $\omega$. For some $d, m \geq 1$, let $\phi(\mathbf{x}) = \mathbf{W}_2 \operatorname{ReLU}(\mathbf{W}_1 \mathbf{x}) + \mathbf{b}$ be a neural network embedding, with $\mathbf{W}_1 \in \mathbb{R}^{m \times k}$, $\mathbf{W}_2 \in \mathbb{R}^{d \times m}$, and $\mathbf{b} \in \mathbb{R}^d$ that satisfy certain regularity assumptions (see Theorem $1'$ in Appendix C.1.1). The following two statements hold:

1. (Similar frequencies are embedded in a shared subspace.) For any fixed $0 < \varepsilon \leq 1$, we have

$$\operatorname{rank}_\varepsilon(\phi(\mathbf{V})) := |\{j \mid \sigma_j(\phi(\mathbf{V})) > \varepsilon \sigma_1(\phi(\mathbf{V}))\}| = \mathcal{O}(1)\varepsilon^{-2}\omega,$$
$$\operatorname{stab-rank}(\phi(\mathbf{V})) := \|\phi(\mathbf{V})\|_F / \|\phi(\mathbf{V})\|_2 = \mathcal{O}(\omega).$$

2. (Different frequencies are embedded in nearly orthogonal subspaces.) There exists a universal constant $\varepsilon > 0$ so that with high probability, $\phi$ with random Gaussian weights satisfies

$$\operatorname{rank}_\varepsilon(\phi(\mathbf{U})) = \Omega(n), \qquad \operatorname{stab-rank}(\phi(\mathbf{U})) = \Omega(n).$$

Moreover, the constants in all $\mathcal{O}$ and $\Omega$-notations are universal.

The $\varepsilon$-rank and stable rank of a matrix $\phi(\mathbf{V})$ or $\phi(\mathbf{U})$ measure the "dimension" of its column space. In Theorem 1, $\mathbf{V}$ can be represented as a collection of patches of similar frequencies and $\mathbf{U}$ as a set of patches of heterogeneous frequencies. In essence, Theorem 1 says that patches with similar frequencies are embedded into a low-dimensional subspace of $\mathbb{R}^d$, with the "effective" dimension bounded proportionally to the bandwidth $\omega$ (see also Yu et al. (2025b)). At the same time, low-spectral and high-spectral modes are embedded in almost orthogonal subspaces. (See Figure 3c.) See Figure 10 in Appendix C.1 for a conceptual illustration of Theorem 1. One may wonder: what role does the patch size $k$ play in this story? If $k$ is small, then all patches share a similar bandlimit, because we must have $\omega \leq k$; as a result, all patches are embedded into a low-dimensional subspace of $\mathbb{R}^d$. With a random Gaussian assumption of the weights, Theorem 1 shows that several neural networks embed low- and high-spectral modes into orthogonal spaces. In trained TSFMs, even though weight matrices are not random Gaussian, we empirically find that the orthogonality is easily preserved. (See Appendix C.1.4.)

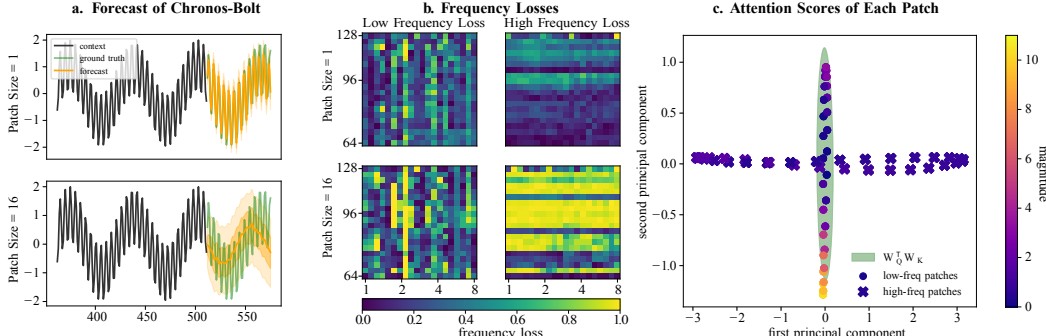

**Figure 3:** In (a), we show the forecasts of a signal consisting of a low-spectral and a high-spectral mode using two pretrained Chronos-Bolt models with patch size $k = 1$ and $k = 16$, respectively. In (b), we vary the frequencies of the two modes, and we evaluate the spectral loss in predicting each mode. We see that Chronos-Bolt with a patch size of 16 fails significantly at capturing the high-spectral information. In (c), we project the embeddings of low-spectral modes and high-spectral modes onto the two principal components, and we show how the attention matrices align with these two principal components on average. We observe that the attention scores are heavily dominated by the low-spectral modes.

Now that we know that when $k$ is large, low-spectral and high-spectral modes are embedded very differently, the next question is: which frequencies does the model learn best? In practice, we find that most TSFMs with a large patch size $k$ learn lower frequencies better than higher frequencies (see Figure 3a,b). This aligns with the conventional notion of "frequency bias" in other deep learning models (Rahaman et al., 2019; Yang & Salman, 2019; Yu et al., 2025a). A full understanding of this phenomenon can be obtained by analyzing the training dynamics of Transformers, but even without going into that level of detail, we have an intuitive understanding: in many time series, the dominant large-scale structure lies in the low-frequency components, while high-frequency content often appears more irregular or noisy. As a result, $\mathbf{W}_Q$, $\mathbf{W}_K$, and $\mathbf{W}_V$ tend to align more closely with the subspace containing low-spectral information (see Figure 3c and Appendix C.1.4).

**PERIODICITY.** The *periodicity bias* refers to a model's capability of recognizing and aligning (quasi-)periodic motifs in the input. Periodicity is a fundamental property of many time-series applications. By "periodicity," we refer to loosely recurring patterns rather than cycles that repeat at a *fixed* frequency. The perioidicity bias is controlled by several design choices. First, patching can directly reshape the periodic structure. If the period of the recurring motif does not align with the chosen patch size, the patching operation effectively samples the signal at off-grid locations, introducing aliasing effects that alter the perceived periodicity. These distortions accumulate across layers and change how the model "sees" repetition in the input (see Figure 4c for a large-scale experiment over 10,000 sequences showing decreased motif-matching quality

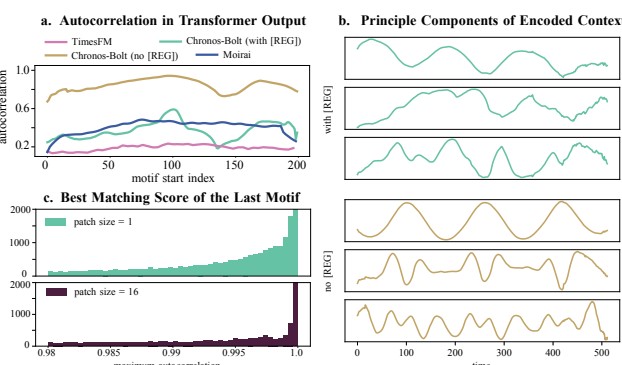

**Figure 4:** In (a), we show the autocorrelation in the output of a Transformer given a periodic input context, which measures the periodicity preservation of a model. In (b), we show leading principal components of the encoded context of a Chronos-Bolt model, with or without the [REG] token, given a periodic input. In (c), we show the best matching score, aggregated on 10,000 randomly sampled time series in Chronos' training dataset, with or without patching. The best matching score is computed by comparing the last motif of length 64 to all previous motifs.

as $k$ increases). Intuitively, patching replaces the original time axis with a coarser "patch axis." Then, periodic patterns that do not divide evenly into this new grid become misaligned, which makes it more challenging for the model to detect consistent repetitions. Second, the underlying

architecture influences how well periodicity is preserved. Encoder-only models tend to better maintain periodicity because they attend bidirectionally over the entire context, allowing the model to reinforce long-range regularities. In contrast, decoder-only or encoder–decoder architectures may dilute periodic structure when information flows primarily forward or is concentrated at a single aggregation position (see Figure 4a). In decoder-style attention, each position primarily attends to earlier tokens. Then, the periodic structure that requires comparing future and past motifs cannot be cleanly captured. Third, we find that the presence of a [REG] token can significantly disrupt periodic patterns. Although the [REG] token is intended as a designated location for aggregating global information, it can inadvertently overwrite earlier positions when unmasked. Our analysis shows that large values at the [REG] token propagate backward through self-attention and contaminate otherwise periodic channels in the hidden representation. This effect is visible in Figure 4b, where leading principal components lose their sinusoidal structure when the [REG] token is present. Conceptually, the [REG] token acts as a strong anchor point in attention space; when its value is large, it "pulls" nearby representations toward itself, breaking the smooth periodic patterns. In Appendix C.2, we provide a mechanistic study demonstrating that the channels whose periodicity collapses are precisely those receiving large contributions from the [REG] token, and that masking this token restores periodicity.

**Summary:** Frequency and periodicity biases are two connected views of how TSFMs represent time. *Frequency bias* is controlled by the patch size $k$, which determines whether embeddings emphasize low- or high-spectral modes. *Periodicity bias* is controlled by the alignment of patch size $k$ with the underlying recurrent motifs, architectural choices, such as encoder versus decoder design, and the use of an unmasked [REG] token.

## 3 GEOMETRIC BIAS: HOW DO TSFMS LEARN GEOMETRY?

In this section, we identify a *geometric bias*, which we define as the way an embedding choice distorts or preserves the input domain's geometric properties (induced by the standard inner product $\langle \cdot, \cdot \rangle$ on $\mathbb{R}^k$). This bias changes which local versus global patterns and scales are easiest to learn. There are two broad classes of embeddings: *quantization-based* (featured, e.g., in Chronos) and *continuous* (featured in most TSFMs, including Chronos-Bolt, TimesFM, and Moirai). A quantization-based embedding partitions $\mathbb{R}$ into regions $D_1, \ldots, D_V$ and defines $\phi_Q : \mathbb{R} \to \mathbb{R}^d$ as a step function that is constant on each $D_i$. A continuous embedding instead parameterizes $\phi_C : \mathbb{R}^k \to \mathbb{R}^d$ directly with a neural network. We use Chronos to represent the quantization-based embedding and Chronos-Bolt with a patch size of $k = 1$ to be representative of the continuous embedding. Note that we pretrain a new Chronos-Bolt model with $k = 1$ to eliminate the temporal bias from larger patch sizes, as discussed in Section 2, and to be more directly comparable with Chronos.

In a continuous embedding, the topology of $\mathbb{R}$ is inherently preserved by the continuity of $\phi_C$; but in a quantization-based embedding, the ordering of the real line must be (re)learned during training. Since each $\phi_Q(D_i)$ is initially a random vector, training rarely recovers the full geometry of $\mathbb{R}$ in $\mathbb{R}^d$. Here, we focus on three geometric notions (angle between a pair of vectors, distance between a pair of vectors, and norm of a vector), and we describe how they are affected by different design choices, and how they in turn affect the behavior of a TSFM. At first glance, it may seem that it is generally good to preserve the geometry (i.e., the angle, distance, and norm) of the input space, as a continuous embedding does. We show that while not preserving the geometry does lead to certain problems, it is also surprisingly beneficial in some cases. We outline key considerations in this section (see Appendix D for details).

**ANGLES.** Our first measurement of the geometric bias is the *angular bias*, which makes a TSFM to favor local information over global information. This angular bias is caused by difference in the angles between the embedded vectors:

$$\theta_Q(x,y) = \arccos\left(\frac{|\phi_Q(x) \cdot \phi_Q(y)|}{\|\phi_Q(x)\|_2 \|\phi_Q(y)\|_2}\right), \quad \theta_C(x,y) = \arccos\left(\frac{|\phi_C(x) \cdot \phi_C(y)|}{\|\phi_C(x)\|_2 \|\phi_C(y)\|_2}\right),$$

for the two types of embedding. In Figure 5a, $\theta_Q(x,y)$ is significantly larger than $\theta_C(x,y)$ as a function of $|x - y|$, which makes an input appear more different from its close neighbors. This difference increases with the number of bins used in quantization (see Appendix D.4).

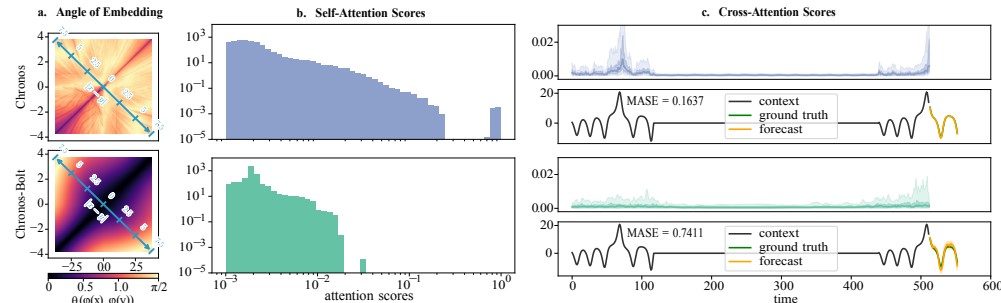

**Figure 5:** Panel (a) shows the angles between the embedded vectors $\phi(x)$ and $\phi(y)$ for Chronos' embedding $\phi_Q$ and Chronos-Bolt's embedding $\phi_C$, respectively, where $x$ and $y$ denote two real numbers to be embedded. Panel (b) shows a histogram of all self-attention scores when inferring the two models on a corpus of contexts. Panel (c) shows the forecasts from both models on a context formed by repeating a motif of a chaotic system. Chronos achieves a significantly lower MASE than Chronos-Bolt. The cross-attention scores are aggregated across all layers when generating the first prediction token. See Appendix D for details.

The angular bias affects both self-attention and cross-attention in a Transformer. In Figure 5b, the self-attention scores of Chronos are bimodal, i.e., tokens place high attention weights on their immediate neighbors, and low attention weights on distant tokens. In contrast, the attention scores of Chronos-Bolt are more evenly distributed, indicating a broader mixing of contextual information. A similar pattern appears in the right panels for cross-attention. Chronos focuses almost entirely on the most relevant part of the context, whereas Chronos-Bolt attends more broadly across the context. Among other things, this makes Chronos better at "parroting" — one reason why it performs better than other TSFMs on chaotic systems (Zhang & Gilpin, 2025b). This comes at the cost of less mixing across the context, which can hurt performance on tasks that require synthesizing information from distant parts of the time series or jointly interpreting multiple interacting patterns. Notably, for quantization embeddings, the size of the vocabulary also controls the bias (see Appendix D.4).

**DISTANCES.** Our second measurement of the geometric bias is the *distance bias*, which makes a TSFM magnify small scales, and makes nearby numbers appear more distinct in the hidden space. This distance bias is caused by the difference in the relative distance, defined as:

$$d_Q(x,y) = \frac{\|\phi_Q(x) - \phi_Q(y)\|_2}{\|\phi_Q(x)\|_2 + \|\phi_Q(y)\|_2}, \quad d_C(x,y) = \frac{\|\phi_C(x) - \phi_C(y)\|_2}{\|\phi_C(x)\|_2 + \|\phi_C(y)\|_2}.$$

Just as the learned quantization embedding $\phi_Q$ bends the input space $\mathbb{R}$ in the hidden space $\mathbb{R}^d$, as we saw via angle measurement, it also stretches or squeezes $\mathbb{R}$. While a continuous embedding is naturally expected to map nearby numbers to nearby vectors in the hidden space, a learned $\phi_Q$ is generally less scale-preserving (see Figure 6a).

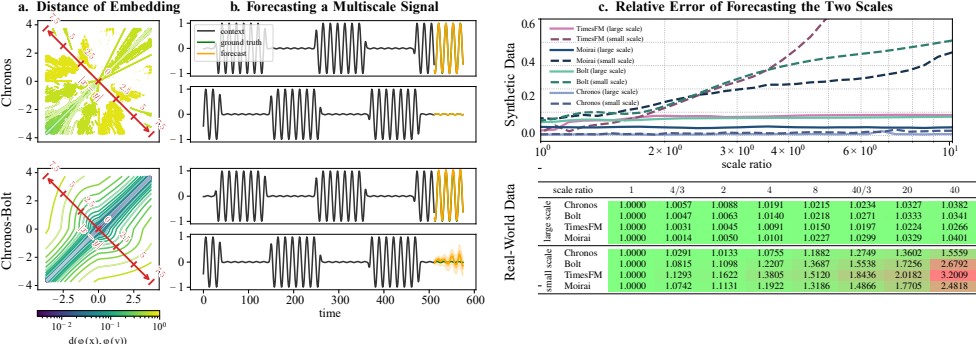

**Figure 6:** Panel (a) shows the distance between the embedded vectors for Chronos' embedding $\phi_Q$ and Chronos-Bolt's embedding $\phi_C$. Panel (b) shows Chronos' and Chronos-Bolt's forecasts of a multi-scale time series. Panel (c) quantifies the forecasting quality by computing the relative MSE, as we increase the ratio between the two scales. See Appendix D for details.

The distance bias results in a mismatch in scale preservation. As a result, when a multi-scale structure is present, Chronos is more sensitive to and better at learning fine-scale patterns than a Chronos-

Bolt model. Figure 6b well-illustrates this phenomenon: while both Chronos and Chronos-Bolt are good at learning the large-scale pattern, only Chronos nicely learns the small-scale patterns. A more systematic quantification is given in Figure 6c, where we show that, as the ratio between the two scales increases, Chronos-Bolt, TimesFM, and Moirai, all of which rely on a continuous embeddings, are less capable of learning the fine-scale pattern. Whether this is an advantage or a drawback is again task-dependent (see Appendix A): it can be valuable when fine-scale detail is important, e.g., when learning high-frequency information or when learning to perform context parroting; but may be unnecessary, or even distracting, in simpler situations when large-scale trends dominate.

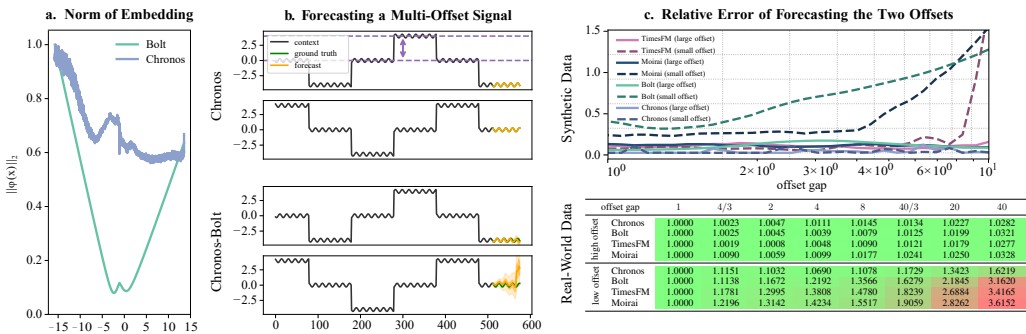

| offset gap | | 1 | 4/3 | 2 | 4 | 8 | 40/3 | 20 | 40 |
|---|---|---|---|---|---|---|---|---|---|
| high offset | Chronos | 1.0000 | 1.0023 | 1.0047 | 1.0111 | 1.0145 | 1.0134 | 1.0227 | 1.0282 |
| | Bolt | 1.0000 | 1.0025 | 1.0045 | 1.0039 | 1.0079 | 1.0125 | 1.0199 | 1.0321 |
| | TimesFM | 1.0000 | 1.0019 | 1.0008 | 1.0048 | 1.0090 | 1.0121 | 1.0179 | 1.0277 |
| | Moirai | 1.0000 | 1.0090 | 1.0059 | 1.0099 | 1.0177 | 1.0241 | 1.0250 | 1.0328 |
| low offset | Chronos | 1.0000 | 1.1151 | 1.1032 | 1.0690 | 1.1078 | 1.1729 | 1.3423 | 1.6219 |
| | Bolt | 1.0000 | 1.1138 | 1.1672 | 1.2192 | 1.3566 | 1.6279 | 2.1845 | 3.1620 |
| | TimesFM | 1.0000 | 1.1781 | 1.2995 | 1.3808 | 1.4780 | 1.8239 | 2.6884 | 3.4165 |
| | Moirai | 1.0000 | 1.2196 | 1.3142 | 1.4234 | 1.5517 | 1.9059 | 2.8262 | 3.6152 |

**Figure 7:** Panel (a) shows the norm of the embedded vectors $\phi_Q(x)$ and $\phi_C(x)$. Panel (b) shows the forecasts of a multi-offset time series for Chronos and Chronos-Bolt, respectively. Panel (c) quantifies the forecasting quality by computing the relative (to each stage's amplitude) MSE of the forecasts as we increase the offset. See Appendix D for more details.

**NORMS.** Our last measurement of the geometric bias is the *norm bias*, which makes a TSFM better at learning motifs that are larger in magnitude. This norm bias concerns the mapping of a single element, rather than pairs of elements, as in the other two cases. For a continuous embedding $\phi_C$, many TSFMs use a ReLU-activated MLP. Because the ReLU activation is positive-homogeneous, $\phi_C$ maps small input values to small vectors in $\mathbb{R}^d$ and large values to large vectors. A quantization-based embedding $\phi_Q$ behaves differently. Since the mapping of each bin, $\phi_Q(D_i)$, is initialized from the same distribution, it has no inherent bias toward preserving norms (see Figure 7a).

The norm bias is caused by the tendency of $\phi_C$ to map large numbers to vectors with large norms. In Figure 7b, the scale of each period is the same; the only difference is their offsets. When a motif is large in magnitude, its corresponding embedded vectors are also large in magnitude. In Transformer's hidden space, they appear more dominating than the smaller vectors associated with a motif with a small magnitude (see Appendix D.2.2). Conversely, a period near zero is mapped to small vectors that are more easily overwhelmed by the rest of the context. Figure 7c illustrates this effect: as the offset increases (equivalently, as the period's oscillation amplitude decreases while the offset stays fixed), Chronos-Bolt struggles to forecast the near-zero period. For regular signals with multiple offsets, this can introduce an undesirable imbalance; but for sparse signals, it can be advantageous, by allowing the model to focus on nonzero entries and ignoring the zeros. (See Appendix A for beneficial and deleterious examples.)

> **Summary of the Geometric Bias:** Geometric bias concerns how the geometry of the input domain is preserved by embeddings, and it is controlled by the choice between continuous and quantization-based strategies. From the angle perspective, quantization induces an *angular bias*, making the model favor local information. From the distance perspective, continuous embeddings advocate a *distance bias*, making fine-scale patterns less learnable. From the norm perspective, continuous embeddings induce a *norm bias*, weighting large-magnitude elements more heavily, while quantization embeddings treat inputs more evenly.

# 4 REGRESSION-TO-THE-MEAN BIAS: DO TSFMS REGRESS TO THE MEAN?

In this section, we identify a *regression-to-the-mean* bias, which is the tendency of a TSFM to collapse future variability toward a central forecast (mean or median) versus maintaining sharper, more extreme or multimodal outcomes under uncertainty. When the future is stochastic, should a

model settle on a safe, compromised prediction, e.g., regress to the mean forecast, or should a model aggressively commit to one of the possible outcomes? Here, we analyze the effects of various design choices on how aggressively (or not) a TSFM regresses to the mean (see Appendix E for details).

How a model reacts to uncertainty heavily depends on how it is trained. Most TSFMs are trained with either an $L^2$-based loss (e.g., mean-squared loss) or an $L^1$-based loss (e.g., quantile loss). A notable exception is Chronos, which is trained with a cross-entropy loss, like LLMs. If our target comes from an unknown probability distribution with large variance, then:

- $L^2$**-based loss:** favors the *mean* of the distribution, by averaging all possible outcomes.
- $L^1$**-based loss:** favors the *median*, which gives a central split of the probability mass in half.
- **Cross-entropy loss:** models the full probability distribution (Stewart et al., 2023) and can settle on a/the "*mode*." This can avoid a "compromise" between possible outcomes, and it allows the model to represent distributions with large variances and stronger heterogeneity more faithfully.

A $L^2$- or $L^1$-based loss explicitly tailors the forecast to minimize pointwise error metrics such as MAE or MSE. From traditional wisdom, a low MAE or MSE is usually good, but time series forecasting is not all about pointwise fitting. For instance, in chaotic systems, the fractal dimension is a measurement of the long-term geometry of the trajectories, and regressing to the median or the mean can severely damage it (Zhang & Gilpin, 2025a). (See Appendix A for cases where the bias is "beneficial" or "deleterious.") Unlike mean- or median-based forecasts, "regressing to the mode" lets the prediction reflect a sharp, high-probability outcomes, instead of a smoothed central tendency.

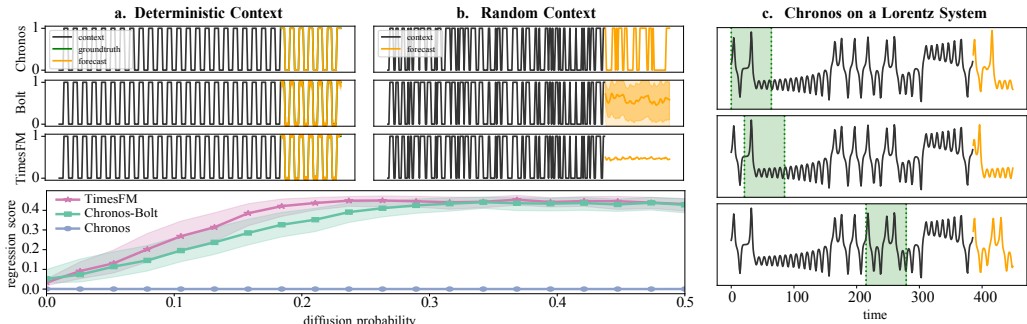

**Figure 8:** Panels (a) and (b) show the forecasts from three TSFMs of a purely deterministic and a purely random context, respectively. We also quantify the "regression score" by computing $\min(|\hat{y}|, |1 - \hat{y}|)$, where $\hat{y}$ denotes the forecast from a model. As the context involves more uncertainty, TimesFM and Chronos-Bolt regress to the mean and the median, respectively, more than Chronos. Panel (c) shows three examples where Chronos "parrots" three distinct outcome branches from the context of a Lorentz chaotic system. See Appendix E for more details.

We illustrate the regression-to-the-mean bias using the examples in Figure 8. In (a), we show a periodic walk between $0$ and $1$. Since the process is fully deterministic, Chronos (cross-entropy loss), Chronos-Bolt ($L^1$ loss), and TimesFM ($L^2$ loss) produce similar forecasts. Panel (b) uses a context generated by a random walk. With probability $1/2$ on each branch, the uncertainty in the context drives the forecast of TimesFM toward the mean, Chronos-Bolt toward the median (any value in $[0, 1]$), and Chronos toward the mode (either $0$ or $1$). We quantify this effect by "bridging" the purely deterministic and purely random cases. For each $0 \leq p \leq 1/2$, we take a periodic walk and diffuse it with probability $p$. As $p$ increases, the context becomes more uncertain and models trained with a continuous loss regress more strongly than the one trained with cross-entropy.

> **Summary of the Regression-to-the-Mean Bias:** An $L^2$- or $L^1$-based loss tends to collapse future variability into a central forecast, reflecting mean or median behavior and usually resulting in a good pointwise fitting. In contrast, cross-entropy-based training represents the full distribution and preserves sharper outcomes, instead of aggressively regressing to a central tendency.

## 5 MIXTURE OF BIASES: OUTLIER HANDLING

In the previous three sections, we have identified three key knobs that affect three fundamental implicit biases in the development of TSFMs. Of course, in practical settings, these biases can interact

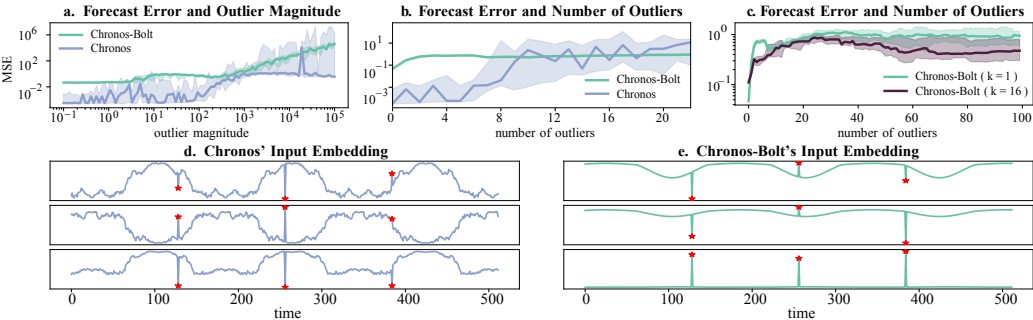

**Figure 9:** Top row shows the MSE of the baseline models of forecasting a sinusoidal wave with randomly injected outliers. Panel (a) shows the effect of increasing the outlier's magnitude, and panels (b) and (c) show the MSE as we increase the number of outliers. We repeat the experiments are 30 times. Panels (d) and (e) show the first principal component of the embedded input in Chronos and Chronos-Bolt as we increase the magnitude of the outlier.

in complex ways, and the effect of fiddling with a given knob can be nontrivial to characterize. In this section, we present outlier handling as an illustration of how different biases can combine in natural ways. We use a sinusoidal wave as the context and inject outliers into it, in two scenarios: (1) increasing the magnitude of a fixed number of outliers; (2) increasing the number of outliers sampled i.i.d. from a fixed Gaussian distribution. Our results, as shown in Figure 9, are the following:

1. **Temporal bias.** Comparing the two Chronos-Bolt models with patch sizes $k = 1$ and $k = 16$ shows that, without outliers, the $k = 1$ model performs better. As outliers are injected, they inject high-frequency perturbations, making the larger patch size more effective at denoising (see Figure 9c). This crossover behavior is consistent with our temporal-bias analysis: larger patches suppress high-frequency variation by averaging over a wider window. Injected spikes are smoothed out while the underlying sinusoid is largely preserved in the embedded sequence.

2. **Geometric (Distance) bias.** When outlier contamination is limited, i.e., their magnitude and number are relatively small, Chronos outperforms Chronos-Bolt by reducing the apparent scale of outliers through its quantization-based embedding (see Figure 9d). In Chronos, extreme values are mapped into nearby bins whose embeddings are not much farther away to those of the clean signals. Isolated spikes no longer dominate local distances in the hidden space, which makes the learning of the underlying sinusoidal wave more tractable.
   **Geometric (Norm) bias.** Chronos-Bolt's continuous embedding causes large-magnitude outliers to dominate the embedded context. As the outlier magnitude increases, this dominance leads to progressively worse forecasts (see Figure 9a,e). The large embedded vectors in Chronos-Bolt pull attention toward the timestamps corresponding to outliers, which reweighs the context in favor of the outliers.

3. **Regression-to-the-Mean bias.** Although Chronos handles outliers well on average, its performance variance is larger than that of Chronos-Bolt. This is because the regression-to-the-mean bias can cause Chronos to produce an "outlier" when it is uncertain whether the observed outliers are part of the underlying context (see Figure 9b). This leads to occasional large forecast deviations even when the average error is small, which highlights a fundamental trade-off between robustness in expectation and higher conditional variance for TSFMs that seek dominant modes.

## 6 CONCLUSION

We identify and characterize three broad classes of inductive biases in TSFMs, which are induced by various common design "knobs", These biases influence the strengths and weaknesses of TSFMs across tasks in subtle and sometimes counterintuitive ways. Our findings indicate that there is no single choice of patch size, embedding strategy, or loss function that is uniformly optimal across all forecasting regimes. Each induces biases that help on certain types of data, and hinder performance on others. This highlights a central challenge for TSFMs: unlike language, time-series data exhibit widely varying characteristics, which makes universal modeling substantially more challenging. Understanding these biases helps explain why different models excel in different settings, and clarifies trade-offs among architectural and training choices. Lastly, our work suggests that dynamically adapting these biases, e.g., through in-context learning, retrieval mechanisms, or reinforcement-learning-based adaptation, may be a more promising path toward broadly applicable TSFMs.

ACKNOWLEDGMENTS

The authors would like to thank William Gilpin and Yuanzhao Zhang for many fruitful discussions on this work.

REPRODUCIBILITY STATEMENT.

Code used to produce the experiments in this paper, along with scripts for data processing and figure generation, is available at https://github.com/amazon-science/TSFM-Biases.

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

# A    TWO SIDES OF EACH BIAS

In the main article, we showed many examples in which a bias may be "good" or "bad" in certain situations; and we also emphasized that each bias controls a trade-off, and that there are important applications on both ends of the spectrum of a bias. In this section, we show concrete synthetic examples illustrating when a bias may be "good" or "bad." (These examples illustrate learnings from developing Chronos-Bolt to "improve" Chronos on well-established benchmarks, and the observation that Chronos-Bolt performs worse than Chronos on the qualitatively-different chaotic nonlinear dynamics data.) The examples in this appendix are *not* cherry-picked, but they are highly context-specific; for a more generic and systematic analysis, see the main article. See Table 1 for an overview of our discussion.

The way to read Table 1 is as follows: for each bias, we compare two models on the two different ends of the spectrum. Then, we come up with two examples: in the first example, we show that having this bias is good for this particular example; in the second example, we show the opposite. A green background always indicates that a forecast is qualitatively better, while a red background shows otherwise.

## A.1    TEMPORAL BIAS

As described in Section 2, there are two biases related to the temporal bias: the frequency bias; and the periodicity bias.

### A.1.1    FREQUENCY BIAS

The frequency bias has been known to the deep learning community for almost a decade; and it is often thought of as an implicit regularization. Driven by this, we first come up with an example where the context is

$$x(t) = \sin(t) + \text{noise}, \tag{1}$$

where the noise is obtained by sampling every entry from the discrete sequence i.i.d. according to the standard Gaussian distribution $\mathcal{N}(0, 1)$. In this case, $\sin(t)$ is the useful low-frequency information that one wants to learn, while the Gaussian noise is high-frequency content that we do not wish to learn. We see that with a larger patch size, and therefore a stronger frequency bias, Chronos-Bolt is more capable of "denoising."

On the other hand, the example in Figure 3 has already shown a case where biasing towards low frequencies is not desirable:

$$x(t) = \sin(\alpha t) + \sin(\beta t), \qquad \alpha \ll \beta. \tag{2}$$

In this case, both the low-frequency information $\sin(\alpha t)$ and the high-frequency information $\sin(\beta t)$ contain useful information, but Chronos-Bolt with a patch size of 16 only enables us to learn the low-frequency content.

### A.1.2    PERIODICITY BIAS

To understand the pros and cons of preserving the periodicity bias, we can first consider a signal formed from a chaotic system's data:

$$x(t) = \mathbb{1}_{[0,\Delta]}(t)F(t) + \mathbb{1}_{[T-\Delta,T]}(t)F(t - T + \Delta), \tag{3}$$

where $F(\cdot) : [0, \Delta] \to \mathbb{R}$ is a motif taken from a Lorentz oscillator. Since $F$ is repeated twice in the context, and that the task can be solved by simply "matching" the repeating motifs, we see that a model that preserves periodicity better, i.e., Chronos-Bolt, performs better than one that destroys periodicity more, i.e., TimesFM.

However, if one focuses too much on the periodic patterns, then one may overlook the important non-periodicities. Consider the following context:

$$x(t) = \mathbb{1}_{R_1} t \sin(t) + \mathbb{1}_{R_2} \sin(t), \tag{4}$$

where $R_1$ is the disjoint union of all even ones from an evenly divided set of intervals, and $R_2 = [0, \infty) \setminus R_1$ is the complement of it. We see in Table 1 that Chronos-Bolt focuses heavily on the periodic patterns on $R_2$, resulting in a bad forecast, whereas TimesFM gives a more accurate partition of $R_1$ and $R_2$ in its forecast.

**Table 1:** Illustration of when inductive biases "help" and when they "hurt" performance.

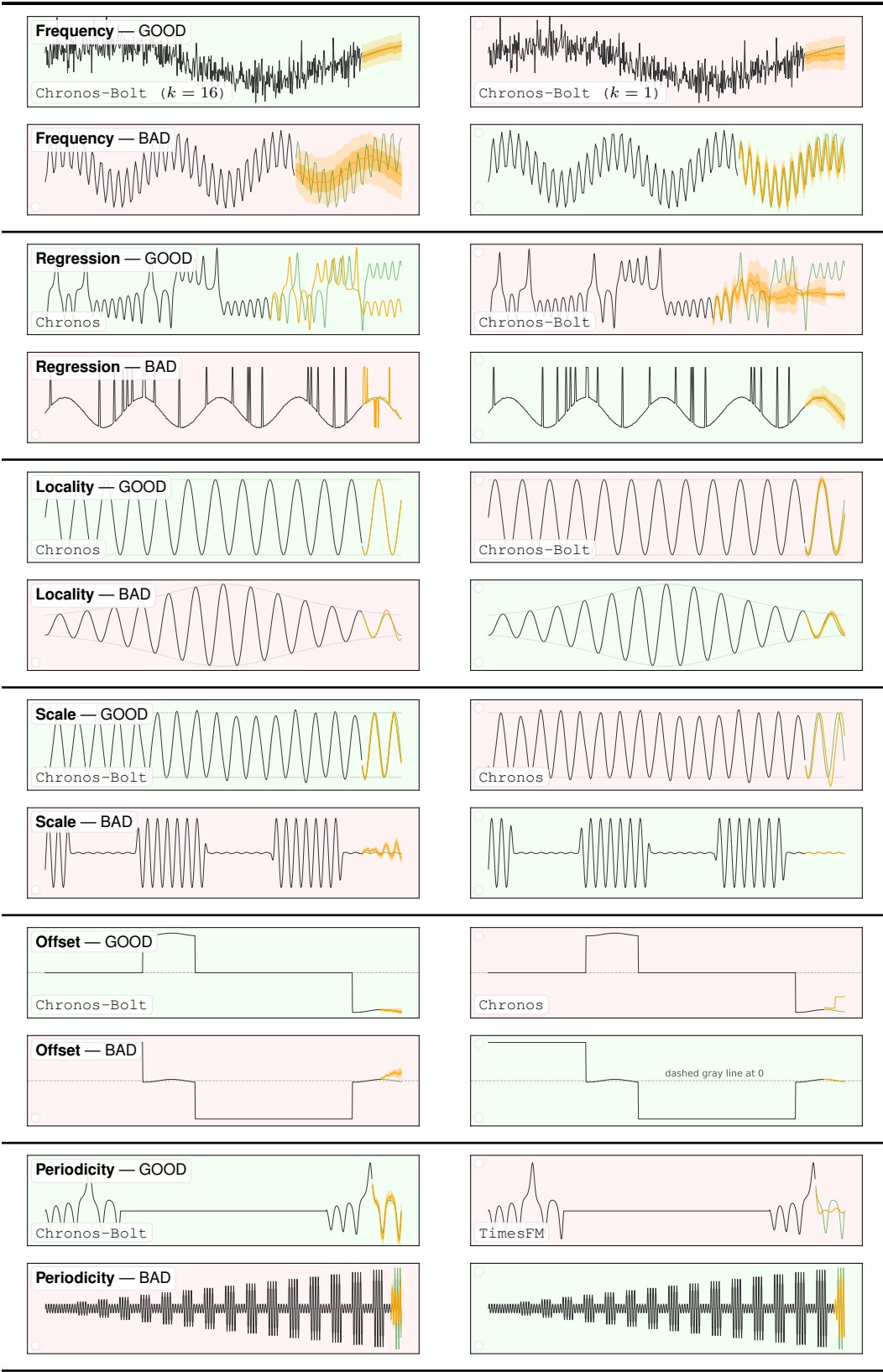

## A.2 GEOMETRIC BIAS

As described in section 3, there are three parts to the geometric bias: an angular bias, which concerns the angles between the embeddings; a distance bias, which is due to the distance between the embeddings; and a norm bias, which originates from the homogeneity of the ReLU activation. We show interesting examples for each of them.

### A.2.1 ANGULAR BIAS

A model that has an angular bias, e.g., Chronos, tends to tailor the "parroting" strategy. When there is a matching motif in the earlier context, then the model will parrot what is after it. Therefore, if we set the context to be a perfectly periodic signal

$$x(t) = \sin(t), \tag{5}$$

then we see that Chronos' forecast is pinpoint, while Chronos-Bolt suffers from certain small errors.

However, parroting can also be dangerous, not only because of the fact that sometimes there is not too much to parrot, but also since it can prevent the model from identifying more subtle patterns. Consider the following context:

$$x(t) = \sin(t)S(t), \tag{6}$$

where $S(t)$ is a bidirectional scaling factor that vanishes as $t \to \pm\infty$. In this case, Chronos is able to identify the recent context to a previous motif. Unfortunately, this "drags" Chronos into a caveat: simply parroting the previous context results in a forecast with an increasing trend, whereas the actual trend is clearly decreasing over time. In this case, Chronos-Bolt gets a better forecast than Chronos.

### A.2.2 DISTANCE BIAS

Similar to the frequency bias, fine-scale oscillation may sometimes be simply due to random noise. In that sense, the distance bias could be helpful for denoising. Consider the following context:

$$x(t) = \sin(t) + \text{noise}, \tag{7}$$

where the noise is obtained by sampling every entry from the discrete sequence i.i.d. according to the standard Gaussian distribution $\mathcal{N}(0, \alpha)$ for some small $\alpha$, and then passing it through a low-pass filter. The reason why we choose $\alpha$ small is that we want to make the noise small-scale, and the reason why we pass it through the low-pass filter is to decouple the example from the frequency bias. Here, we see that Chronos learns these small-scale noises, while Chronos-Bolt, having a distance bias, only learns the large-scale trend.

Of course, when small-scale properties appear to be important, Chronos performs better than Chronos-Bolt. One example is shown in Figure 6, where

$$x(t) = \sin(t)S(t), \tag{8}$$

where $S(t)$ is a scaling factor that periodically alternates between large and small values.

### A.2.3 NORM BIAS

Regarding the norm bias, it may not be clear why one should treat signals at different offsets differently. Indeed, if we consider the signal labeled "Offset—BAD" in Table 1, we see that Chronos is much better at continuing the signal near zero than Chronos-Bolt. However, sometimes the small signals in the input domain are really "physically" less important than the large signals. One important example is with a sparse signal. For example, consider the signal labeled "Offset—GOOD" in Table 1. This signal is sparse, consisting of mostly zeros and a few non-zero motifs. Essentially, the most important information is contained in these non-zero motifs. Chronos-Bolt's MLP embedding assigns the zero values small embeddings, and it focuses more on the non-zero parts, resulting in a better prediction than Chronos.

### A.3 REGRESSION-TO-THE-MEAN BIAS

As described in Section 4, it may be intuitive that regression algorithms should regress to the mean, but an example where this bias is not desired is provided by the chaotic systems data from Zhang & Gilpin (2025a). (Probabilistic forecasting has become an important paradigm in real-world applications, from weather and climate prediction (Gneiting & Katzfuss, 2014), to energy demand forecasting (Hong et al., 2016), to financial risk management (Gneiting et al., 2007); and evidence suggests that similar results may hold more generally for TSFMs.) In particular, we use the Lorentz oscillator to generate our context. We see that since the system is chaotic, neither forecast stays close to the ground-truth, but Chronos' prediction resembles the context much more than Chronos-Bolt's. That is, while from an MSE or MAE perspective, Chronos is not doing so much better than Chronos-Bolt, it performs better on other aspects, such as the frequency loss or the fractal dimension (Zhang & Gilpin, 2025a). More colloquially, without showing the ground-truth, Chronos gives a more credible continuation of the context and Chronos-Bolt.

Clearly, of course, in many other cases, regression-to-the-mean is desirable. A significant example of this arises when outliers are introduced. For example, consider the following context:

$$x(t) = \sin(t) + \text{Bernoulli}(p)\delta_t, \tag{9}$$

where $\delta_t$ is the Dirac-delta and Bernoulli$(p)$ is i.i.d. random variable with a small probability $0 < p < 1$. This signal can model a demand curve that follows some seasonal trend but can be excited by several random events. In this case, we see that Chronos sometimes predicts these outliers, which we do not want, while Chronos-Bolt generally avoids them.

## B RELATED WORK

In this section, we describe several lines of related work.

### B.1 TIME SERIES MODELING AND FOUNDATION MODELS

Classical time series forecasting (Hyndman & Athanasopoulos, 2018) is grounded in statistical methods such as ARIMA (Makridakis & Hibon, 1997) and exponential smoothing (Gardner Jr, 1985). More recently, deep learning approaches, e.g., N-BEATS (Oreshkin et al., 2020), have become prominent. Sequence models such as DeepAR (Salinas et al., 2020) popularized probabilistic forecasting at scale, while the Temporal Fusion Transformer (Lim et al., 2021) combined attention mechanisms with interpretability for multi-horizon prediction. Building on these advances, recent work has focused on pretraining general-purpose time series models that transfer across datasets and tasks. Notable examples include TimesFM (Das et al., 2024), Chronos (Ansari et al., 2024a), Moirai (Woo et al., 2024), and Time-MoE (Shi et al., 2025). These models vary in tokenization strategies, training corpora, and zero-shot protocols, but they all share the common goal of developing a single "foundation model" that generalizes across domains and horizons. Such TSFMs, which we compare along several axes in Figure 2, include Chronos (Ansari et al., 2024a), TimesFM (Das et al., 2024), Moirai (Woo et al., 2024), PatchTST (Nie et al., 2022), Time MOE (Shi et al., 2025), Chronos-Bolt (Ansari et al., 2024b), iTransformer (Liu et al., 2023), Moment (Goswami et al., 2024), Timer (Liu et al., 2024c), GPT4TS (Zhou et al., 2023), UniTime (Liu et al., 2024b), and TOTEM (Talukder et al., 2024).

### B.2 DESIGN CHOICES OF TSFMS

Design choices for TSFMs have caught research attention, e.g., with empirical work in Liang et al. (2025); Zhao et al. (2025). Beyond these benchmark studies, several papers have investigated specific design dimensions of TSFMs. For instance, recent work has examined the impact of tokenization strategies (Nie et al., 2022; Talukder et al., 2024), scaling laws for model size and dataset coverage (Woo et al., 2024; Shi et al., 2025), and architectural choices such as state-space models versus Transformers (Gu & Dao, 2023; Ma et al., 2024). Other efforts have focused on evaluation protocols and generalization across domains, highlighting the importance of systematic comparisons to guide the development of more reliable TSFMs.

### B.3 DEEP LEARNING MODELS FOR CHAOTIC SYSTEMS FORECASTING

We have used time series from chaotic dynamical systems data as part of our investigation. Recent work (Rackauckas, 2025) has highlighted the promise and limitations of using deep learning models to learn chaotic systems. A major line of work from Gilpin's group has made fundamental contributions, including providing a benchmark of many state-of-the-art methods—ranging from reservoir computing to TSFMs—showing that large, domain-agnostic models can achieve accurate predictions up to two dozen Lyapunov times into the future (Gilpin, 2021). Further empirical studies demonstrate that TSFMs like Chronos can perform zero-shot forecasting of chaotic dynamics, preserving the geometric and statistical properties of chaotic attractors even when point forecasts degrade (Zhang & Gilpin, 2025a;b; Lai et al., 2025). Beyond TSFMs, physics-informed networks leverage known physical constraints to extend predictability limits (Doan et al., 2020; Steger et al., 2022), while knowledge-based hybrid models combining mechanistic and data-driven components reliably forecast complex chaotic behaviors far longer than either approach alone (Pathak et al., 2018).

### B.4 FREQUENCY BIAS

The frequency bias is related to the spectral bias of neural networks, which was observed and studied in Rahaman et al. (2019); Yang & Salman (2019); Xu (2020). The term "spectral bias" comes from the spectral decomposition of neural tangent kernels (NTKs) (Jacot et al., 2018), which approximate the training dynamics of overparameterized networks (Arora et al., 2019; Su & Yang, 2019; Cao et al., 2019). By analyzing the eigenfunctions of NTKs, Basri et al. (2019); Bietti & Mairal (2019) formally proved the presence of spectral bias in two-layer overparameterized networks with uniform

input distributions; and these results were later extended to nonuniform inputs (Basri et al., 2020; Yu et al., 2023b). Several approaches have since been proposed to tune the strength of this bias, including Sobolev-norm-based training strategies (Vlassis & Sun, 2021; Yu et al., 2023b; Tsay, 2021; Son et al., 2021; Czarnecki et al., 2017; Zhu et al., 2021; Son, 2023; Liu et al., 2024a) and frequency-aware initialization methods (Yu et al., 2023a; 2025a).

## C  DETAILS OF THE TEMPORAL BIAS (FROM SECTION 2)

In this section, we provide more details on the temporal bias, which was discussed in Section 2. We discuss both the frequency bias (Appendix C.1) and the periodicity bias (Appendix C.2).

### C.1  DETAILS OF THE FREQUENCY BIAS

In this subsection, we provide more details on the frequency bias, one of the two temporal biases (the other being the periodicity bias; see Appendix C.2), from section 2. Our discussion is split into several parts. First, we prove Theorem 1 and discuss some of its consequences (Appendix C.1.1). Then, we provide more results on the attention matrices in a Transformer with patching to corroborate the frequency bias (Appendix C.1.2). Finally, we show the existence and non-existence of the frequency bias in two more interesting scenarios (Appendix C.1.3).

#### C.1.1  PROOF AND DISCUSSION OF THEOREM 1

THE PROOF

The proof of Theorem 1 is an application of concentration inequalities and matrix norm bounds once we realize that the inverse Fourier transform is an orthogonal transformation, mapping distinct Fourier modes to orthogonal subspaces in the patch space $\mathbb{R}^k$.

**Theorem 1′.** Let $k$ be a patch size, $1 < n < k$ be any positive integer, and $\omega \leq \min(k/n, n) - 1$ be a bandwidth. Let $B_1, B_2, \ldots, B_n \subset [k] := \{1, \ldots, k\}$ be $n$ mutually disjoint sets whose bandwidths are below $\omega$, i.e., $|B_i| \leq \omega$ for every $1 \leq i \leq n$. Let unit vectors $\mathbf{v}_1, \ldots, \mathbf{v}_n \in \mathbb{R}^k$ be $n$ patches that belong to the same band, i.e., $\text{supp}(\hat{\mathbf{v}}_i) \subset B_j$ for all $1 \leq i \leq n$ and a common $j$. Let unit vectors $\mathbf{u}_1, \ldots, \mathbf{u}_n \in \mathbb{R}^k$ be $n$ patches that belong to all different bands, i.e., $\text{supp}(\hat{\mathbf{u}}_i) \subset B_i$ for all $1 \leq i \leq n$. Let $\mathbf{V} = \begin{bmatrix} \mathbf{v}_1 & \cdots & \mathbf{v}_n \end{bmatrix} \in \mathbb{R}^{k \times n}$ and $\mathbf{U} = \begin{bmatrix} \mathbf{u}_1 & \cdots & \mathbf{u}_n \end{bmatrix} \in \mathbb{R}^{k \times n}$, and let $\phi(\mathbf{x}) = \mathbf{W}_2 \text{ReLU}(\mathbf{W}_1 \mathbf{x}) + \mathbf{b}$ be a neural network embedding, with $\mathbf{W}_1 \in \mathbb{R}^{m \times k}$, $\mathbf{W}_2 \in \mathbb{R}^{d \times m}$, and $\mathbf{b} \in \mathbb{R}^d$. The following two statements hold:

1. (Patches of similar frequencies are embedded in a shared subspace.) Let $0 < \varepsilon \leq 1$ be given. Given any neural network width $m$ and any $\mathbf{W}_1$ and $\mathbf{W}_2$ such that $\|\phi(\mathbf{V})\|_2 = \Omega(1) \|\mathbf{W}_2\|_2 \|\mathbf{W}_1\|_2 \|\mathbf{V}\|_2$ and any bias term $\mathbf{b}$ such that $\|\mathbf{b}\,\mathbf{1}_n^\top\|_F = \mathcal{O}(1) \|\mathbf{W}_2 \text{ReLU}(\mathbf{W}_1 \mathbf{V})\|_F$, we have

$$\text{rank}_\varepsilon(\phi(\mathbf{V})) := |\{j \mid \sigma_j(\phi(\mathbf{V})) > \varepsilon \sigma_1(\phi(\mathbf{V}))\}| = \mathcal{O}(1) \varepsilon^{-2} \omega,$$
$$\text{stab-rank}(\phi(\mathbf{V})) := \|\phi(\mathbf{V})\|_F / \|\phi(\mathbf{V})\|_2 = \mathcal{O}(\omega).$$

2. (Patches of different frequencies are embedded in nearly orthogonal subspaces.) There exists a universal constant $\varepsilon > 0$ so that the following statement holds. Assume that $\mathbf{W}_1$ is a random Gaussian matrix whose entries follow i.i.d. $\mathcal{N}(0, \alpha)$ distribution and $\mathbf{W}_2$ is a random Gaussian matrix whose entries follow i.i.d. $\mathcal{N}(0, \beta)$ distribution, where $\alpha$ and $\beta$ are any positive numbers. Set the bias term to be $\mathbf{b} = -\mathbf{W}_2 \mathbf{1}_m / \sqrt{2\pi}$. Then, for any $0 < \delta < 1$, $m = \Omega(\log(1/\delta))$, $d = \Omega(\log(n/\delta))$, $m > n$, and $m/d = \Theta(1)$, with probability no less than $1 - \delta$, we have

$$\text{rank}_\varepsilon(\phi(\mathbf{U})) = \Omega(n), \qquad \text{stab-rank}(\phi(\mathbf{U})) = \Omega(n).$$

Moreover, the constants in all $\mathcal{O}$ and $\Omega$-notations are universal.

*Proof of Theorem 1′.* We prove the two statements separately.

**Proof of the First Statement.** If the supports of $\hat{\mathbf{v}}_1, \ldots, \hat{\mathbf{v}}_n$ all belong to a set $B_j$ with a cardinality no more than $\omega$, then the vectors $\hat{\mathbf{v}}_1, \ldots, \hat{\mathbf{v}}_n$ are contained in a subspace of dimension $\leq \omega$. Since the discrete Fourier transform is an orthogonal operator, the vectors $\mathbf{v}_1, \ldots, \mathbf{v}_n$ are also contained in a subspace of dimension $\leq \omega$. Therefore, we have that $\text{rank}(\mathbf{V}) \leq \omega$ and that

$$\|\mathbf{V}\|_F \leq \sqrt{\sigma_1(\mathbf{V})^2 + \cdots + \sigma_n(\mathbf{V})^2} \leq \sqrt{\omega\,\sigma_1(\mathbf{V})^2} = \sqrt{\omega}\,\sigma_1(\mathbf{V}).$$

Since the ReLU activation function cannot increase the norm of an input matrix, we therefore have

$$\|\Phi(\mathbf{V})\|_F \leq \|\mathbf{W}_2 \operatorname{ReLU}(\mathbf{W}_1 \mathbf{V})\|_F + \|\mathbf{b}\,\mathbf{1}_n^\top\|_F$$
$$= \mathcal{O}(1)\|\mathbf{W}_2 \operatorname{ReLU}(\mathbf{W}_1 \mathbf{V})\|_F \leq \mathcal{O}(1)\|\mathbf{W}_2\|_2\|\operatorname{ReLU}(\mathbf{W}_1 \mathbf{V})\|_F$$
$$\leq \mathcal{O}(1)\|\mathbf{W}_2\|_2\|\mathbf{W}_1\mathbf{V}\|_F \leq \mathcal{O}(1)\|\mathbf{W}_2\|_2\|\mathbf{W}_1\|_2\|\mathbf{V}\|_F \leq \mathcal{O}(\sqrt{\omega})\|\mathbf{W}_2\|_2\|\mathbf{W}_1\|_2\|\mathbf{V}\|_2.$$

By our assumption, we have that $\sigma_1(\Phi(\mathbf{V})) \geq \Omega(1)\|\mathbf{W}_2\|_2\|\mathbf{W}_1\|_2\|\mathbf{V}\|_2$, and the statement about the stable rank follows immediately. Thus, we have

$$\operatorname{rank}_\varepsilon(\Phi(\mathbf{V})) \leq \frac{\sum_{j=1}^n \sigma_j(\Phi(\mathbf{V}))^2}{\varepsilon^2 \sigma_1(\Phi(\mathbf{V}))^2} \leq \frac{\|\Phi(\mathbf{V})\|_F^2}{\varepsilon^2 \sigma_1(\Phi(\mathbf{V}))^2} = \left(\frac{\|\Phi(\mathbf{V})\|_F}{\varepsilon \sigma_1(\Phi(\mathbf{V}))}\right)^2$$
$$\leq \left(\frac{\sqrt{\omega}\|\mathbf{W}_2\|_2\|\mathbf{W}_1\|_2\|\mathbf{V}\|_2}{\varepsilon \Omega(1)\|\mathbf{W}_2\|_2\|\mathbf{W}_1\|_2\|\mathbf{V}\|_2}\right)^2 = \mathcal{O}(1)\varepsilon^{-2}\omega.$$

**Proof of the Second Statement.** We will break the proof into three steps. In the first step, we lower-bound the Frobenius norm of $\Phi(\mathbf{U})$. In the second step, we upper-bound the spectral norm of $\Phi(\mathbf{U})$. This gives us an estimate of the number of large singular values, which we formally prove as the last step.

- **Step I:** Since the ReLU activation function is homogeneous on $[0, \infty)$, we may assume, without loss of generality, that $\alpha = \beta = 1$. Since the vectors $\hat{\mathbf{u}}_1, \ldots, \hat{\mathbf{u}}_n$ are supported on mutually disjoint sets of indices, they are orthogonal to each other. Moreover, since each of them is a unit vector, they are orthonormal. Since the inverse discrete Fourier transform is an orthogonal operator, we have that the matrix $\mathbf{U}$ has orthonormal columns. Therefore, the matrix $\mathbf{W}_1 \mathbf{U}$ is still a random Gaussian matrix whose entries are i.i.d. distributed like $\mathcal{N}(0, 1)$. That is, we clearly have that

$$\mathbb{E}\left[\|\operatorname{ReLU}(\mathbf{W}_1 \mathbf{U})\|_F^2\right] = \frac{1}{2}mn,$$

where the $1/2$ comes from the fact that ReLU fires each neuron with a probability of $1/2$. Moreover, by estimating the lower tail of the $\chi^2$ distribution, we know that for some $m = \Omega(\log(1/\delta))$, with probability no less than $1 - \delta/5$, we have that

$$\|\operatorname{ReLU}(\mathbf{W}_1 \mathbf{U})\|_F^2 \geq \frac{3}{7}mn. \tag{10}$$

From now on, we instantiate $\mathbf{W}_1$ and assume that eq. (10) is achieved. Let $\mathbf{Z} = \operatorname{ReLU}(\mathbf{W}_1 \mathbf{U})$, $N_1 = \|\mathbf{Z}\|_F$, and let $\mathbf{Y} = \mathbf{W}_2 \mathbf{Z}$. Let $\mathbf{w}_i^\top$ be the $i$th row of the matrix $\mathbf{W}_2$. We have

$$\mathbb{E}[\|\mathbf{Y}\|_F^2] = \mathbb{E}[\|\mathbf{W}_2 \mathbf{Z}\|_F^2] = \mathbb{E}\left[\sum_{i=1}^d \|\mathbf{w}_i^\top \mathbf{Z}\|_2^2\right] = \sum_{i=1}^d \mathbb{E}\left[\|\mathbf{w}_i^\top \mathbf{Z}\|_2^2\right]$$
$$= d\mathbb{E}\left[\|\mathbf{w}_1^\top \mathbf{Z}\|_2^2\right] = d\sum_{j=1}^n \mathbb{E}\left[(\mathbf{w}_1^\top \mathbf{z}_j)^2\right].$$

However, since $\mathbf{w}_1^\top \sim \mathcal{N}(\mathbf{0}, \mathbf{I}_k)$, we have

$$\mathbb{E}\left[(\mathbf{w}_1^\top \mathbf{z}_j)^2\right] = \|\mathbf{z}_j\|_2^2.$$

This gives us

$$\mathbb{E}[\|\mathbf{Y}\|_F^2] = d\sum_{j=1}^n \mathbb{E}\left[(\mathbf{w}_1^\top \mathbf{z}_j)^2\right] = d\sum_{j=1}^n \|\mathbf{z}_j\|_2^2 = d\|\mathbf{Z}\|_F^2.$$

By another Chernoff bound, we have that for $d = \Omega(\log(n/\delta))$, with probability no less than $1 - \delta/5$, that

$$\|\mathbf{Y}\|_F^2 \geq \Omega(1)d\|\mathbf{Z}\|_F^2 \geq \frac{2}{5}dmn. \tag{11}$$

Clearly, for $d = \Omega(\log(n/\delta))$, with probability no less than $1 - \delta/5$, we have

$$\|\mathbf{W}_2 \mathbf{1}_m\|_2^2 \leq \frac{6}{5}dm \quad \implies \quad \|\mathbf{W}_2 \mathbf{1}_{m \times n}/\sqrt{2\pi}\|_F^2 \leq \frac{6}{5 \times 2\pi}dmn. \tag{12}$$

By a union bound, with probability no less than $1 - 3\delta/5$, we have that eq. (10), (11), and (12) hold. When these hold, the triangle inequality of the Frobenius norm gives us

$$\|\Phi(\mathbf{U})\|_F^2 \geq \|\mathbf{Y}\|_F^2 - \|\mathbf{W}_2\mathbf{1}_{m \times n}/\sqrt{2\pi}\|_F^2 \geq \left(\frac{2}{5} - \frac{3}{5\pi}\right) dmn > \frac{1}{5} dmn. \tag{13}$$

- **Step II:** We can write the output of the neural network as

$$\Phi(\mathbf{U}) = \mathbf{W}_2 \text{ReLU}(\mathbf{W}_1 \mathbf{U}) - \frac{1}{\sqrt{2\pi}} \mathbf{W}_2 \mathbf{1}_{m \times n} = \mathbf{W}_2 \left( \underbrace{\text{ReLU}(\mathbf{W}_1 \mathbf{U}) - \frac{1}{\sqrt{2\pi}} \mathbf{1}_{m \times n}}_{\mathbf{R}} \right).$$

First, since $\mathbf{W}_2$ is a random Gaussian matrix whose entries are i.i.d. $\mathcal{N}(0, 1)$, for $d = \Omega(\log(n/\delta))$ and $m = \Omega(\log(1/\delta))$ defined above and with probability no less than $\delta/5$, we have

$$\|\mathbf{W}_2\|_2 = \mathcal{O}(1)(\sqrt{d} + \sqrt{m}). \tag{14}$$

Now, consider the matrix $\mathbf{R}$. Since $\mathbf{U}$ has orthonormal columns, we know that the matrix $\mathbf{W}_1\mathbf{U}$ still has i.i.d. entries, and so does $\mathbf{R}$. Moreover, each entry in $\mathbf{R}$ clearly has a sub-Gaussian distribution. Let $R$ be an entry of $\mathbf{R}$, then we have

$$\mathbb{E}[R] = \mathbb{E}\left[\text{ReLU}(Z) - \frac{1}{\sqrt{2\pi}}\right] = \mathbb{E}\left[\text{ReLU}(Z)\right] - \frac{1}{\sqrt{2\pi}} = 0,$$

where $Z \sim \mathcal{N}(0, 1)$ and it is well-known that $\mathbb{E}\left[\text{ReLU}(Z)\right] = 1/\sqrt{2\pi}$. Since we assumed that $m = \Omega(\log(1/\delta))$, with probability no less than $1 - \delta/5$, we have

$$\|\mathbf{R}\|_2 = \mathcal{O}(1)(\sqrt{m} + \sqrt{n}). \tag{15}$$

By a union bound, eq. (14) and (15) hold with a probability no less than $1 - 2\delta/5$. When these two equations hold, we have by the sub-multiplicativity of the spectral norm that

$$\|\Phi(\mathbf{U})\|_2 = \|\mathbf{W}_2\mathbf{R}\|_2 \leq \|\mathbf{W}_2\|_2\|\mathbf{R}\|_2 = \mathcal{O}(1)(\sqrt{dm} + m) = \mathcal{O}(1)\sqrt{dm}, \tag{16}$$

where the second last step comes from the fact that $n < m$ and the last step follows from the fact that $d/m = \Theta(1)$.

- **Step III:** By a final union bound, we can assume that eq. (13) and (16) hold with a probability of no less than $1 - \delta$. Let $\sigma_1, \ldots, \sigma_n$ be the singular values of $\Phi(\mathbf{U})$. Then, we have that

$$\frac{\sigma_1^2 + \cdots + \sigma_n^2}{\sigma_1^2} = \frac{\|\Phi(\mathbf{U})\|_F^2}{\|\Phi(\mathbf{U})\|_2^2} = \Omega(1)\frac{dmn}{dm} = \Omega(n).$$

Since the constant in the $\Omega$-notation is universal, there must exist a universal $\varepsilon > 0$ such that at least $\Omega(n)$ of the relative singular values $\sigma_j/\sigma_1 = \sigma_j^2/\sigma_1^2$ are $> \varepsilon$. The proof is complete.

$\square$

INTERPRETATION AND DISCUSSION OF TECHNICAL DETAILS

A pictorial description of Theorem 1 can be found in Figure 10.

As a next step, notice that our theorem relies on a list of crucial assumptions. We anticipate some questions on them, and now we discuss why certain assumptions are made.

- *Why do we assume that the support of a vector $\mathbf{u}$ or $\mathbf{v}$ is strictly in a band?*

  We make this assumption for the simplicity of the proof and the cleanliness of the final statement. In practice, when $\omega$ is small, one hardly encounters a patch whose support is strictly limited within a narrow band; instead, it is often the case that a patch has its Fourier coefficients dense in a band and are supported with small values elsewhere. To prove a result in this case, one can view these patches as small perturbations of strictly band-limited patches, and a statement will follow using a Weyl-type perturbation analysis.

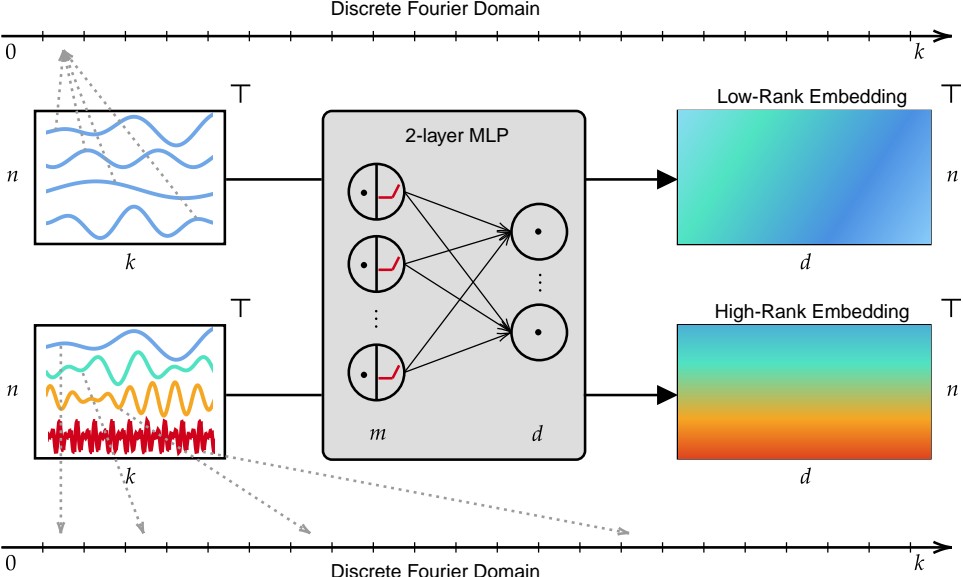

**Figure 10:** An illustration of Theorem 1: if we sample a number of patches from the same narrow frequency band, then they are embedded into a small subspace; while if we sample the patches with different frequency contents, then they are embedded into different subspaces of the hidden space.

- *In the first statement, why do we assume that $\|\Phi(\mathbf{V})\| = \Omega(1)\|\mathbf{W}_2\|_2\|\mathbf{W}_1\|_2\|\mathbf{V}\|_2$?*

  If we assume no ReLU activation is used, then this condition simply reduces to the general tightness of the sub-multiplicativity of the spectral norm. However, introducing the ReLU activation function usually only increases the spectral norm of the output. Hence, the lower bound is natural to assume.

- *In the first statement, why do we assume that $\|\mathbf{b}\mathbf{1}_n^\top\|_F = \mathcal{O}(1)\|\mathbf{W}_2 ReLU(\mathbf{W}_1\mathbf{V})\|_F$?*

  This assumption simply says that when computing the output of a two-layer ReLU-activated MLP, we are not asymptotically dominated by the bias term, which is a natural one to make.

- *In the second statement, why do we assume that $\mathbf{b} = -\mathbf{W}_2\mathbf{1}_m/\sqrt{2\pi}$?*

  This is perhaps the most arguable assumption that we have made. It is a technical assumption that we will further empirically justify in our numerical experiments. At a high level, if we assume that $\mathbf{W}_1$ and $\mathbf{W}_2$ are random Gaussian matrices, the ReLU's output $ReLU(\mathbf{W}_1\mathbf{V})$ does not have a mean of $0$. Since it does not have a mean of zero, in expectation, it will introduce a rank-1 perturbation in the output of the MLP. The bias term is there to recenter the intermediate output $ReLU(\mathbf{W}_1\mathbf{V})$ to avoid the rank-1 component. Without the bias term, the output will be the sum of a large rank-1 matrix and a full-rank matrix, making our statement about the stable rank invalid, but the statement about the $\varepsilon$-rank remains true.

NUMERICAL EXPERIMENT

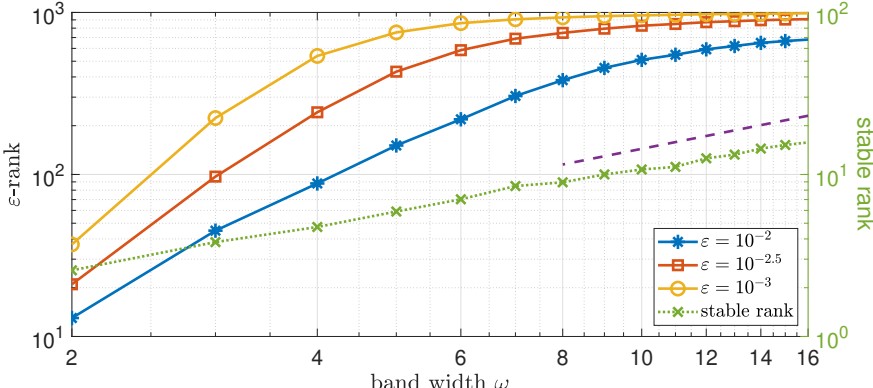

**Figure 11:** The $\varepsilon$-ranks and stable ranks of the output matrix of a randomly sampled MLP layer, given an input matrix sampled from a single frequency band whose width is limited by $\omega$. As $\omega$ increases, the ranks of the output matrix grow. The reference line has a slope of $1$ in this log-log plot.

We conclude our discussion of Theorem 1 by presenting three numerical experiments to corroborate the theorem statement and elaborate on some technical assumptions made. Our first experiment validates that the bandwidth $\omega$ limits the rank of the input embedding. For this experiment, we set $d = 1024$, $m = 4096$, $n = 1024$, and $k = 64$. Our MLP is obtained by randomly sampling Gaussian matrices $\mathbf{W}_1$ and $\mathbf{W}_2$ and defining the bias term to be the centering factor $\mathbf{W}_2\mathbf{1}_m$ as explained earlier. For a fixed bandwidth $\omega$, we sample $n$ patches whose supports are contained in a fixed band of width $\omega$. We then simulate the two-layer MLP to compute the output matrix.

In Figure 11, we see that as $\omega$ increases, the stable rank of the output matrix follows the linear growth, which corroborates the first statement in Theorem 1. The $\varepsilon$-ranks are slightly more complicated because they are upper bounded by $\min(d, n)$ and saturate easily. In practice, we see that the growth of the $\varepsilon$-ranks is initially faster than linear and eventually slows down.

In the second experiment, we compare the two statements made in Theorem 1. To this end, we design an experiment, where we fix the bandwidth to $\omega = 2$. We still use $d = 1024$, $m = 4096$, and $k = 64$. However, we change $n$, the number of patches sampled. For each $n$, we sample these patches in two ways:

- **A sampling that resembles the first statement:** we sample the $n$ patches from a fixed single band that has a width of $\omega$.

- **A sampling that resembles the second statement:** we sample the $n$ patches from $n$ mutually disjoint bands in the Fourier domain. Note that this requires $\omega \cdot n \geq k$, which is the reason why we choose a small $\omega$ for this experiment.

We then simulate the two-layer MLP sampled in the same way as explained above.

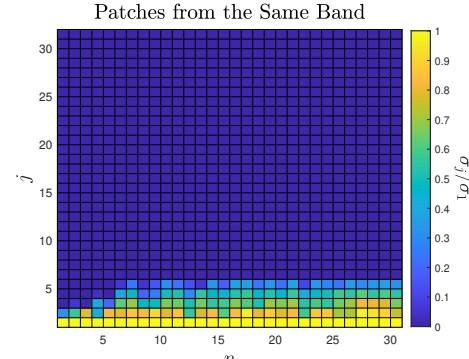 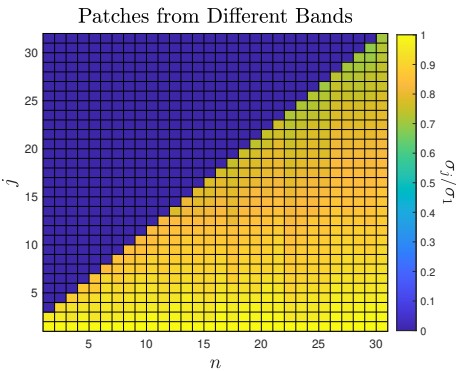

**Figure 12:** The relative singular values of the output matrix. On the left, we sample all $n$ patches in the input matrix from the same $\omega$-band in the Fourier domain. On the right, we sample $n$ patches from the $n$ mutually disjoint $\omega$-bands in the Fourier domain.

From Figure 12, we see that if we sample the $n$ patches from the same band, then there is a limited number of large singular values, no matter how large $n$ is. This corroborates the first statement. On the other hand, if we sample the $n$ patches from $n$ different Fourier bands, and therefore inject different frequencies into each of them, then all singular values are fairly large — in fact, none of the singular values are below $\sigma_1/2$ — indicating that the output matrix is numerically full-rank. This corroborates the second statement in Theorem 1.

The last experiment we show concerns the necessity of the bias term **b** in the second statement of Theorem 1. As explained earlier, this term is used to centralize the intermediate output. In this experiment, we do not use the bias term, and we repeat the experiment shown in Figure 12. We see that compared to the case with the bias term, the output matrices are essentially the sum of a large rank-1 matrix and a smaller full-rank matrix. That is, for a reasonably small $n$ and a reasonably large $\varepsilon$, the $\varepsilon$-rank of the output matrix is still large and growing with $n$.

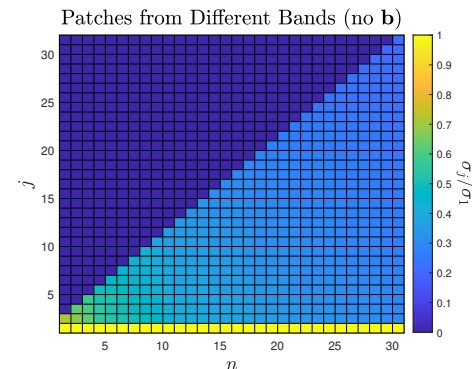 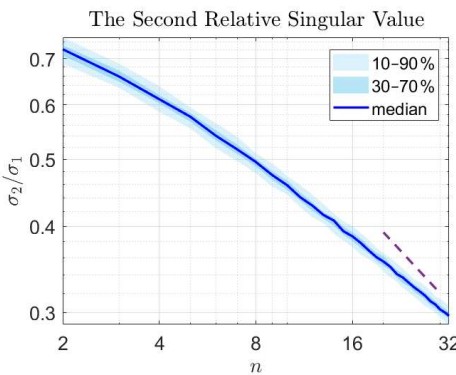

**Figure 13:** On the left, we reproduce the experiment in Figure 12, but without the bias term **b** in the neural network. We see that removing the bias term makes the full-rank output a sum of a large rank-one matrix and a small full-rank matrix. On the right, we show the gap between the rank-one term and the full-rank term as the number of sampled patches $n$ increases, by measuring the second relative singular value $\sigma_2/\sigma_1$ of the output matrix. The reference line has a slope of $-1/2$.

As $n$ increases, one needs to set $\varepsilon$ smaller to still have most of the relative singular values of the output larger than $\varepsilon$. In fact, theoretically, it is relatively easy to show that we need $\varepsilon \sim 1/\sqrt{n}$, so that the $\varepsilon$-rank of the output matrix, without the bias term $\boldsymbol{b}$, is proportional to $n$ as it increases. This is still much larger than the output of patches from the same band. In Figure 13, we also repeat this experiment for $100$ randomly sampled neural networks without the bias term, and we show the second relative singular value $\sigma_2/\sigma_1$ of the output matrix, which measures the gap between the large

rank-one matrix and the small full-rank matrix. We see that this gap seems to grow slightly slower than $\sqrt{n}$, which is suggested by the theory.

### C.1.2 VISUALIZATION OF ATTENTION SCORES

Recall that, in Figure 3, we saw that the low- and high-frequency patches are embedded in two nearly orthogonal subspaces of the hidden space. It is also the case that the attention scores corresponding to the low-frequency patches are much larger than those corresponding to the high-frequency ones. In Figure 14, we show the magnitudes of the averaged (across all heads) attention scores in the first layer of the Chronos-Bolt's encoder. The first one shows the attention scores when we give $\sin(8t)$ as the input, and the second one shows the attention scores when we input $\sin(117t)$. We can see that since the attention kernel $\mathbf{W}_Q^\top \mathbf{W}_K$ is much better aligned with the low-frequency patches, the scores in the first panel have a much higher order of magnitude than the scores in the second panel. Therefore, when we superpose the two Fourier modes to form the composite input, the attention scores mainly follow the periodicity of $8$ instead of $117$.

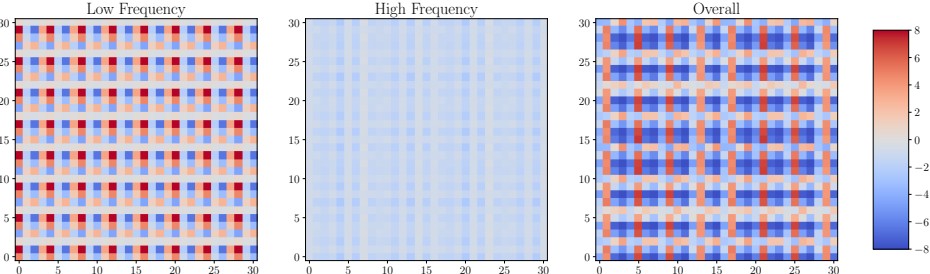

**Figure 14:** The attention scores before softmax, i.e., $\mathbf{Q}^\top \mathbf{K} \in \mathbb{R}^{L \times L}$, where $L$ is the length of the input sequence (i.e., the number of patches). We see that overall, if the patches are low-frequency, then the attention scores have much larger magnitudes than if the patches are high-frequency. Therefore, when a signal contains both low-frequency content and high-frequency one, the attention scores are dominated by the (periodicity of) the low-frequency information.

### C.1.3 ADDITIONAL EVIDENCE: SUPERPOSING TWO FREQUENCIES WITHOUT ALIASING

Recall that, in Figure 3, we saw an example where our context is sampled from the signal

$$f(t) = \sin(8t) + \sin(117t).$$

In that case, we know that if we use a patch size of $k = 16$, then the low- and high-frequency information is stored separately into different subspaces. Moreover, since the attention scores are dominated by the low-frequency information, the high-frequency patterns are ruined during attention, resulting in a poor learning capability of the high-frequency information.

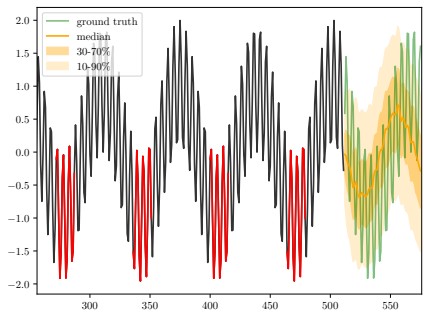 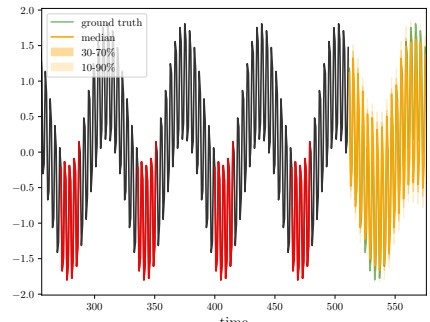

**Figure 15:** On the left, we show the forecast of a Chronos-Bolt model (with patch size $k = 16$) where the periodicity of high-frequency information in the context does not align with that of the low-frequency information. On the right, we show the forecast when the periodicities of the high- and low-frequency align. We use a red color to indicate the same motif in the low-frequency sinusoidal wave, where we see that the high-frequency information in different motifs looks very different on the left, while it looks the same on the right.

Here, we show one additional example where if we change the high-frequency mode by a little bit, then we can completely change the models' performance. In particular, consider the following new context:

$$f(t) = \sin(8t) + \sin(128t).$$

In some sense, 128 is even "larger" than 117; but when we sample this signal and use a Chronos-Bolt model with $k = 16$ to predict, we see that the performance becomes much better (see Figure 15)! What makes that happen? Notice that 128 is an integral multiple of 8, but 117 is not. Therefore, even if the high-frequency information is attended in the way dominated by the low-frequency, it does not cause any issue in preserving the high-frequency information. That is, we do not suffer from aliasing errors. (As an analogy, when you use your cellphone's camera to take a picture of the screen of your television, you often notice some stripes — this is due to the mismatch between the resolutions of your digital camera and the television screen — but there are certain cases where the stripes do not appear, which happen exactly when the resolutions match each other.) For example, in Figure 15, we use red colors to indicate the same motif in each period of the low-frequency wave $\sin(8t)$. On the left, when the periodicities of $\sin(8t)$ and $\sin(117t)$ do not match, we see that the high-frequency information in each motif is different; but since the periodicities of $\sin(8t)$ and $\sin(128t)$ match each other, we have exactly the same high-frequency content in each red motif.

The significance of this example is two-fold:

1. First, it validates our hypothesis that the attention scores are mainly driven by the low-frequency content that results in the frequency bias when using a large patch size.

2. Second, it shows that the frequency bias is not a guaranteed notion. When the periodicities of the high- and low-frequency match each other, its impact is significantly reduced.

**Another Use Case of the Frequency Bias: Concatenation.** Previously, our discussion mainly focused on the case where we have a superposition of a low- and high-frequency, because this is usually the way that people care about the frequency bias. However, there is another way that two different frequencies can come into the same context — via concatenation.

 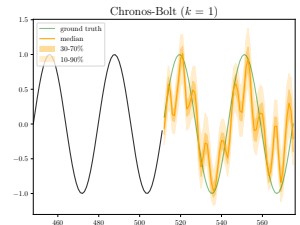 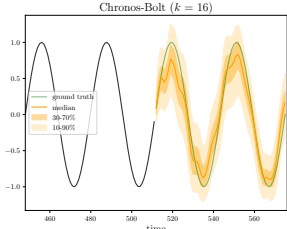

**Figure 16:** We show the forecast of two models, Chronos-Bolt with a patch size $k = 1$ and one with $k = 16$, given a context formed by concatenating low-frequency modes and high-frequency ones. We see that the high-frequency information significantly bleeds into Chronos-Bolt's forecast when $k = 1$, but the forecast is better when the patch size $k = 16$.

## Angles between Embedded Vectors

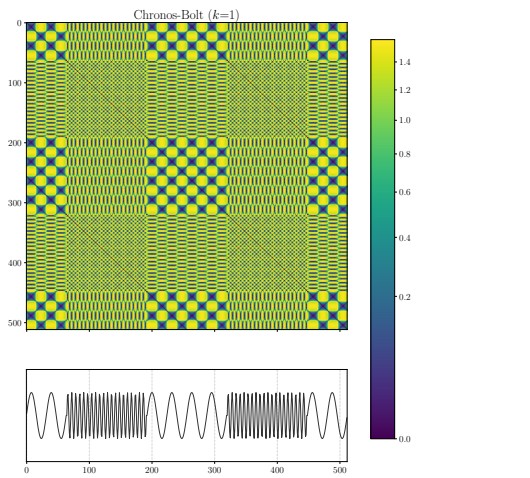 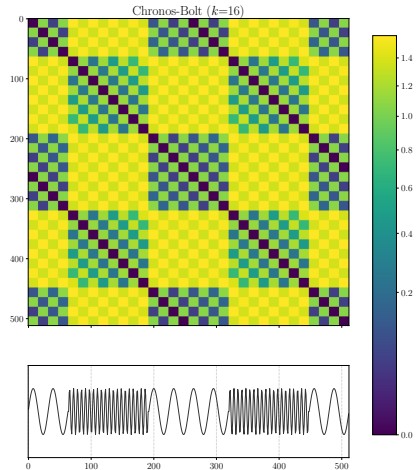

## Magnitude of Attention Scores

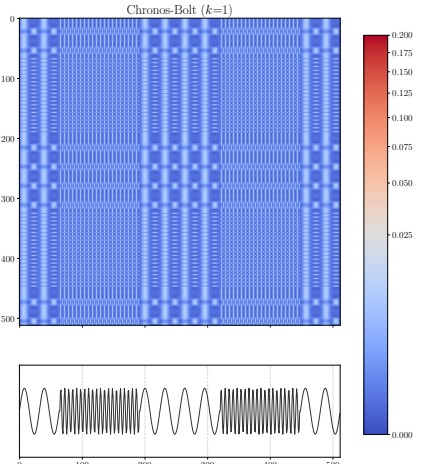 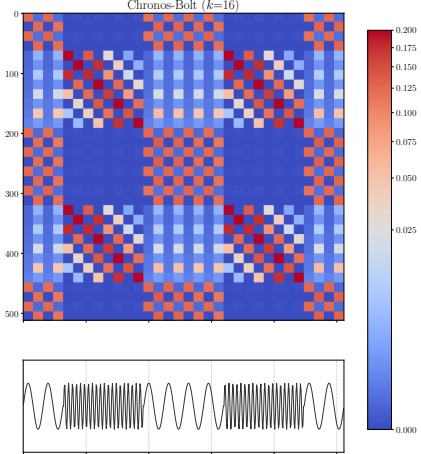

**Figure 17:** We show the angles between different pairs of embedded patches in the hidden space, given the context in Figure 16, as well as the magnitudes of the attention scores.

In Figure 16, we show the forecast of Chronos-Bolt models, with a patch size of $k = 1$ and 16, respectively, given an input formed by concatenating low- and high-frequency motifs. In this setting, we see that having a larger patch size helps prevent the high-frequency information from bleeding into the low-frequency forecast.

This finding is not surprising, and it provides yet another example of Theorem 1. The fact that high- and low-frequency patches are orthogonal to each other in the hidden space means that they are unlikely to interact with each other in a Transformer (see the top row of Figure 17). From the attention scores shown in Figure 17, we see that with a patch size of 1, a lot of the high-frequency information is attended to the low-frequency information. However, when $k = 16$, the high-frequency patches mainly talk to the high-frequency ones, and similarly for the low-frequency patches, but the low-frequency patches rarely attend to the high-frequency ones. This explains why we have less contamination using a larger patch size.

### C.1.4 Development of the Frequency Bias During Training

In the paper, we see that the attention matrices in a pretrained Chronos-Bolt model are naturally better aligned with low-frequency patches. In this section, we show how the frequency bias evolves during training.

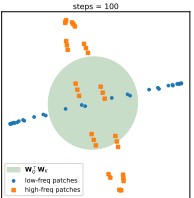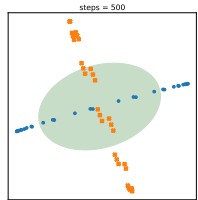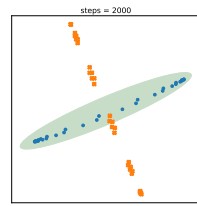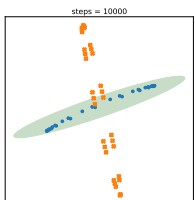

**Figure 18:** The evolution of the kernel weights $\mathbf{W}_Q^\top \mathbf{W}_K$ in Chronos-Bolt during training. For each step number shown in the figure, we embed many low-frequency patches along with many high-frequency patches. We then project these patches onto their leading two principal components in the hidden space. The light green Ellipse shows the projection of the attention kernel $\mathbf{W}_Q^\top \mathbf{W}_K$ in this two-dimensional space.

In Figure 18, we save a few checkpoints of Chronos-Bolt during training and project the embeddings of low- and high-frequency patches onto their leading principal components. We see a clear orthogonal relationship between the low-frequency embeddings and the high-frequency embeddings. In addition, Figure 18 shows how the weight matrices $\mathbf{W}_Q$ and $\mathbf{W}_K$ are trained to obtain the frequency bias.

### C.2 Details of the Periodicity Bias

In this subsection, we provide more details on the periodicity bias, one of the two temporal biases (the other being the frequency bias; see Appendix C.1), from section 2. Our discussion is split into two parts. First, we perform a more detailed analysis of the [REG] token's role in the preservation of the periodicity (Appendix C.2.1). Then, we briefly discuss how we measure the periodicity bias induced by patching (Appendix C.2.2).

### C.2.1 An Investigation of the [REG] Token

Recall Figure 4, where we noted that when the [REG] token is present, Chronos-Bolt's Transformer becomes significantly worse at preserving periodic patterns. To understand why this happens, we present a case study where our input is a sinusoidal wave. We pass this sinusoidal wave into Chronos-Bolt's encoder (patch size $k = 1$) and compute the context $\mathbf{z} \in \mathbb{R}^{d \times L}$ after applying the first attention layer. Here, we have $d = 512$ and $L = 512$. There are 8 heads, so the first head of $\mathbf{z}$ is a matrix $\mathbf{z}^{(1)} \in \mathbb{R}^{64 \times 512}$. We plot all 64 channels of $\mathbf{z}^{(1)}$ in Figure 19. We see clearly some channels where the periodicity of the input context is seriously affected. Those channels are highlighted in red.

Channels of the Encoded Context

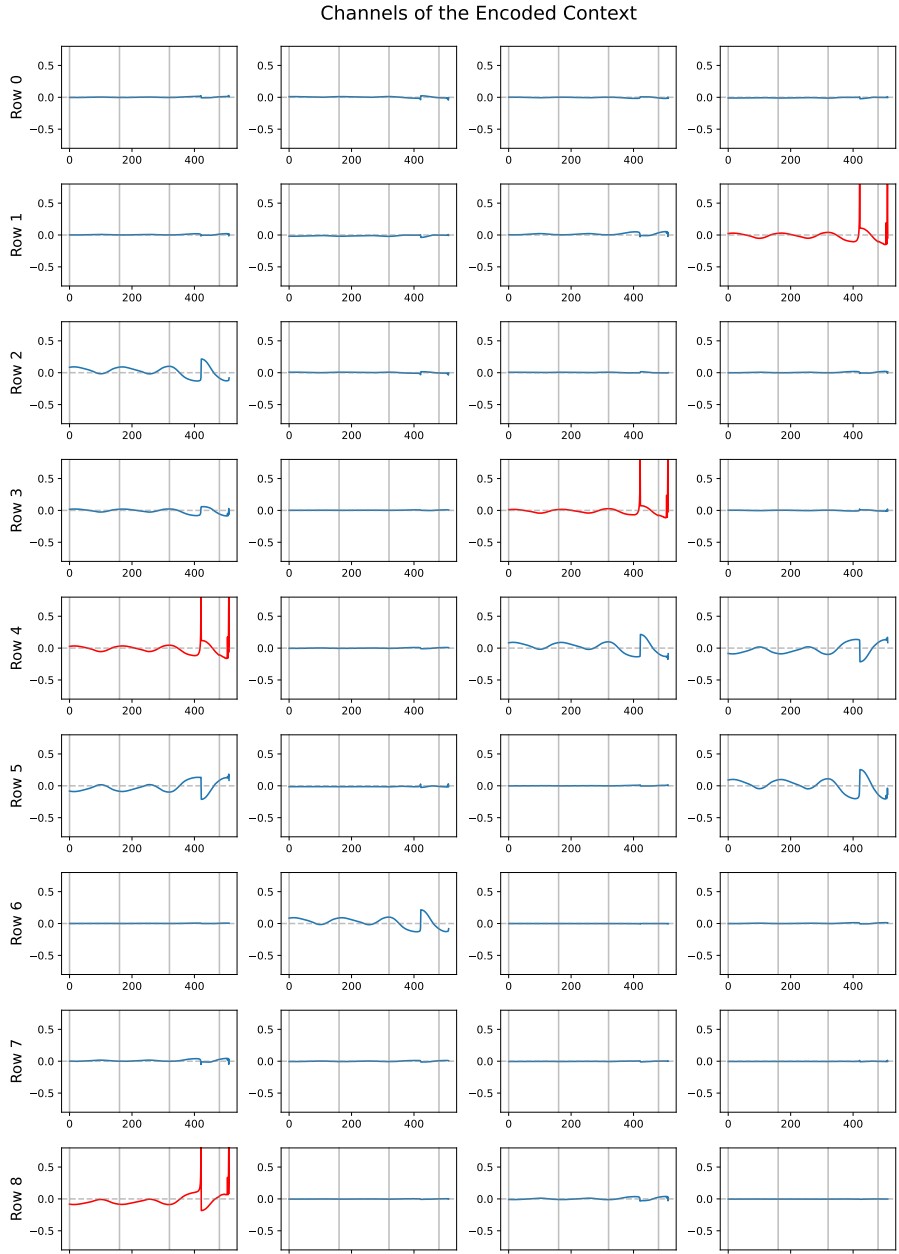

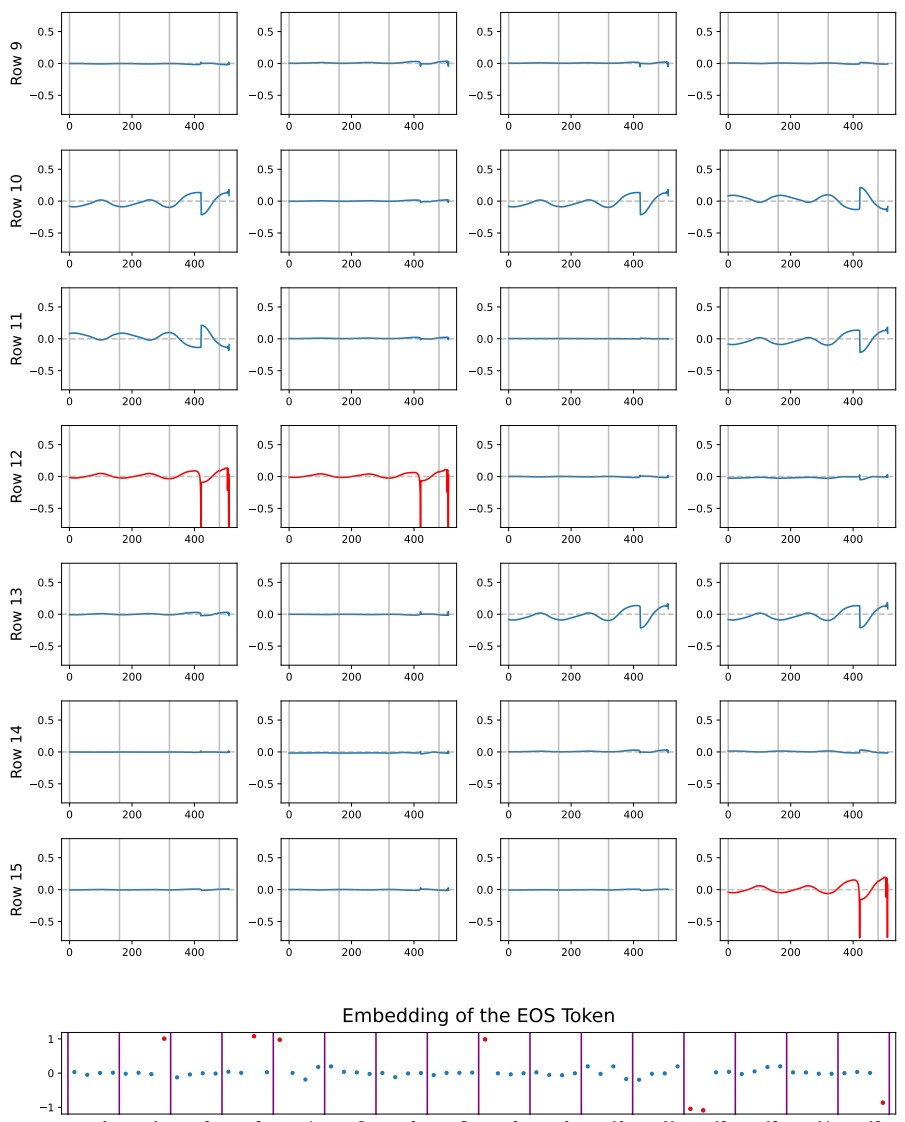

**Figure 19:** We give a sinusoidal wave to the encoder of a Chronos-Bolt model (patch size $k = 1$) and take the pre-$\mathbf{W}_O$ output from the first attention layer. We plot the $64$ channels in the first head of this output, in the row-major order. We see that the periodicity in several channels, highlighted in red, is much less-well preserved than in others. The bottom panel shows the $64$ values corresponding to the first head of the `[REG]` token in the first attention layer, where we see the large values correspond to exactly the very non-periodic channels.

Now, we claim that these serious losses of periodicity are primarily due to the existence of the `[REG]` token. In the bottom panel of Figure 19, we show all $64$ values corresponding to the `[REG]` token. We see that there are clearly a couple of them that are large in magnitude. More strikingly, there is a strong correlation between the magnitude of the `[REG]` token and the loss of periodicity. That is, the entries of large values in `[REG]` are exactly the channels in $\mathbf{z}^{(1)}$ that have a poor periodicity.

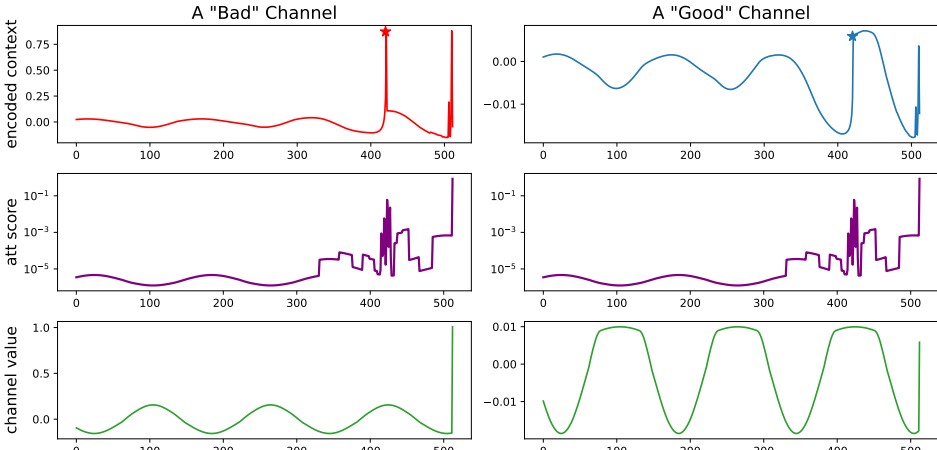

**Figure 20:** We show how a very non-periodic channel and a more periodic channel from Figure 19 are computed. On the top row, we plot these two channels, and the stars are the positions that we investigate in the bottom two panels of the figure. The starred position is computed by taking the inner product of the attention scores vector, shown in the middle row, and the value vector, shown in the bottom row. We see that the attention scores used to compute the starred position are particularly large at the `[REG]` token, and the large value of the `[REG]` token seriously destroys the periodicity of the signal.

This is not hard to understand. If the `[REG]` token has a large value, without masking, this value is written into a previous position that has a large attention score on it, perturbing the periodic pattern. To illustrate this with a clear example, we take a "bad" channel of $\mathbf{z}^{(1)}$, where the periodicity is severely destroyed, and a "good" channel. In Figure 20, we see that the main difference happens at the starred position, where in the bad channel, the output blows up at this position. In the middle row of Figure 20, we show the attention scores that are used to compute this position. We see that the last attention score is very large, meaning that a lot of the `[REG]` token's value is written into the starred position. More precisely, the starred position is computed by taking the inner product of the attention scores and the channel values:

$$\mathbf{z}_{k,j^*} = \sum_{i=1}^{L} s_{j^*}^i V_{k,i},$$

where $\left(s_{j^*}^1, \ldots, s_{j^*}^L\right)$ are the self-attention scores shown in purple and $(V_{k,1}, \ldots, V_{k,L})$ are the values in the $k$th channel shown in green. Since $s_{j^*}^L$ is large, whether the periodicity is well-preserved reduces to a question of whether $V_{k,L}$ is large. That is why we see the strong coupling of the periodicity and the magnitude of the `[REG]` token's value in Figure 19.

The question that remains is: does removing the `[REG]` token help? The answer to this is positive. In Figure 21, we show two models: in the first model, we compute Chronos-Bolt's encoded values as usual, which corresponds to exactly the figures shown in Figure 19; in the second model, we apply a masking to prevent the `[REG]` token's value from being written into the previous tokens, i.e., the `[REG]` token is write-only. We show both random channels before and after applying the channel mixing layer $\mathbf{W}_O$. Clearly, masking the `[REG]` token helps the preservation of the periodicity.

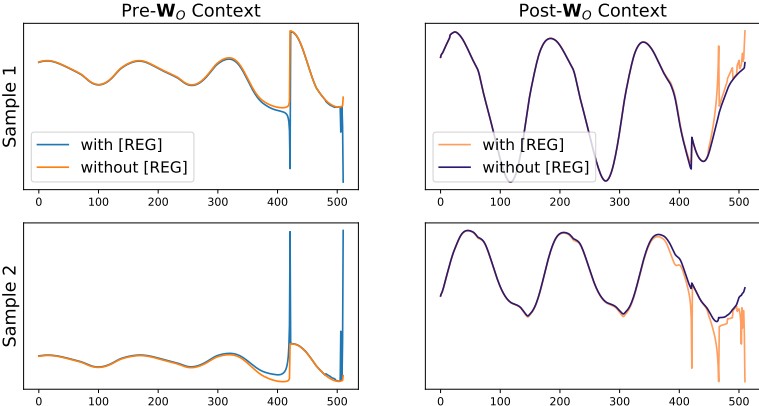

**Figure 21:** We give a sinusoidal wave to the encoder of a Chronos-Bolt model (patch size $k = 1$) and take the pre-$\mathbf{W}_O$ and post-$\mathbf{W}_O$ output from the first attention layer. We compute the output in two ways: using the default setting where every position is attended to another position; and applying a mask to prevent the value of the [REG] token from being written into the previous outputs. We see that with masking, the periodicity is much better preserved.

### C.2.2 DESIGN OF EXPERIMENTS IN FIGURE 4

Recall Figure 4, where we explained that patching also hides the periodic patterns. This is also not surprising. The simplest way to view this is by considering aliasing. If the patch's periodicity mismatches the periodicity of the input signal, then we lose periodic information by patching. To show this, we generate two histograms in Figure 4 as follows:

1. Randomly sample a sequence $\mathbf{x} \in \mathbb{R}^L$ from Chronos' training dataset (Ansari et al., 2024a), which includes both real-world and synthetic time series. Sample until we have $L \gg 64$.

2. Take the last $64$ elements of $\mathbf{x}$: $\mathbf{x}^* = \mathbf{x}_{(L-64+1):L}$.

3. Break the context into patches and form motifs of length 64: $\mathbf{x}^{(1)} = \mathbf{x}_{1:64}$, $\mathbf{x}^{(2)} = \mathbf{x}_{(1+k):(64+k)}$, $\mathbf{x}^{(3)} = \mathbf{x}_{(1+2k):(64+2k)}, \ldots$, where none of them overlaps $\mathbf{x}^*$.

4. Compute the $R^2$-score between $\mathbf{x}^*$ defined in step 2 and each $\mathbf{x}^{(i)}$ formed in step 3. We call the maximum $R^2$ score the "best matching score."

We repeat this step for over $10,000$ sampled sequences to form the histograms in Figure 4. Figure 22 shows more results for different patch sizes $k$.

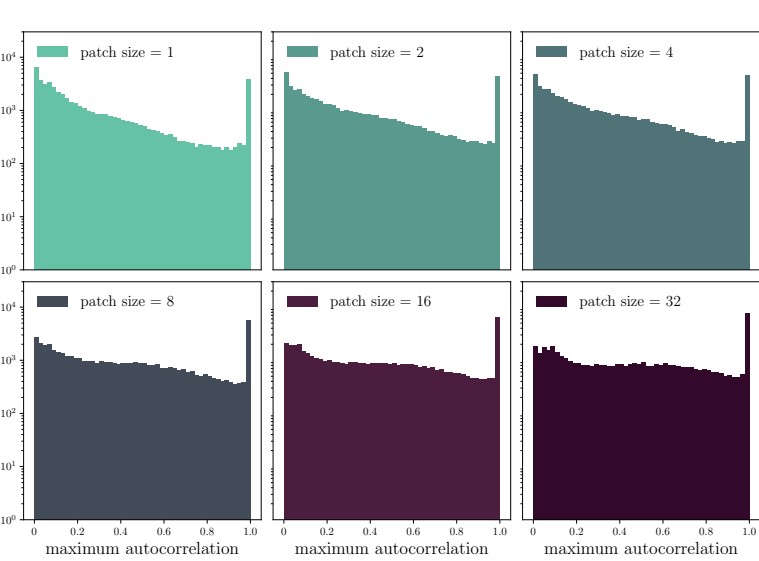

**Figure 22:** The maximum correlation between the last motif in a context and each motif in the earlier context, which are defined by patching. We see that as the patch size increases, correlations generally become smaller, indicating more challenges in identifying the periodicity in the context.

**Figure 23:** The minimum distance between the last motif in a context and each motif in the earlier context, which are defined by patching. We see that as the patch size increases, distances generally become larger, indicating more challenges in identifying the periodicity in the context.

In addition to the $R^2$-score, we also consider another metric, which measures the relative distance between the last motif and earlier ones. This is done by changing step 4 from above into the following:

4'. Compute the relative distance between $\mathbf{x}^*$ defined in step 2 and each $\mathbf{x}^{(i)}$ formed in step 3:

$$\|\mathbf{x}^* - \mathbf{x}^{(i)}\|_2/(\|\mathbf{x}^*\| + 10^{-8}),$$

where $10^{-8}$ is there to prevent a zero denominator.

In Figure 23, we see that as the patch size increases, distances generally become larger, indicating more challenges in identifying the periodicity in the context.

## D  DETAILS OF THE GEOMETRIC BIAS (FROM SECTION 3)

In this section, we provide more details on the geometric bias, which was discussed in Section 3. The discussion is made up of three parts: the *angles* between the embedded vectors, which lead to a *locality preference* (Appendix D.1); the *distance* between the embedded vectors, which gives rise to a *scale-dependent treatment* (Appendix D.2); and the *norm* of the embedded vectors, yielding the *offset-aware embedding* (Appendix D.3). This appendix is broken into three parts, corresponding to each of the three biases, respectively.

### D.1  DETAILS OF THE ANGULAR BIAS

#### D.1.1  DESIGN OF EXPERIMENTS IN FIGURE 5

Recall Figure 5, where we presented an experiment where we collected the self-attention scores of both the Chronos and Chronos-Bolt models, which revealed that the self-attention scores of the Chronos model have more of a bimodal distribution, while those of the Chronos-Bolt model followed a more unimodal distribution. To collect this set of self-attention scores, we sample 1000 random samples from Chronos and Chronos-Bolt's in-domain evaluation corpus, and we pass them into the two models to collect all self-attention scores. Due to the space limitation, however, we were only able to show the histogram in the log–log scale, which hid some useful information.

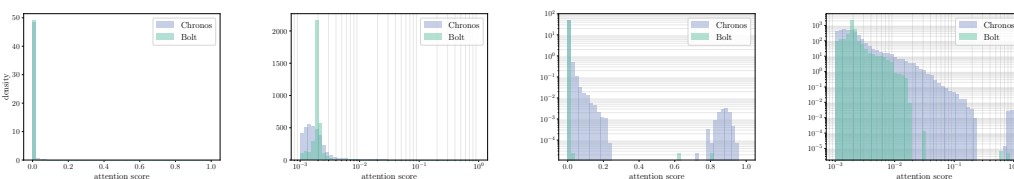

**Figure 24:** We reproduce the histograms in Figure 5 on four different scales: linear, semilog-x, semilog-y, and log–log, respectively. We can clearly see the bimodal distribution of Chronos' self-attention scores in the semilog-y plot, while the semilog-x plot highlights the existence of small self-attention scores in Chronos, compared to Chronos-Bolt's mediocre ones.

In Figure 24, we reproduce this histogram on all four common scales: the linear scale, the semilog-x scale, the semilog-y scale, and the log–log scale, respectively. From these plots, we can see at least two clean messages:

1. From the semilog-y plot, i.e., the third panel, we see a clear bimodal distribution in Chronos' self-attention scores, which does not appear in Chronos-Bolt's self-attention scores. Note that the modes in Chronos' bimodal distribution do not have the same weight — that is essentially why the two modes are hard to visualize when not using a logarithmic y-scale. This is not surprising, however, because if a single self-attention score is large, then it implies that the remaining $L-1$ singular values are all small, so the weight of the large-score mode must be much lower than the weight of the small-score mode.

2. From the semilog-x plot, i.e., the second panel, we see that the small attention scores in Chronos are much smaller than the attention scores in Chronos-Bolt. For Chronos-Bolt, most self-attention scores are small, but not tiny. This corroborates our intuition that Chronos-Bolt performs more context mixing by putting a relatively small and more uniformly distributed attention score in each element across the sequence.

#### D.1.2  THREE EXAMPLES TO ILLUSTRATE "PARROTING"

In the main article, and in previous work, e.g., Zhang & Gilpin (2025a;b), the idea of "parroting" arises. This basically says that Chronos, in contrast to Chronos-Bolt and other models that use a continuous embedding, tends to compute the output by "matching" the motif immediately prior to the forecasting start point to a motif in the earlier context, and then predicting by copying what comes immediately after. We identified that this "parroting" strategy (described in detail in Zhang &

Gilpin (2025a;b)) is mainly due to the angular bias of the model. Here, we provide three examples to clearly illustrate the idea and the mechanism of "parroting," and we suggest when it may be "good" or "bad."

**Example I: Parroting helps Chronos to pinpoint a periodic context.** The example in Figure 25 features a sinusoidal context wave. For this context, it is very easy for the Chronos model to parrot the context. In the top panels, we show the distribution of all cross-attention scores when Chronos and Chronos-Bolt (patch size $k = 1$) make the first forecast. In Chronos, self-attention scores are put very heavily and locally near the position corresponding to the forecast position in every period, whereas Chronos-Bolt puts the cross-attention scores more evenly across the entire context. As a result, one cannot eyeball a difference between Chronos' forecast and the ground truth, but Chronos-Bolt does not achieve such a pinpoint prediction.

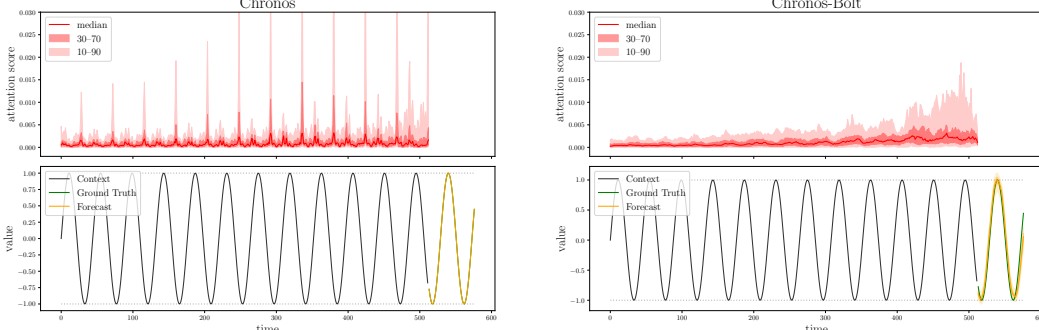

**Figure 25:** We show an example where Chronos' parroting strategy makes it perform better than Chronos-Bolt on a perfectly periodic signal. Above the forecast plots, we show the density of all cross-attention scores in the decoder of the models. We see that Chronos' cross-attention scores are highly dense where the next forecast corresponds in the earlier periods, while Chronos-Bolt's cross-attention scores are more evenly distributed.

**Example II: Parroting makes Chronos lazy and prevents advanced "reasoning."** In the next example, we still base our context on the same sinusoidal wave, but we scale it into a bidirectional envelope. This envelope is maximized in the middle and decays towards the two ends. The finesse of this design is that if a model naively performs parroting, then it is easy to match the last motif to a motif at the beginning, and predict the increasing context after it. By looking at the cross-attention scores, we note that this is exactly what Chronos does, and consequently, the signal it predicts has an increasing trend, while it should be decreasing. Chronos-Bolt, on the other hand, makes a relatively accurate prediction by attending mostly to the later context than the earlier ones to apply more advanced "reasoning."

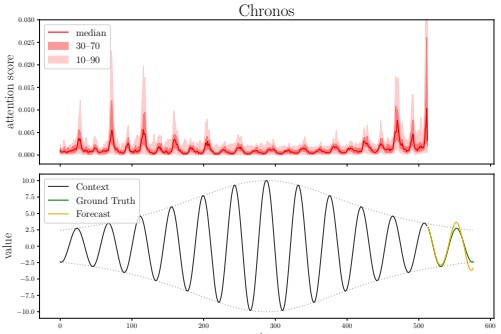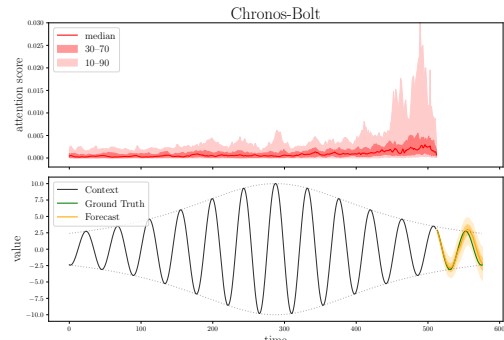

**Figure 26:** We show an example where Chronos' parroting strategy harms its performance. From the cross-attention scores, we see that Chronos forecasts the future by parroting the earlier match motif, but unfortunately, the earlier motif exhibits an increasing trend, while the true trend in the forecast should clearly be decreasing. Chronos-Bolt performs much better by not parroting and mainly focusing on the recent history.

**Example III: Chronos can still "reason" if there is nothing to parrot.** Looking at the second example, one may wonder: is it that Chronos cannot perform advanced "reasoning" so that it gets the trend wrong? To test this, we redesign our envelope: instead of having a bidirectional envelope, we use a unidirectional one that is monotonically decreasing. The major distinction here is that we do not have an increasing period at the beginning of the context, which therefore prevents Chronos from parroting. In Figure 27, we see that when Chronos has nothing to parrot, it attends to the recent history and gets the overall trend right; in this case, by simply "removing" the first decreasing part from our context, we can make Chronos perform as well as Chronos-Bolt again.

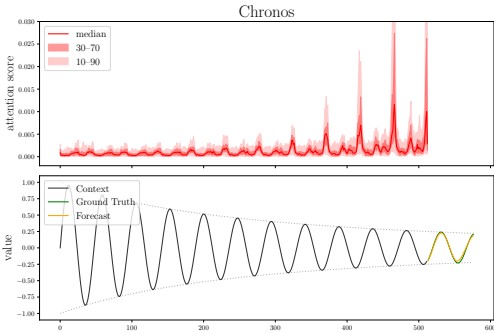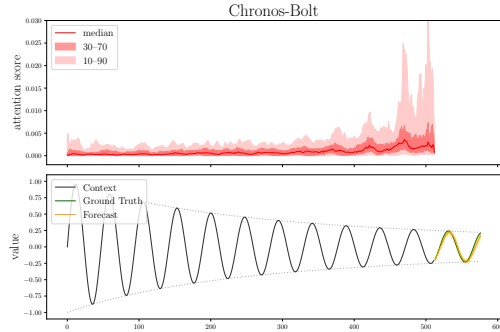

**Figure 27:** We show an example where Chronos and Chronos-Bolt perform similarly. In this example, the scale of context is monotonically decreasing, making there little to parrot. Both models show some capabilities for "reasoning" about the decreasing scale of the signal.

## D.2 Details of the Distance Bias

### D.2.1 Design of Experiments in Figure 6

Recall Figure 6, where we showed an experiment based on the real-world datasets that Chronos has a better capability of learning the fine-scale signals than Chronos-Bolt. We achieved this by augmenting Chronos' in-domain evaluation corpus, which contains 15 popular real-world univariate time series benchmark datasets. Given an input context $\mathbf{x} \in \mathbb{R}^L$ and a target $\mathbf{y} \in \mathbb{R}^T$, we split $\mathbf{x} = (\mathbf{x}_1, \mathbf{x}_2)$ evenly into two parts $\mathbf{x}_1$ and $\mathbf{x}_2$. Then, we fix a scale ratio $\alpha \geq 1$, we consider two regimes:

1. Predicting the large motif: we scale $\mathbf{x}_1$ to $\mathbf{x}_1/\alpha$ and form the multi-scale context as $\tilde{\mathbf{x}}_{\text{large}} = (\mathbf{x}_1/\alpha, \mathbf{x}_2)$. Let $\hat{\mathbf{y}} \in \mathbb{R}^T$ be a model's output, we compute the error by calculating the weighted quantile loss (WQL) using the ground truth $\mathbf{y}$ and forecast $\hat{\mathbf{y}}$.

2. Predicting the small motif: we scale $\mathbf{x}_2$ to $\mathbf{x}_2/\alpha$ and form the multi-scale context as $\tilde{\mathbf{x}}_{\text{small}} = (\mathbf{x}_1, \mathbf{x}_2/\alpha)$. Let $\hat{\mathbf{y}} \in \mathbb{R}^T$ be a model's output, we compute the error by calculating the WQL using the ground truth $\mathbf{y}$ and forecast $\alpha\hat{\mathbf{y}}$. Note that we need to scale the model's output by $\alpha$ to renormalize.

To compute the overall loss, we follow Chronos' evaluation strategy (Ansari et al., 2024a) by taking the geometric mean of the WQLs across all 15 datasets measured by comparing a model to a baseline, which is taken to be the model's performance when $\alpha = 1$, i.e., no augmentation is done. Note also that when $\alpha = 1$, predicting the large motif collapses to predicting the small motif.

### D.2.2 HOW DOES A TRANSFORMER PROCESS A MULTISCALE SIGNAL?

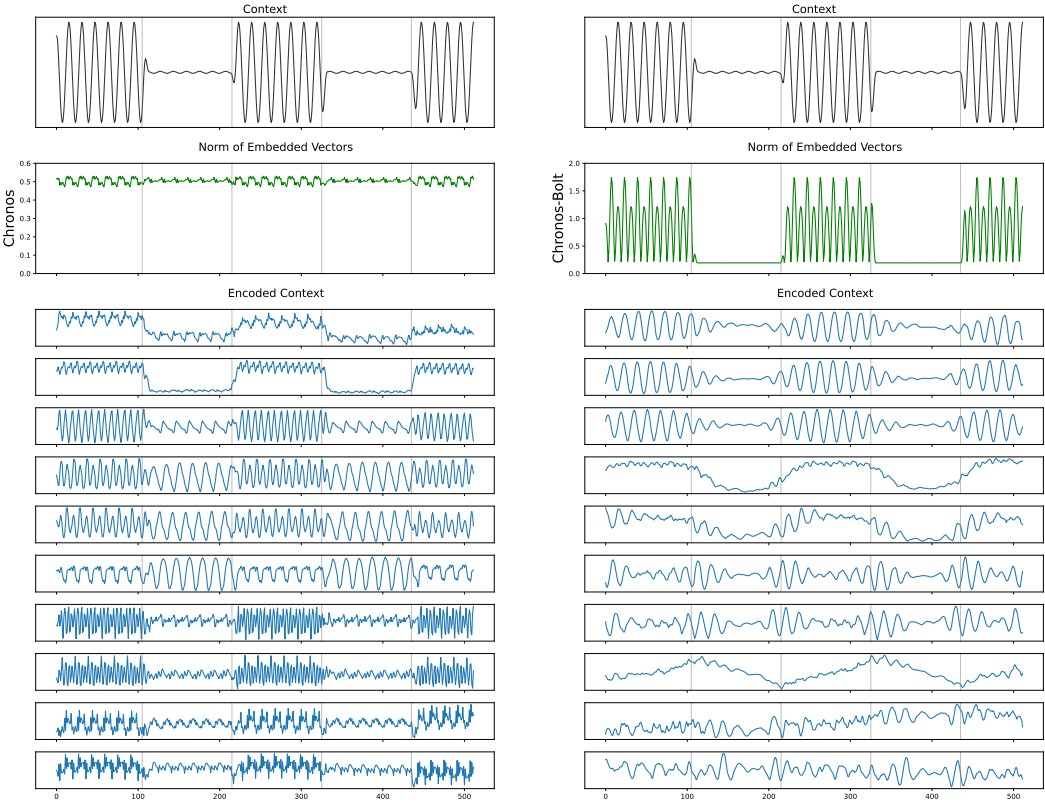

**Figure 28:** We show how each of Chronos and Chronos-Bolt (patch size $k = 1$) processes a multi-scale context, respectively. In the top panels, we show our input multi-scale context. The green curves on the second row of panels show the norm of the embedded vectors in each model, where we see that the fine-scale and the large-scale motifs both become large-scale in the embedded context, whereas the difference between the two scales stays distinguishable, if not more dramatic, in Chronos-Bolt's embedding. Then, we show the encoded context in both models, projected onto the top-10 principal components. We see that Chronos' fine-scale motifs got preserved nicely, while the fine scales are contaminated by the large scales in Chronos-Bolt.

To illustrate the distance bias of Chronos versus Chronos-Bolt better, we perform a detailed analysis given a multiscale context shown in Figure 28. First, given the context $\mathbf{x} \in \mathbb{R}^L$, we let the embedding be $\Phi(\mathbf{x}) = (\boldsymbol{\phi}_1, \ldots, \boldsymbol{\phi}_L) \in \mathbb{R}^{d \times L}$, where $d$ is the hidden dimension of the Transformer. In the panels on the second row of Figure 28, we show the norm $\|\boldsymbol{\phi}_j\|_2$ of these embedded vectors, which indicates a clear multiscale structure in Chronos-Bolt's embedding, but not as much in Chronos'

embedding. This is the root cause of the distance bias. To show that the embedding really matters, we further take the encoded context $\mathbf{z} \in \mathbb{R}^{d \times L}$, which is the sequence out of the encoder Transformer, and we plot $\mathbf{z}$ in its leading 10 principal components along the temporal direction. From Figure 28, we see that the fine-scale oscillation is preserved very well by Chronos' encoder, but it is severely contaminated by the large-scale oscillation in Chronos-Bolt's encoder.

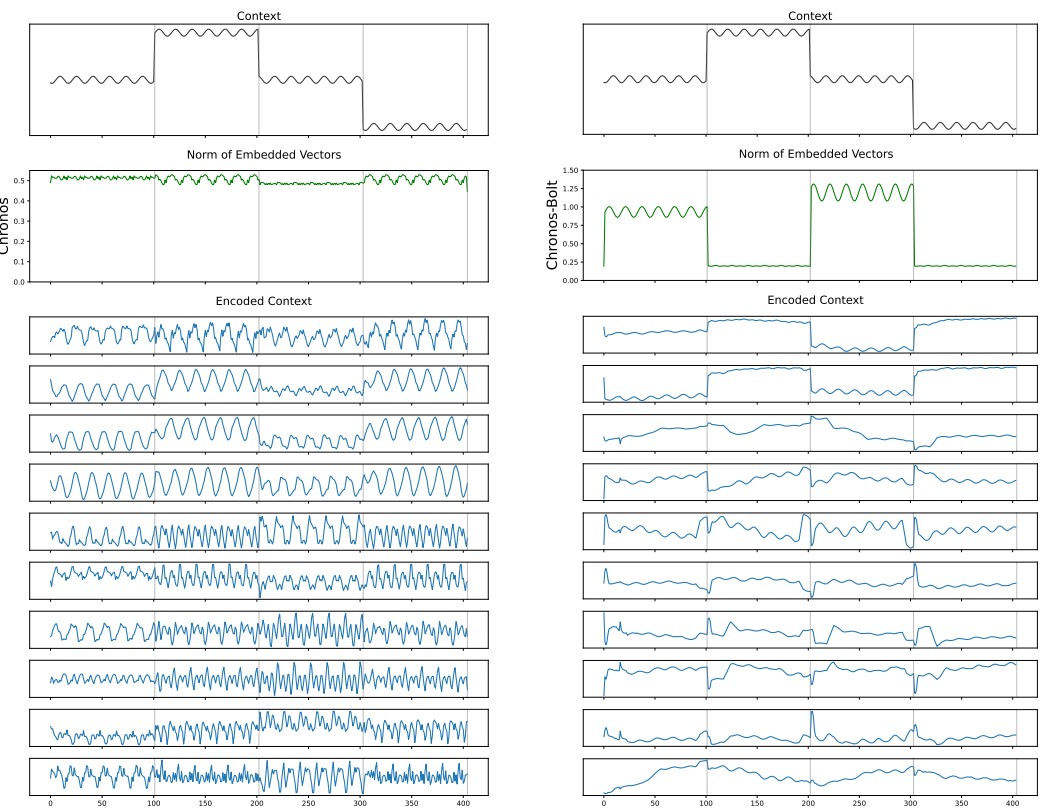

**Figure 29:** We show how each of Chronos and Chronos-Bolt (patch size $k = 1$) processes a multi-offset context, respectively. In the top panels, we show our input multi-offset context. The green curves on the second row of panels show the norm of the embedded vectors in each model, where we see that the high-offset and the low-offset motifs both have large norms in the embedded context, whereas in Chronos-Bolt's embedding, the low-offset motifs have much smaller norms than the high-offset ones. Then, we show the encoded context in both models, projected onto the top-10 principal components. We see that Chronos' low-offset motifs got preserved nicely, while they are contaminated by the high-offset motifs in Chronos-Bolt.

### D.3    DETAILS OF THE NORM BIAS

#### D.3.1    DESIGN OF EXPERIMENTS IN FIGURE 7

Recall Figure 7, where we showed an experiment based on the real-world datasets that Chronos has a better capability of learning the zero-offset signals than Chronos-Bolt. We achieved this by augmenting Chronos' in-domain evaluation corpus, which contains 15 popular real-world univariate time series benchmark datasets. Given an input context $\mathbf{x} \in \mathbb{R}^L$ and a target $\mathbf{y} \in \mathbb{R}^T$, we split $\mathbf{x} = (\mathbf{x}_1, \mathbf{x}_2, \mathbf{x}_3)$ evenly into three parts $\mathbf{x}_1$, $\mathbf{x}_2$, and $\mathbf{x}_3$. Then, fix a offset $\beta \geq 1$, we consider two regimes:

1. Predicting the high-offset motif: we drop $\mathbf{x}_2$ to $\mathbf{x}_2 - \beta$, lift $\mathbf{x}_3$ to $\mathbf{x}_3 + \beta$, and form the multi-scale context as $\tilde{\mathbf{x}}_{\text{high}} = (\mathbf{x}_1, \mathbf{x}_2 - \beta, \mathbf{x}_3 + \beta)$. Let $\hat{\mathbf{y}} \in \mathbb{R}^T$ be a model's output, we compute the error by calculating the WQL using the ground truth $\mathbf{y}$ and forecast $\hat{\mathbf{y}} - \beta$. Note that we need to drop the model's output back by $\beta$ to renormalize.

2. Predicting the low-offset motif: we drop $\mathbf{x}_1$ to $\mathbf{x}_1 - \beta$, lift $\mathbf{x}_2$ to $\mathbf{x}_2 + \beta$, and form the multi-scale context as $\tilde{\mathbf{x}}_{\text{low}} = (\mathbf{x}_1 - \beta, \mathbf{x}_2 + \beta, \mathbf{x}_3)$. Let $\hat{\mathbf{y}} \in \mathbb{R}^T$ be a model's output, we compute the error by calculating the WQL using the ground truth $\mathbf{y}$ and forecast $\hat{\mathbf{y}}$.

To compute the overall loss, we follow Chronos' evaluation strategy (Ansari et al., 2024a) by taking the geometric mean of the WQLs across all 15 datasets measured by comparing a model to a baseline, which is taken to be the model's performance when $\beta = 0$, i.e., no augmentation is done. Note also that when $\beta = 0$, predicting the large motif collapses to predicting the small motif.

### D.3.2 How Does a Transformer Process a Multi-Offset Signal?

To better illustrate the norm bias of Chronos versus Chronos-Bolt, we perform a detailed analysis given a multiscale context shown in Figure 29. First, given the context $\mathbf{x} \in \mathbb{R}^L$, we let the embedding be $\Phi(\mathbf{x}) = (\boldsymbol{\phi}_1, \ldots, \boldsymbol{\phi}_L) \in \mathbb{R}^{d \times L}$, where $d$ is the hidden dimension of the Transformer. In the panels on the second row of Figure 29, we show the norm $\|\boldsymbol{\phi}_j\|_2$ of these embedded vectors, where we see that Chronos' zero-offset motifs are embedded onto large vectors but Chronos-Bolt's are embedded onto small vectors. This is the root cause of the norm bias. To show that the embedding really matters, we further take the encoded context $\mathbf{z} \in \mathbb{R}^{d \times L}$, which is the sequence out of the encoder Transformer, and we plot $\mathbf{z}$ in its leading 10 principal components along the temporal direction. From Figure 29, we see that the zero-offset oscillation is preserved very well by Chronos' encoder, but it is severely contaminated by the large-offset oscillation in Chronos-Bolt's encoder.

### D.4 How Does the Vocabulary Size Relate to the Geometric Bias?

While our discussion of the geometric bias is mainly a comparison between the quantization-based embedding and the continuous embedding, if one quantizes the real line, the number of bins certainly also plays a role. Intuitively, if the number of bins is larger, then it is harder for the model to learn the geometry of the input domain in the hidden space, and consequently the geometry bias is stronger.

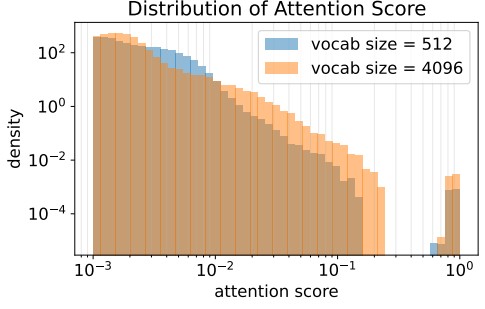 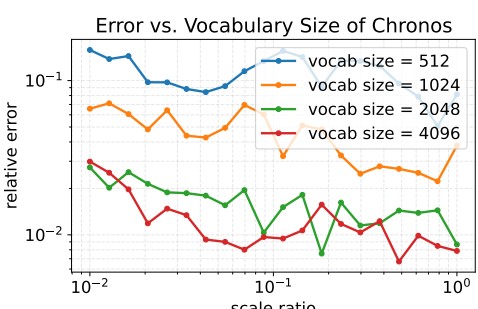

**Figure 30:** On the left, we show the distribution of the attention scores, using a Chronos model with a vocabulary size of 512 and one with a vocabulary size of 4096. On the right, we show the relative error of predicting the fine-scale motif of a multiscale structure, as outlined in Figure 6.

This intuition is corroborated by the two experiments shown in Figure 30. The histogram shows that a larger vocabulary size leads to more large attention scores, introducing a heavier bias for attention to local content. The error plot on the right shows that a larger vocabulary size generally magnifies the scale of a fine-scale motif in the hidden space, enabling better learning.

# E  DETAILS OF THE REGRESSION-TO-THE-MEAN BIAS (FROM SECTION 4)

In this section, we provide more details on the regression-to-the-mean bias. First, we use two examples to illustrate the loss landscape induced by the three standard loss functions: $L^1$-based, $L^2$-based, and cross-entropy (Appendix E.1). Then, we revisit the examples shown in Figure 8, and we show how Chronos learns the underlying probability distribution, instead of regressing to the mean (Appendix E.2). Finally, we give the details about how we build a "bridge" from the deterministic case to the random case in Figure 8 (Appendix E.3).

## E.1  A LOSS-LANDSCAPE ILLUSTRATION OF THREE KINDS OF LOSSES

To illustrate the concept of regression-to-the-mean/median, we consider the case where we have a ground-truth probability distribution on a simple measurable set $\Omega = \{0, 1/2, 1\}$ equipped with the discrete $\sigma$-algebra. Assume first that the ground-truth probability distribution is the one with

$$\mathbb{P}_{\text{truth}}(\{0\}) = \mathbb{P}_{\text{truth}}(\{1/2\}) = \mathbb{P}_{\text{truth}}(\{1\}) = \frac{1}{3}.$$

In each triangle in Figure 31, we show the barycentric coordinate corresponding to a model's output probability distribution on $\Omega$. The three different losses are shown using colors.

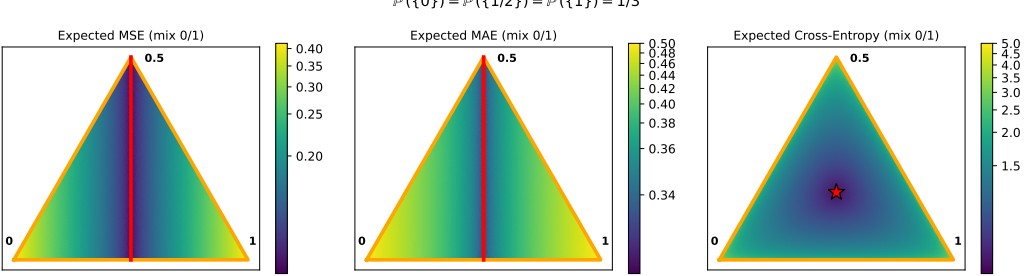

**Figure 31:** Assume we have a ground-truth probability distribution that equals $1/3$ on $0$, $1/2$, and $1$, respectively, and suppose our model generates a probability distribution on the set $\{0, 1/2, 1\}$. We show the loss landscape induced by the ground-truth probability and the model's forecast. The colors indicate the loss, and the barycentric coordinate in each triangle represents the model's predicted probability on $0$, $1/2$, and $1$, respectively. We use red colors to indicate the global minima in the loss landscape induced by each norm.

Note that the MSE and MAE are independent of the probability distribution: they only require a final prediction. In that sense, one can imagine that the entire triangle collapses to the line segment from $0$ to $1$ via orthogonal projection. That is essentially why our final forecast regresses to $1/2$, which is both the mean and median of $\mathbb{P}_{\text{truth}}$ in this example. For cross-entropy, the optimal is at the barycenter of the triangle, corresponding to exactly the ground-truth distribution.

Next, we consider a slightly different ground-truth probability distribution:

$$\mathbb{P}_{\text{truth}}(\{0\}) = \mathbb{P}_{\text{truth}}(\{1\}) = \frac{1}{2}, \qquad \mathbb{P}_{\text{truth}}(\{1/2\}) = 0.$$

From Figure 32, we see that MSE still regresses to the mean, which is the same as in the previous example; the cross-entropy loss now eliminates $1/2$ from its support, injecting more flexibility. The MAE loss, which tailors the median, gives us a completely flat loss landscape, where every distribution is an optimum.

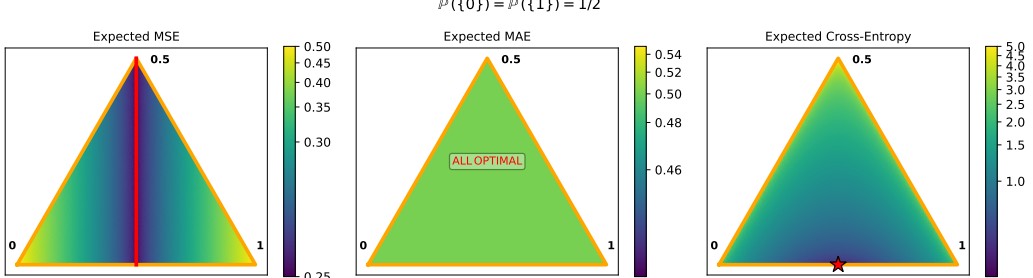

**Figure 32:** The plots show the same things as in Figure 31, except that we now set the ground-truth probability distribution to be a bi-modal one with a probability of $1/2$ on 0 and 1, respectively.

The only question that remains is: if all points are optimal against an MAE loss, then why is it that empirically, the model usually prefers one closer to the mean (see Figure 8)? One answer is: although every number between $[0, 1]$ is a median of this probability distribution, the medians have different stability. That is, if we compute the gradient of the MAE *with respect to* the probability $p$ at $p = 2$, assuming that

$$\mathbb{P}_{\text{truth}}(\{0\}) = p, \qquad \mathbb{P}_{\text{truth}}(\{1\}) = 1 - p, \qquad \mathbb{P}_{\text{truth}}(\{1/2\}) = 0,$$

then this gradient is large near the two endpoints 0 and 1 and small in the middle (see Figure 33). Why does that matter? In practice, even if the probability distribution is $1/2$ on both 0 and 1, when we sample the data, they will sometimes "lean" towards 0 and sometimes towards 1. Given that, "the median of the medians" gives us a more robust prediction.

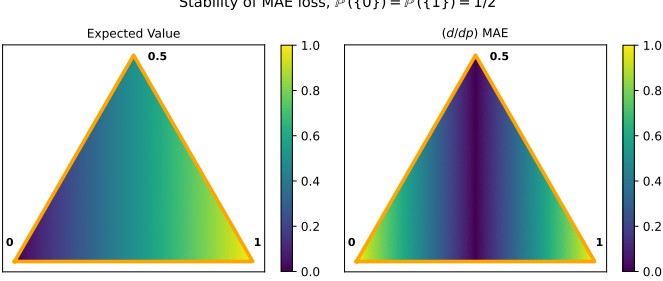

**Figure 33:** On the left, we show the expected value of the model's forecast in the barycentric coordinate (Schumaker & Yu, 2022). On the right, we show the stability of the MAE loss given the ground-truth in Figure 32, where the colors indicate the magnitude of the gradient $|(d/dp)\text{MAE}(\mathbb{P}_{\text{truth}}(p), \mathbb{P}_{\text{model}})|$ at $p = 1/2$, where $\mathbb{P}_{\text{truth}}(p)$ is $p$ at 0 and $1 - p$ at 1.

## E.2 HOW CONFIDENT CHRONOS IS AT BIFURCATION?

In Figure 8, we saw that when the context consists of only 0's and 1's, Chronos, in its forecast, only generates these two numbers. Since Chronos performs regression via classification, we can understand this better by looking into the logits of Chronos' forecast. Here, we look into these logits when Chronos forecasts the two contexts — one deterministic and one random — in Figure 8. Recall that the Chronos generates the output via binning: it preassumes a fine grid $\mathcal{B} = \{B_1, B_2, \ldots, B_V\}$ on the real-line and computes a probability distribution over $\mathcal{B}$. Here, when Chronos makes a forecast, we keep track of its generated probability associated with each $B_j$. In particular, there are two bins $B_{j_0}$ and $B_{j_1}$ whose values correspond to (the nearest grid point from) 0 and 1, respectively.

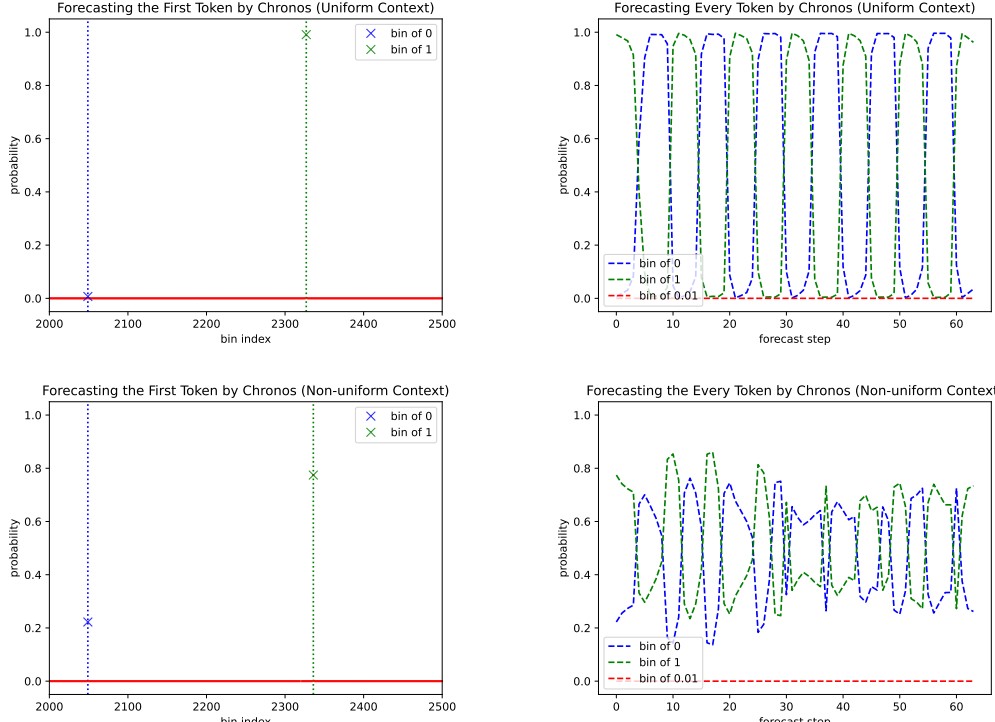

**Figure 34:** We compute the logits when we apply a Chronos model to predict the two synthetic signals shown in Figure 8. The top two panels show the results corresponding to the deterministic periodic walk between 0 and 1, while the bottom two panels show the results corresponding to a random walk. On the left, we show the logits associated with the first prediction. Each red dot corresponds to the logit associated with one bin, and the logits associated with the bin of "0" and the bin of "1" are highlighted with a blue and a green cross-mark, respectively. We see that the probabilities are dense on $\{0, 1\}$ and tiny elsewhere. On the right, we show the logits corresponding to the bins of "0" and "1", respectively, as we make more forecast steps. As a reference, we also show the logits corresponding to the bin of "0.01", which are nearly zero.

From Figure 34, we see that Chronos is almost always very certain in that its forecast is either 0 or 1. That is, the only two non-trivial probabilities that it produces are at 0 and 1 — all remaining probabilities are almost zero, even for the bins that are immediately next to 0 or 1. Incidentally, we should note that the fact that Chronos is capable of distinguishing 0 and 1 from their nearest bins is also due to the locality and distance bias (see section 3). The only major difference between forecasting the deterministic and the random contexts is that Chronos is much more confident in which one, between 0 and 1, to pick when the context is deterministic, whereas given a random context, Chronos generates a close probability on 0 and on 1. This is not surprising, as it aligns with our discussion that the cross-entropy loss tailors a model to learn the "ground-truth" probability distribution.

### E.3   How to Inject Uncertainty in Figure 8?

In Figure 8, we show an interesting quantification, where we start from a totally deterministic case, where we walk periodically between the two branches 0 and 1, and end up with the totally deterministic case, where for each time step, we randomly land at 0 or 1 with an even probability. Here, we show the details of how to gradually inject uncertainty and make a smooth transition from the deterministic case to the random case.

To this end, we first define the periodic context. Here, we alternately take $5$ steps of each branch. That is, our context is given by

$$x_t = \mathbb{1}_{[0,5)\cup[10,15)\cup[20,25)\cup\cdots}.$$

Given a probability of diffusion, we simply add a random "XOR" perturbation to this context with probability $1 - p$. That is, the new context is given by

$$\hat{x}_t = x_t \oplus \text{Bernoulli}(1 - p),$$

where the Bernoulli distribution at every $t$ is i.i.d.. It is clear that $\mathbb{P}(\hat{x}_t)$ is $p$ on one branch of $0$ and $1$ and $1 - p$ on the other, and that when we increase $p$ to $1/2$, our context becomes a random walk.

To understand how much the model regresses to $1/2$, the mean or the median of the entire sequence, we define the "regression score" by

$$\text{reg\_score}(y) = \min(|1 - y|, |y|).$$

This is shown in Figure 8, where our experiments are repeated $100$ times and the error bars show the $30\%$ to $70\%$ quantiles of all regression scores.

### E.4 THE EXPERIMENT IN FIGURE 8 IS ROBUST TO NOISES.

In Figure 8, we show an experiment with a 0-1 walk to show that a model trained with a cross-entropy loss does not always regress to the mean, unlike the other models trained with an $L^1$- or $L^2$-based loss. The only source of stochasticity, however, comes from the probability of landing on a $0$ or on a $1$. In this section, we show that the results hold seamlessly for noisy data. That is, we hold the context the same but inject random Gaussian noises to each timestamp.

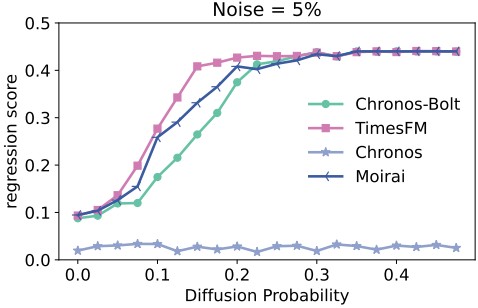 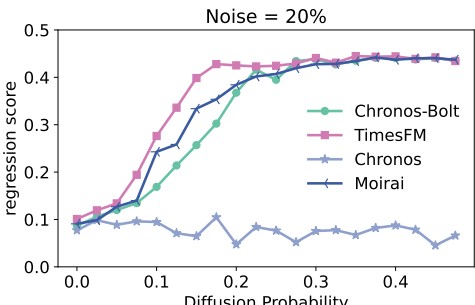

**Figure 35:** We inject random Gaussian noises to the experiments in Figure 8. We see that even with noises, a regression-based model still regresses to the mean, whereas the classification-based model does not.

From Figure 35, we note that with a substantial amount of noises, Chronos still does not regress to the mean, unlike other regression models, when the underlying probability distribution becomes bimodal.

# F THE OUTLIER HANDLING

In this appendix, we provide a detailed analysis of a particular case of outlier handling. We see that the context is a sinusoidal wave with many outliers injected. For this context, Chronos' embedding largely dampens the magnitudes of the outliers in the hidden space, as projected by the geometric bias. Consequently, the clean sinusoidal segment is preserved better during encoding. Also note, however, that Chronos tends to reproduce some of the outliers as it learns a probability distribution that hits both the clean sinusoidal structure and the outlier space; on contrary, Chronos-Bolt does not parrot a particular outlier.

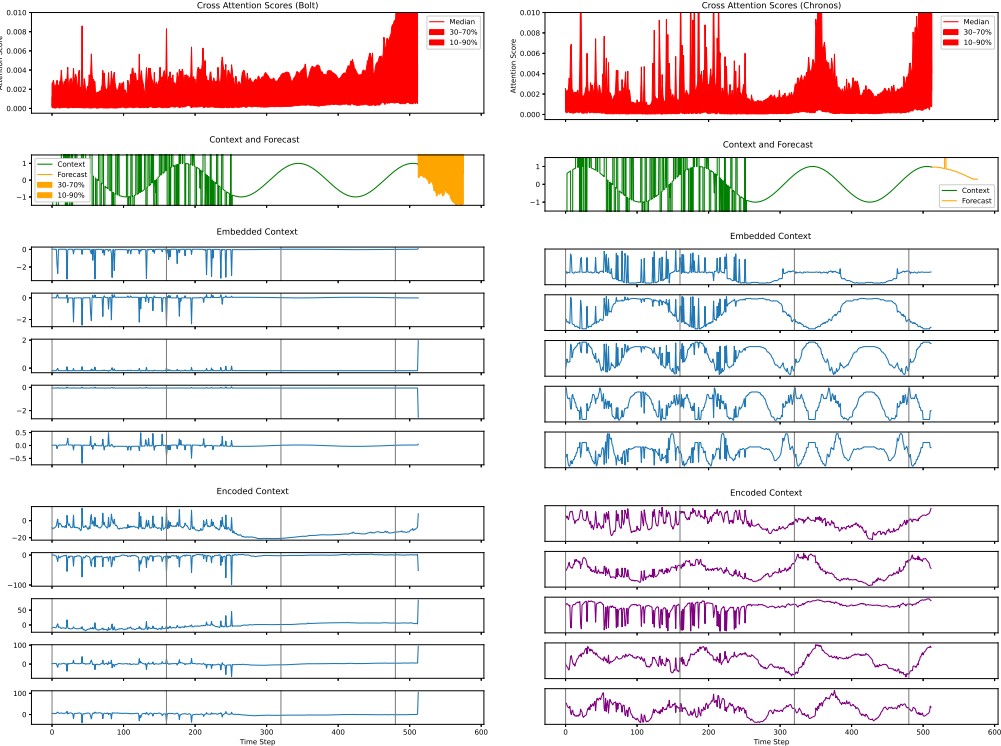

**Figure 36:** We show more detailed analysis of the outlier handling. All projections are onto the first 5 principal components of the matrix in the hidden space.

## G  THE SIMPLICITY BIAS

The term "simplicity bias" refers to the tendency of neural networks to favor simpler hypotheses, even when more complex, and potentially more robust, solutions are equally compatible with the data. This bias has roots in classical learning theory, which uses concepts like VC dimension and Rademacher complexity to explain how restricting effective hypothesis complexity enables generalization (Wolpert & Macready, 2002). Empirically and theoretically, the simplicity bias has been observed in both shallow architectures, e.g., two-layer networks tending to depend on only a low-dimensional projection of the input (Morwani et al., 2023), and deep transformers, which typically learn low-order interactions first before higher-order ones (Belrose et al., 2024; Rende et al., 2024). Some work (Goldblum et al., 2024) argues that deep models inherently favor lower-order polynomial fits when possible (Wilson, 2025), while other studies explore how parameters such as effective rank further constrain learned representations toward simpler, low-rank subspaces (Huh et al., 2021). Together, these findings underscore how a simplicity bias, though still not fully explained, provides a key inductive advantage in TSFMs, helping them generalize while highlighting the trade-offs between robustness and expressivity. In this section, we provide some discussion of this simplicity bias.

The first thing to note is that, unlike the other biases discussed in this paper (where there is typically a trade-off — tuning a design decision "knob" improves performance on one end but sacrifices it on the other, in ways that depend on the specific model), the simplicity bias is different. First, it is more of a "design goal" than a "bias" (at least in the sense we have used the word so far). Second, and relatedly, recent work has revealed that larger neural networks often exhibit this simplicity bias, which favors simpler solutions despite their expressive power. While the precise mechanism behind this phenomenon remains an active area of research, it is an important consideration for TSFMs. In particular, from an expressiveness perspective, even small models are already highly capable, but larger models seem to introduce an additional simplicity bias that helps prevent overfitting. To maintain a simplicity bias via a larger model, however, we also pay significant costs in compute and memory. A promising direction, perhaps informed by our insights in this paper, is to retain this bias while reducing overhead through post-training compression techniques.

### G.1  ILLUSTRATION OF THE SIMPLICITY BIAS

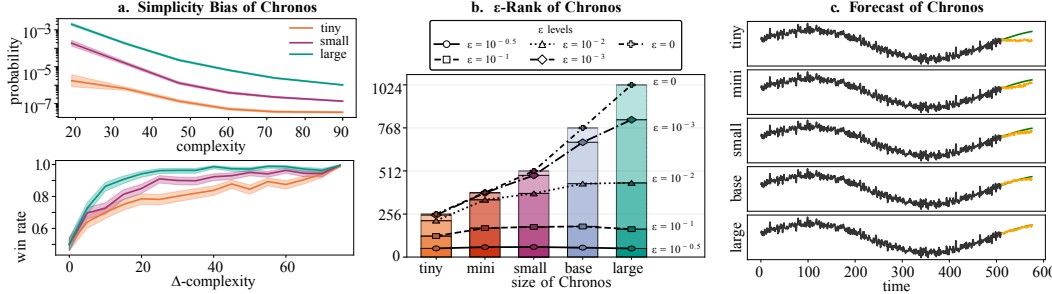

**Figure 37:** The simplicity bias concerns how a model prefers simple solutions over complicated ones, enhancing the generalization capability (see the right column). In the left column, we present two targeted experiments: first, we construct time series using a different number of basis elements, which we call the complexity, and we compute the model's probability of forecasting the correct future; next, we fix a context and evaluate chances that a model computes a higher probability for a simpler solution than a more complex one, which we call the "win rate." The larger model consistently prefers simpler solutions more than the smaller ones. The middle panel shows the compressibility of the attention matrices in pretrained Chronos models, indicating that larger attention matrices mainly contribute to over-parameterization rather than expressiveness (see also Yu et al. (2025b)).

In the experiment shown in Figure 37, we design a context $\mathbf{x}_{1:L}$ with two continuations: a simple one $\mathbf{y}^{(s)}_{L:(L+T)}$; and a complex one $\mathbf{y}^{(c)}_{L:(L+T)}$. We compare the rate that a model prefers the simple

solution over the complex one: $\mathbb{P}_{\text{model}}\left(\mathbf{y}^{(s)}_{L:(L+T)}\middle|\mathbf{x}_{1:L}\right) > \mathbb{P}_{\text{model}}\left(\mathbf{y}^{(c)}_{L:(L+T)}\middle|\mathbf{x}_{1:L}\right)$. The finesse of this experiment design is that when the $\Delta$-complexity between the two continuations is zero, then any model prefers them equally; also, when the $\Delta$-complexity gets large, eventually any model simply always prefers the simple solution. Walking from one extreme case to the other, in Figure 37, we see that Chronos (large) consistently prefers simple solutions more than the smaller models.

> **Summary of the Simplicity Bias:** The *simplicity bias* refers to the tendency of TSFMs to favor simpler solutions, which helps explain their robustness despite their large flexibility. This bias is amplified by model size, as scaling up introduces stronger preferences toward simpler hypotheses. It is also closely tied to compressibility, suggesting that models can retain a simplicity bias while still being pruned or compressed for efficient inference.

We now provide details about the two experiments shown in the first column of Figure 37.

### G.2 PREDICTION PROBABILITY AS A FUNCTION OF DATA COMPLEXITY

First, we discuss the details of the first panel in the first column of Figure 37.

**Data generation.** Each synthetic series is split into a context of length $L$ and a future of length $T$, $\mathbf{y} = (y_1, \ldots, y_L, y_{L+1}, \ldots, y_{L+T})$. The context is produced by a simple parametric mechanism that the model can plausibly capture with $L$ samples. We use two base families: a linear trend $y_t = at + b$; and a single sinusoid $y_t = A_0 \sin(2\pi f_0 t + \phi_0)$. The future is obtained by continuing the same base mechanism and then optionally adding $M \geq 0$ sinusoidal components,

$$y_{L+t} = y^{\text{base}}_{L+t} + \sum_{m=1}^{M} A_m \sin(2\pi f_m t + \phi_m), \qquad t = 1, \ldots, T,$$

with a short ramp-in at the beginning of the horizon to avoid discontinuities. To control the intrinsic complexity of a sample in a way that is measurable and reproducible, we use a bit-budget proxy $K(\mathbf{y})$ that counts the number of bits required to specify the added components. For each new component, we allocate $k_f$ bits to choose a frequency from a coarse-to-fine grid, $b_A$ bits to quantize amplitude in $[A_{\min}, A_{\max}]$, and $b_\phi$ bits to quantize phase in $[0, 2\pi)$; hence $K(\mathbf{y}) \approx K_{\text{base}} + \sum_{m=1}^{M}(k_f + b_A + b_\phi)$. Small nonzero complexities are achievable by allowing very coarse choices (e.g., $k_f = 0, 1, 2$) and low-bit quantization for $(A, \phi)$; larger complexities are obtained by increasing these bit budgets and/or $M$. After generation, we standardize the future with a shared scaling computed from the simple continuation (mean $\mu_S$, std $\sigma_S$) so that the magnitude of log-likelihoods is not dominated by trivial scale differences: $\tilde{y}_{L+t} = (y_{L+t} - \mu_S)/\sigma_S$.

**Result computation.** For a model $\mathcal{M}$, we score each future under teacher forcing, using the Chronos tokenizer to encode the context as input IDs and the future as labels. The model's total log-likelihood of the future given the context is

$$\log p_{\mathcal{M}}(\mathbf{y}_{L+1:L+T} \mid \mathbf{y}_{1:L}) = \sum_{t=1}^{T} \log p_{\mathcal{M}}\big(y_{L+t} \mid \mathbf{y}_{1:L}, y_{L+1:L+T-1}\big),$$

and we normalize by $T$ to obtain an average log-probability per token $s(\mathbf{y}) = \frac{1}{T}\log p_{\mathcal{M}}(\cdot)$. To summarize how likelihood changes with intrinsic complexity, we bin the samples by $K(\mathbf{y})$ into $B$ evenly populated[2] bins and compute, for each bin $b$, the sample mean $\bar{s}_b$ and a 95% confidence interval using the normal approximation $\bar{s}_b \pm 1.96\,\hat{\sigma}_b/\sqrt{n_b}$, where $\hat{\sigma}_b$ is the sample standard deviation and $n_b$ the number of sequences in the bin. The baseline plot overlays the bin means for all models as a function of the corresponding bin centers in bits. For downstream plotting, we also save a compact $3 \times B$ array per model containing the lower bound, mean, and upper bound in each bin, along with the vector of bin centers. Interpreting the figure is straightforward: curves that lie higher indicate that the model assigns more probability mass on average to sequences at that complexity level; a gentle downward slope with increasing bits is expected, as additional structured variation makes the future harder to predict.

---

[2] In practice, we use a helper that assigns indices by quantiles of $K$; we report the mean of $K$ within each bin as its horizontal location.

### G.3 WIN RATE OF THE SIMPLE SOLUTION

Next, we discuss the details of the second panel in the first column of Figure 37.

**Data generation.** This Occam-pairs experiment asks whether a model prefers a simpler continuation over a more complex one, when both are plausible given the same past. We first draw a context $\mathbf{y}_{1:L}$ from the same base families as above (linear or single sinusoid) and form a simple future by continuing the base mechanism into the next $T$ steps. We then create a complex future by adding controlled extra structure to the simple one:

$$\mathbf{y}_t^{(C)} = \mathbf{y}_t^{(S)} + \sum_{m=1}^{M} A_m \sin(2\pi f_m t + \phi_m), \qquad t = 1, \ldots, T,$$

with the same ramp-in convention and the same shared standardization with respect to the simple future. The key scalar we vary across trials is the complexity gap $\Delta K$, defined as the additional bit budget needed to describe the extra components. As above, each component pays $k_f + b_A + b_\phi$ bits; thus $\Delta K = \sum_{m=1}^{M}(k_f + b_A + b_\phi)$ (optionally with a combinatorial term if we explicitly encode a subset of a large frequency grid). To ensure well-behaved behavior at the endpoints, we include two calibrations. At $\Delta K = 0$ we set $M = 0$ and, if observational noise is added, we add the same noise realization to both futures. This makes the two futures exchangeable, so no model can reliably prefer one over the other. At very large $\Delta K$ we gradually turn off preview leakage from the context and increase the amplitude gain of the extra components over a window in $\Delta K$, so that the complex future departs cleanly and strongly from the pattern established in the context.

To prevent the win-rate curves from trivially saturating near 1 (as they would if the complex future were always obviously inconsistent with the context), we apply a lightweight scheduler that couples nuisance knobs to the target gap $\Delta K$. As $\Delta K$ increases, we gradually reduce observation noise and ramp-in length, and we suppress any preview of the complex components in the context. At small and mid $\Delta K$ these choices keep the two futures similarly plausible from the same past, so wins are earned rather than guaranteed. We also regularize the construction with shared standardization (both futures are scaled by the simple future's mean and variance), matched energy of the added components across $\Delta K$ (so higher bit budgets translate to finer specification rather than unbounded amplitude), and an explicit tie tolerance $\varepsilon$ so indistinguishable pairs contribute $0.5$ on average. Two anchors complete the calibration: at $\Delta K = 0$ we set $M = 0$ and share the same noise realization across the two futures; while at very large $\Delta K$ a soft right-end "anchor" smoothly turns off preview and ramp and slightly boosts gain so that all models approach 1 without a discontinuous jump. Together, the scheduler and regularization make the mid-range of $\Delta K$ informative, avoiding the confusion that would arise if readers compared to the baseline plot (where single-sequence likelihoods often decline quickly with complexity) and wondered why pairwise win rates would otherwise sit near 1 for almost all $\Delta K$.

**Result computation.** For each triple $(\mathbf{y}_{1:L}, \mathbf{y}_{1:T}^{(S)}, \mathbf{y}_{1:T}^{(C)})$ and model $\mathcal{M}$ we compute two teacher-forced log-likelihoods,

$$\ell_S = \log p_{\mathcal{M}}\big(\mathbf{y}^{(S)} \mid \mathbf{y}_{1:L}\big), \qquad \ell_C = \log p_{\mathcal{M}}\big(\mathbf{y}^{(C)} \mid \mathbf{y}_{1:L}\big),$$

and form the difference $\Delta\ell = \ell_S - \ell_C$. A trial counts as a *win for simple* if $\Delta\ell$ is positive by more than a small tie tolerance $\varepsilon$; if $|\Delta\ell| \leq \varepsilon$ we record a half-win so that truly indistinguishable pairs contribute $0.5$ on average. Repeating this for many independently generated pairs at the same $\Delta K$ yields an empirical win rate $W(\Delta K) = \frac{1}{n}\sum_i w_i \in [0, 1]$. We summarize uncertainty with a Wilson score interval at $95\%$ confidence, which is appropriate for a proportion and behaves well even when $W(\Delta K)$ is near 0 or 1. Plotting $W(\Delta K)$ against the bit gap produces the Occam curve for each model. Two anchors validate the construction: at $\Delta K = 0$ all models sit near $0.5$ by symmetry; at sufficiently large $\Delta K$ all models approach 1 because the complex future contains clear superfluous structure absent from the past. The region between these anchors is informative: models whose curves rise earlier and remain higher over a broad mid-range of $\Delta K$ are better at recognizing and penalizing unnecessary structure, thus exhibiting a stronger empirical simplicity bias in this controlled setting.

## H    THE USE OF LARGE LANGUAGE MODELS (LLMs)

LLMs are used for polishing the writing and word choices of a few sections in the main text. They are not used in the conceptualization and implementation of research.

