# OpenReview forum: "Understanding the Implicit Biases of Design Choices for Time Series Foundation Models"
_ICLR.cc/2026/Conference — ICLR 2026 Poster_

### Official Review · Reviewer_UNdi · 2025-10-29

**Soundness:** 2
**Presentation:** 3
**Contribution:** 2
**Rating:** 4
**Confidence:** 4

**Summary:**

This paper did a investigation on implicit biases in Time Series Foundation Models (TSFMs). These implicit biases are introduced by several design knobs such as patch size, embedding type, and training loss. Instead of proposing a new model, this paper systematically analyzes how these design choices the model’s internal representations and generalization behavior. In this paper, it mainly identifies and discuss three main biases: temporal bias, geometric bias and regression to the mean bias. Combined theoretical analysis with empirical validation, it explains and shows how each biases cause issues and how to improve them.

**Strengths:**

1: This paper shows a wealth of visualization, including different type of figures. These figures are very easy to understand.

2: The motivation of this paper is very clear. It shifts the focus from performance chasing to understanding inductive biases, highly relevant for Time Series Foundation Models (TSFMs)'s  robustness and transferability.

3: This paper did a deep analysis on each implicit bias, validating findings across several leading Time Series Foundation Models (TSFMs). It also shows practical insights on each bias's investigation.

**Weaknesses:**

1: For theorem 1, it seems that its theoretical scope is a little bit narrow and relies on simplified assumptions. In Theorem 1, it assumes a linearized ReLU MLP and random Gaussian projections to derive the property of patch embeddings. However, in modern Time Series Foundation Models (TSFMs), they have some attention and temporal positional encoding to dominate their feature engineering. We didn't see any assumptions or discussions on these. We are not sure if this paper's conclusions can hold for transformers trained end-to-end.

2: When this paper discussed about low-frequency preference to large patch size, it didn't provide enough quantitative analysis of how this emerges during training. For example, visualization of any frequency-domain activations through epochs may support this claim. More explanations and rigorous proofs are needed to clarify these points.

3: For its geometric bias, its geometry discussion shows that angle between embedded vectors for discrete embeddings is much larger than that for continuous embeddings. Also, this grows with the number of bins. However, this paper didn't measure vocabulary size systematically over this observation shown in its figure. More ablation study should be discussed and provided to show that whether the reported geometry is an inherent property of discretization or an artifact of one particular size.

**Questions:**

Most of questions are mentioned in weaknesses.

1: Could you please give more evidence to support how large patches can lead to low-frequency alignment? especially any end-to-end training support?

2: Could you include structural evaluation metrics to show your conclusions for model's structural correctness?

3: Have you tested your conclusions or findings on real industrial datasets with missing values, multi-seasonality, and domain shifts? How will your findings help the improvements?

---

> ### Author Response · Authors · 2025-11-18
>
> We thank the reviewer for the careful review and insightful comments.
>
> * **W1 + Q1. Practicality of Theorem 1**
>
>   We use Theorem 1 to mainly convey the message that *many* ReLU-activated NN embeddings send the low- and high-frequency patches into nearly orthogonal spaces, which are easy to encounter during training. We agree that the random Gaussian assumption does not apply to trained weight matrices and have added a discussion after Theorem 1 to acknowledge that.
>
>   To showcase the practicality of Theorem 1 in a real TSFM trained end-to-end, we added an experiment in Appendix C.1.4 to show this orthogonality during training. We see that the low- and high-frequency patches are indeed embedded into two orthogonal subspaces at any stage of pretraining.
>
> * **W2. Evolution of the Frequency Bias**
>
>   It is an interesting question how the frequency bias evolves during training. To show this, we added an experiment in Figure 18, which illustrates how the attention weight kernel $\mathbf{W}_Q^\top \mathbf{W}_K$ acts on the low- and high-frequency patch spaces. We find that initially, the kernel $\mathbf{W}_Q^\top \mathbf{W}_K$'s projections onto the low- and high-frequency spaces are roughly equal. During training, the kernel gradually aligns with the low-frequency space than the high-frequency one. We hope this additional experiment helps to better support our discussion of the frequency bias.
>
> * **W3. Vocabulary Size and the Geometric Bias**
>
>   The vocabulary size plays a role in the geometric bias. We thank the reviewer for pointing this out. To demonstrate the effect of the vocabulary size, we have added two additional experiments to the revised paper. (See Appendix D.4.) To summarize, we find:
>
>   * **Anglular (locality) bias:** By plotting the histograms of the attention scores, we show that a larger vocabulary size leads to larger attention scores. This introduces a heavier bias for attention to local content.
>
>   * **Distance (scale) bias:** We also show how models of different vocabulary sizes handle fine-scale motifs given a multiscale context. We show that a larger vocabulary size generally magnifies the scale of a fine-scale motif in the hidden space, which enables the model to achieve a lower error.
>
> * **Q2. Structural Evaluation Metric**
>
>   As each bias controls a trade-off, we do not suggest that a specific design is "correct." In our paper, we use several probes that can be interpreted as structural evaluation metrics: (1) the angles and distance between low- and high-frequency patch embeddings, (2) the norm of the embedded vectors, and (3) the decomposition of the attention kernel $\mathbf{W}_Q^\top \mathbf{W}_K$ into components aligned with the low- and high-frequency subspaces over pretraining (Figure 18). These quantities are task-agnostic and are specifically designed to test whether the trained TSFMs exhibit the structural behaviors predicted by our theory.
>
> * **Q3. Experiments on Real-world Datasets**
>
>   We did provide experiments on real-world data in the bottom-right tables of Figures 6 - 7 of the revised paper (Figures 5 - 6 of the submitted version). In these experiments, we augment the 27 real-world datasets used in Chronos' zero-shot evaluation that contain more noise and missing values. (See Appendix D.2.1 and D.3.1 for more information.) In particular, we showed that the same findings we identified on our synthetic data also hold on the real-world data, i.e., as the scale ratio and offset gap are increased, the performance of the continuous embedding-based models, e.g., Chronos-Bolt, TimesFM, and Moirai, worsens, respectively. The results indicate that the geometric biases apply generally to several TSFMs, and are observed on both synthetic and real-world data.
>
>   To show further evidence on noisy datasets, we also included additional experiments that inject noise into the regress-to-the-mean test dataset. (See Appendix E.4 for more details.) Our results show that the regression-to-the-mean bias holds seamlessly for noise-free and noisy data.
>
> We are happy to answer any follow-up question(s) that the reviewer may have.

---

> > ### Author Response · Authors · 2025-11-27
> >
> > Dear Reviewer UNdi,
> >
> > We sincerely appreciate your time and thoughtful evaluation of our work. Your feedback was very helpful and directly contributed to improving the manuscript, as we outlined in the rebuttal. We hope our clarifications and additional experiments have addressed your concerns, and we would greatly appreciate it if you could consider updating your evaluation. Please feel free to let us know if any questions remain. Thank you again for your time and support.
> >
> > Best regards,
> > Authors

---

> ### Comment · Reviewer_UNdi · 2025-11-28
>
> Thanks for your detailed response. After reading the revised paper, I think my main concerns have been addressed, especially regarding the practicality of Theorem1 and the discussion of vocabulary size. These clarifications and additional experiments strengthen the paper. I will consider increasing my score a little bit. Thanks a lot.

---

### Official Review · Reviewer_8DhK · 2025-10-30

**Soundness:** 3
**Presentation:** 2
**Contribution:** 4
**Rating:** 6
**Confidence:** 3

**Summary:**

The paper presents a study of implicit biases that follow from the design choices of time-series foundation models. Five biases are categorized into two categories, namely temporal (2) and geometry (3) biases. The biases are subtle but are revealed through experiments and proofs (given in the appendix).

**Strengths:**

* The topic and analysis are novel.
* I expect this paper to be highly significant for time-series modelling relying on deep learning in general and not only for research in time-series foundation models.
* The paper is extremely thorough.
* The structure of the paper is nice.
* I believe the conclusion to be stronger than what is stated in the paper (see question section).

**Weaknesses:**

While the writing generally is good, this paper has some issues that I would like to point out.
* The different biases are not defined explicitly in text. For example, what is meant by temporal and geometric bases? Please specify.  What is the frequency bias and the periodicity bias? What are the angels, distances and norms biases? Explicitly state this, preferably right after the bold heading for each term.
* For temporal bias, frequency, periodicity and seasonality are mentioned, but while frequency and periodicity have their own bias, seasonality is not mentioned again explicitly. What is the difference between seasonality and periodicity?
* On line 149: “In general, these models show different inductive biases …” Which? Be explicit and give examples.
* The term frequence seems to be overloaded and sometimes mean frequency as measured by Hz and at other times mean “Fourier modes”. When it means one thing and when it means another thing is not explicit. Thie reduces the readability. I would suggest to not use frequency to indicate “Fourier modes” and use a different term instead?
* For geometry bias, paragraph starting on line 267, it is not clear what geometry bias means, as mentioned above, nor which biases are grouped under this term. This is stated for temporal biases.
* The term “breaking the geometry” is used on line 280, but as it is not clear what is meant by geometry, the term is not easily understood.
* Line 314: Please explain what is meant by more complex reasoning.
* Not clear what is meant by “the geometry of the input domain”.
* Section 5 Mixture of Biases is a bit short. It would improve with more thorough explanations.

In summary, this paper is not very easy to read. However, this should be fairly easy to correct given that more space is available for final submission (and in the rebuttal phase).

Given the depth of the analysis and the complexity of the topic, it would probably improve by having less restrictions on page numbers. Publishing the paper in a journal could be an alternative to a conference because of the share amount of work that is reported. I realize that few journals have the reputation of ICLR and that publication might take longer.

There are issues with the PDF. It is heavy. The problem is especially noteable on page 7. I assume this is because of Figures 5 and 7. Is it because the figures contain text? Is the text the cause of the problem? They seem like vector graphics and not high-resolution images when I zoom in. Still, the page loads terribly slowly. This issue should be rectified if accepted.

Improving the text will not only improve the presentation score, but also the soundness, as it will be easier to evaluate it with better presentation.

**Questions:**

On line 346, what seems like a very important insight is mentioned, but only implicitly. I read it as: Because of the findings in this paper, foundation models for time-series might not be possible because the different design decisions will affect the performance on different tasks. Therefore, TSFMs are task specific, which is not the case for language models as they all operate on the same type of representation (language) with similar characteristics, as words and sentences. Is this the conclusion, or am I misunderstanding something? If this is the case, then the argument should probably be made explicitly in the conclusion.

Does the summary of results in Figure 1 that is described in text from line 93 reflect the summaries given in the boxes after the description of the biases? For example, Patch Size $\rightarrow$ Temporal bias. However, patch size is a subset of temporal bias, but it also contains architectural choices and unmasked [REG] tokens as well? Did I understand this correctly?

---

> ### Author Response · Authors · 2025-11-18
> **Official Comment by Authors (1/2)**
>
> We thank the reviewer for the careful review and insightful comments.
>
> * **Definition of Inductive Biases**
>
>   We have added the definitions of our identified temporal, geometric, and regression-to-the-mean biases after their section headers in the revised version. We also explicitly defined what frequency/periodicity and angle/distance/norm mean in each corresponding paragraph.
>
> * **Seasonality vs. Periodicity**
>
>   We used the word periodicity for patterns that repeat, but not necessarily at a fixed frequency. By seasonality, we look for a similar pattern that repeats at roughly the same frequency. (See Footnote 1.) We have made this distinction clearer in Section 2.
>
> * **Clarifications of the Frequency Bias**
>
>   We have revised the sentence that starts with "In general, these models show different inductive biases" into "Most of these models have been shown, both theoretically and empirically, to be better at learning the low frequencies than high frequencies." This claim is supported by the list of references in the prior sentence.
>
> * **Usage of the Word "Frequency"**
>
>   You are correct that our use of the word "frequency" was overloaded. For the name of the bias, we keep it to be "frequency bias" since this is a standard terminology in the literature. To improve clarity, when referring to the basis functions of the discrete Fourier transform, we now use the term "spectral modes", and we reserve "frequency" only for its conventional meaning in the time-domain sense. We believe this resolves the ambiguity and significantly improves readability.
>
> * **Sub-biases Under the Geometric Bias**
>
>   We have added a definition of the geometric bias in Section 3. We also explicitly mentioned that the geometric bias involves the three sub-biases, and for each of them included a definition.
>
>
> * **Meaning of "Breaking the Geometry"**
>
>   We have explicitly included a discussion of the notions of "geometry" that we consider in this section. We also changed the term "breaking the geometry" into "not preserving the geometry" to enhance the interpretability.
>
> * **Meaning of "More Complex Reasoning"**
>
>   By "complex reasoning," we refer to forecasting tasks where the model must combine information from multiple, potentially distant parts of the input sequence/time series, detect interactions between trends and seasonalities, or integrate multi-scale patterns. Models with highly local attention (like Chronos) excel at short-range copying but can struggle in these settings. We revised the text to clarify this meaning.
>
> * **Meaning of "Geometry of the Input Domain"**
>
>   We have changed the phrase "geometry of input domain" into "geometric properties of input domain" and highlighted its meaning in the opening sentence of Section 3, which is the geometry induced by the standard inner product $\langle \cdot, \cdot \rangle$ on the input space as a Hilbert space. We also define the three notions (angle, distance, and norm) that we consider in the paper.
>
> * **Discussion of the Mixture of Biases**
>
>   We agree that the discussion of the mixture of biases was overly brief due to the space constraint. Given the additional page during the rebuttal period, we have expanded our discussion of the outlier handling case study with more details of how different biases are mixed together.

---

> ### Author Response · Authors · 2025-11-18
> **Official Comment by Authors (2/2)**
>
> * **Q: Infeasibility of TSFMs**
>
>   Thank you for raising this point. We agree that this is an important and subtle implication of our work.
>
>   * Our results suggest that it is not straightforward to design a single TSFM whose inductive biases behave uniformly well across all forecasting settings. Our findings highlight that time-series forecasting is inherently task- and data-regime-dependent, much more so than language modeling, where input structure is more homogeneous. This makes the idea of a single universally-optimal architecture still an open research problem.
>
>   * Given this, we believe a promising direction is not to search for a one-size-fits-all TSFM, but instead to explore mechanisms that allow models to adapt their biases to the context, e.g., in-context learning, retrieval-augmented modeling, or reinforcement-learning-based adaptation. These approaches may allow TSFMs to dynamically adjust which inductive biases are activated for a given input regime, rather than baking in a single static set of biases. We have revised the conclusion to make this perspective explicit as directions for future work.
>
> * **Q: Summary in Figure 1**
>
>   You are correct that Figure 1 only illustrates a subset of the biases we studied in the paper. As it would be difficult to visually incorporate an illustration of every bias that we studied in the paper, we updated the caption of Figure 1 to highlight that we demonstrated only "examples of the three classes of inductive biases."
>
> * **Conference-style and PDF Size**
>
>   With the additional page in the revision, we were able to expand on the definitions and clarifications of the biases, related work, and discussion of the outlier bias.  We also appreciate the note about the PDF size and slow loading. Upon acceptance, we will re-render the figures (e.g., by rasterizing text-heavy vector components or reducing unnecessary detail) to ensure fast loading and smooth viewing.
>
> We are happy to answer any follow-up question(s) that the reviewer may have.

---

> > ### Author Response · Authors · 2025-11-27
> >
> > Dear Reviewer 8DhK,
> >
> > We sincerely appreciate your time and thoughtful evaluation of our work. Your feedback was very helpful and directly contributed to improving the manuscript, as we outlined in the rebuttal. We hope our clarifications and additional experiments have addressed your concerns. Please feel free to let us know if any questions remain. Thank you again for your time and support.
> >
> > Best regards,
> > Authors

---

### Official Review · Reviewer_cdYf · 2025-10-30

**Soundness:** 3
**Presentation:** 2
**Contribution:** 3
**Rating:** 6
**Confidence:** 4

**Summary:**

The paper discusses how different architectural choices in modern time series foundation models lead to learning biases in the resulting models. In particular, it focuses on biases in modeling the dynamics of input time series and in the structure of the predictions. Each observation is supported by convincing empirical results and well-designed experiments. I found the paper interesting and believe it is a relevant contribution to our understanding of the properties of current time series forecasting architectures. Although there are some presentation issues, I believe the paper is a solid contribution to the conference.

**Strengths:**

* The paper studies interesting and important phenomena in time series forecasting models.
* The findings are particularly relevant in the context of foundation models.
* I appreciate the focus on clarifying the impact of design choices that are often overlooked.
* The empirical analysis is interesting and well-designed.

**Weaknesses:**

### Main weaknesses

The writing and presentation could be improved, and several claims would benefit from additional discussion and supporting evidence.

- The introduction could do a better job summarizing the main takeaways of the paper and explaining why the discussed phenomena are particularly important in the context of foundation models (i.e., when learning a transferable model). Some sentences are difficult to contextualize without having read the full paper — for example:  “that time is continuous and this continuity should be maintained, and that regression algorithms should regress to the mean,” or “while a quantization-based embedding introduces a continuous-to-discrete ‘unrounding’ bias, and then relies on training to (imperfectly) recover continuous information in the hidden space.”  Moreover, the example on Chronos vs. Chronos-Bolt performance on chaotic systems is not self-contained and not particularly informative without the cited paper. I would use the introduction to provide more context and clearer motivations for the study. To clarify, I find the motivations sound — I am only referring to the quality of the presentation.
- I found Theorem 1 quite difficult to parse, as the terminology and assumptions are packed into a few lines. I understand its purpose, but I recommend streamlining the presentation and expanding the discussion on assumptions and their implications. While the appendix provides detailed explanations (which I appreciate), the main text does not adequately contextualize the theorem or its significance. For instance, the assumptions on the weight and bias matrices seem unreasonable without further discussion.
- The discussion on periodicity bias — specifically the impact of architecture (encoder vs. decoder vs. encoder-decoder), REG token, and patch size — is rather limited in the main body. I understand the space constraints, but as it stands, some claims appear less convincing without looking at the appendices.
- In Section 3, please clarify the notation. If I understand correctly, *x* and *y* refer to arbitrary patches, and since *k = 1*, these are scalar. Please make this explicit in the text.
- “When a period is far from zero, its embedded vectors occupy a larger portion of the embedded context, which makes it easier for the model to learn. [...]”  This is unclear and should be elaborated on further. Please clarify and/or provide an example.
- Please include the related work section in the main paper (you can make it more compact). Since much of the analysis focuses on comparing Chronos and Chronos-Bolt, a more in-depth comparison of the two architectures should be provided — particularly regarding the tokenization mechanism used in Chronos. I understand the space limitations, but I would recommend moving some empirical results to the appendix (or making the introduction more concise) to make room for a clearer context.
- While some parts of the paper would benefit from more discussion and detail (preliminaries, related work, etc.), other parts could be streamlined (e.g., Sections 3 and 4).

### Additional comments

- The paper does not mention plans to release the code, but doing so would be very helpful. I also encourage the authors to include a reproducibility statement, as suggested by the ICLR guidelines.
- Most empirical results are based on synthetic datasets with no noise (systematic noise, not outliers). How do the authors expect the presence of noise to affect their observations?

Overall, I liked the paper, and if the authors address the issues above and improve the presentation, I'd be happy to increase my score.

### Minor comments

* What do you mean by “first-order structure” (e.g., line 221)?
* Rather than “periodicity,” I would suggest using “seasonality,” which is the more common term in the time series literature.
* In Figure 4a, please clarify exactly what *x* and *y* represent.
* All models used in the study are pretrained, correct?
* It would be interesting to include experiments on more varied datasets, both synthetic and real.

**Questions:**

Please comment on the above weaknesses.

---

> ### Author Response · Authors · 2025-11-18
> **Official Comment by Authors (1/2)**
>
> We thank the reviewer for the careful review and insightful comments.
>
> * **Presentation of the Introduction**
>
>   We appreciate your comments on the presentation of the introduction. We have revised our introduction to incorporate your comments and improve the delivery of our main messages. To summarize:
>
>   * We have added more context and motivation in the first three paragraphs. We also echoed these important messages that the design choices of TSFMs are tightly coupled with the nature of tasks/datasets in the last paragraph.
>
>   * We have clarified the two sentences (i.e., "that time is continuous and this continuity should be maintained, and that regression algorithms should regress to the mean," and "while a quantization-based embedding introduces a continuous-to-discrete 'unrounding' bias, and then relies on training to (imperfectly) recover continuous information in the hidden space.") that were initially difficult to contextualize without a full read of the paper.
>
>   * We removed the paragraph on comparing Chronos vs Chronos-Bolt, and replaced it with a broader discussion of why understanding the design choices of TSFMs is beneficial to the understanding of TSFMs applying to different benchmarks.
>
> * **Interpretation of Theorem 1**
>
>    We have added some explanations of the roles of $\mathbf{V}$ and $\mathbf{U}$ after Theorem 1 statement to help its interpretation. While we agree that the random Gaussian assumption does not apply to pretrained TSFMs, Theorem 1 shows that there are many neural networks that embed low- and high-frequency patches into orthogonal spaces. To empirically validate this orthogonality for pretrained models, we saved several checkpoints of Chronos-Bolt during training. Our new results in Appendix C.1.4 show that the conclusion of Theorem 1 indeed holds throughout pretraining.
>
> * **Discussion of the Periodicity Bias**
>
>   Yes, we agree that since the periodicity bias depends on various knobs, e.g., patch size, architecture choice, and [REG] token, it should have an expanded discussion. We have expanded this in the main text for ease of understanding without the need to look into the appendix for the details.
>
> * **Clarification of the Norm Bias Discussion**
>
>   We have made the sentence that used to start with "when a period is far from zero" clear in its meaning in the last paragraph of Section 3. We also linked it to Figure 29 in Appendix D.3, which provides a nice illustration of this point on a multi-offset context example.
>
> * **More on Preliminaries and Related Work**
>
>   We have added a related work section in the introduction. We have also moved Figure 2 from Appendix A (old Figure 9) to the main body. Figure 2 illustrates the various TSFM design choices. It also shows how these TSFMs rely on different "knobs," and how each bias that we discuss in the paper relates to these knobs.
>
> * **Code Release**
>
>   We have attached the scripts that we used to evaluate the TSFMs as a zip file associated with this submission, and we will release the code for reproducibility of our work upon publication. We have added the reproducibility statement to our paper.

---

> ### Author Response · Authors · 2025-11-18
> **Official Comment by Authors (2/2)**
>
> * **Noisy Datasets**
>
>   We did provide experiments on real-world data in the bottom-right tables of Figures 6 - 7 of the revised paper (Figures 5 - 6 of submitted version). In these experiments, we augment the 27 real-world datasets used in Chronos' zero-shot evaluation, that contain more noise and missing values, with a multi-scale or a multi-offset structure. (See Appendix D.2.1 and D.3.1 for more information.) In particular, we showed that the same findings we identified on our synthetic data also hold on the real-world data, i.e., as the scale ratio and offset gap are increased, the performance of the continuous embedding-based models, e.g., Chronos-Bolt, TimesFM and Moirai worsens, respectively. The tables mirror the corresponding synthetic data plot above them.
>
>   To show more evidence on noisy datasets, we also included additional experiments that inject noises into the regress-to-the-mean test dataset. (See Appendix E.4 for more details.) Our results show that the regression-to-the-mean bias holds seamlessly for noise-free and noisy data.
>
> * **Minor Comments**
>
>   * By "first-order structure," we refer to the dominant, large-scale patterns in a time series, i.e., those that explain most of the variation when the signal is decomposed into frequency components. To avoid ambiguity, we have revised the manuscript to use clearer terminology of "dominant large-scale structure."
>
>   * We appreciate the suggestion of using the word "seasonality" instead of "periodicity." We choose the word "periodicity" to refer to loosely recurring patterns rather than cycles that repeat at a fixed frequency, in which case the word "seasonality" would be more appropriate. (See Footnote 1). We are happy to adjust the wording if that is clearer.
>
>   * We have clarified the meanings of the real numbers $x$ and $y$ in the previous Figure 4 (Figure 5 in the revised version).
>
>   * Most models we used in our experiments are pretrained checkpoints that are publicly available. One exception is that for ablation purposes, we also pretrained a Chronos-Bolt checkpoint with a patch size of $k = 1$. This eliminates the temporal biases induced by patching and makes our comparison between Chronos and Chronos-Bolt fairer when we discuss the other geometric and regress-to-the-mean biases.
>
>   * For experiments on real-world and noisy datasets, see our above comment.
>
> We are happy that you liked the paper, and hope that with these clarifications and improvements to the presentation, you would consider raising your score, as mentioned. Please let us know if you have any other questions.

---

> > ### Comment · Reviewer_cdYf · 2025-11-21
> >
> > Dear authors,
> >
> > Thank you for your rebuttal. Most of my concerns have been addressed, although some remain. These are mainly related to (1) the clarity of Theorem 1 and its role in the paper, and (2) limitations in the discussion of background and related work at the beginning of the paper.
> >
> > I still find the theorem quite difficult to parse. You might consider including only a proposition that provides a high-level summary of the theorem, and referring readers to the appendix for the full details. Conversely, an expanded related work section and the inclusion of a Preliminaries section, where both the Chronos and Chronos-Bolt architectures are described in more detail, would (in my opinion) substantially improve the presentation.
> >
> > Nevertheless, setting these (possibly subjective) points aside, I believe the paper is a good contribution, and the revision has significantly improved the presentation (which I am confident will be improved further in a camera-ready version). I will increase my score.
> >
> > -------
> > Minor: Spotted a typo in Figure 17: "Magtinude of Attention Scores"

---

> > > ### Author Response · Authors · 2025-11-21
> > >
> > > Thank you for carefully reviewing our rebuttal. We appreciate that you found it a significant improvement and have increased your rating. We agree with your further feedback, and have made the following changes:
> > >
> > > * On lines 143-150, we added more contexts on the comparison of Chronos versus Chronos-Bolt to enhance the readability of our paper to a general audience.
> > >
> > > * We put a more concise (and interpretable) version of Theorem 1 in the main manuscript, leaving a full statement to Theorem 1' in Appendix C.1.1.
> > >
> > > We have also corrected the typo in Figure 17; thank you for your close attention to the details.

---

### Official Review · Reviewer_SCK8 · 2025-10-31

**Soundness:** 3
**Presentation:** 3
**Contribution:** 2
**Rating:** 6
**Confidence:** 3

**Summary:**

This paper presents a systematic investigation into the implicit biases of modern Time Series Foundation Models (TSFMs). Rather than proposing a new model, the authors aim to understand how common design choices ("knobs") influence model behavior. They identify three primary classes of biases: 1) Temporal Bias, induced by patch size, which affects how models handle different frequencies and periodicities; 2) Geometric Bias, arising from the choice between continuous and discrete (quantization-based) embeddings, which impacts how the model perceives locality, scale, and offsets; and 3) Regression-to-the-Mean Bias, driven by the training loss function (e.g., L1/L2 vs. Cross-Entropy), which determines a model's behavior under uncertainty. The study uses a mix of theory and controlled empirical evaluations, primarily comparing Chronos and Chronos-Bolt, to demonstrate how these biases manifest and interact, ultimately shaping a model's suitability for different types of time series data (e.g., standard benchmarks vs. chaotic systems).

**Strengths:**

The paper addresses a critical need in the TSFM literature. By shifting the focus from "what is SOTA" to "why models behave the way they do," it provides lasting insights that will remain relevant even as new models emerge. This is the kind of scientific inquiry that fosters deeper understanding.

The combination of theory, controlled synthetic experiments, and analysis on real data is a major strength. The use of Chronos vs. Chronos-Bolt as a primary case study is an elegant experimental design choice that isolates the effects of specific design decisions.

The paper is exceptionally clear. The three-part framework of Temporal, Geometric, and Regression-to-the-mean biases is intuitive, well-supported, and provides a powerful lens through which to view TSFM design. The authors do an excellent job of explaining not just that a bias exists, but how the specific design choice leads to it.

The paper provides practical takeaways. For example, it explains why a model like Chronos (with quantization and cross-entropy loss) might be better for chaotic systems (due to less regression-to-the-mean and a different geometric bias), while a model like Chronos-Bolt might be more robust for noisy, trend-based series. This helps bridge the gap between theory and practice.

**Weaknesses:**

While the focus on Chronos/Chronos-Bolt is a strength for control, it is also a potential weakness for the generality of the conclusions. The paper does include other models like TimesFM and Moirai in some experiments (which is great!), but the core narrative and many of the detailed analyses are tightly coupled to the Chronos family. It would strengthen the paper to either include more direct evidence from a wider variety of architectures or to more explicitly frame the conclusions in terms of the design choices themselves (e.g., "models using continuous embeddings and L1 loss...") rather than implying they hold for all TSFMs.

The case study on outlier handling in Section 5 is a good first step toward understanding how biases interact. However, this aspect feels somewhat underexplored compared to the detailed analysis of each individual bias. The real world is messy, and models are always operating under a combination of these effects. A slightly deeper discussion or another case study (e.g., handling non-stationarity) could have made the "mixture of biases" a more central contribution.

The inclusion of Theorem 1 is a strength. However, the theorem relies on standard but strong assumptions (e.g., random Gaussian weights) that are not met in a fully trained network. A brief discussion on the expected qualitative persistence of these results in trained models would be helpful to bridge the gap for practitioners. For instance, do we expect the orthogonality of embeddings for different frequencies to be as clean, or just a general tendency?

**Questions:**

N/A

---

> ### Author Response · Authors · 2025-11-18
>
> We thank the reviewer for the careful review and insightful comments.
>
> * **Experiments with Models Beyond Chronos/Chronos-Bolt**
>
>   We chose to mainly compare Chronos and Chronos-Bolt for experimental control with their similar backbone, size, and pretraining (which makes the comparisons fairer), and importantly because they more broadly represent the different design choices in TSFMs, e.g,:
>
>     - Quantization embedding and cross-entropy loss function in Chronos.
>     - Continuous embedding and regression loss function in Chronos-Bolt, which is also used in other TSFMs, e.g., Moirai, TimesFM, and TimeMoE (See Figure 2 for summary, which we moved from Appendix A to the main body).
>
>   As the reviewer noted, where necessary, we did include comparisons to other TSFMS, e.g.,
>
>     - TimesFM (decoder-only) and Moirai (encoder-only) in Figure 4 to test the effect of the architecture design choice on the periodicity bias.
>     - TimesFM and Moirai behave poorly and similarly to Chronos-Bolt on multiscale (Figure 6) and multi-offset (Figure 7) time series.
>     - Lastly, to differentiate between $L^1$ regression in Chronos-Bolt and $L^2$ regression, we also compare to TimesFM in Figure 8 and show that Chronos is the only TSFM that does not regress to median/mean.
>
>   To further show that our analysis is general to more TSFMs, we have added additional comparisons to other TSFMs and have updated the conclusion. To summarize:
>
>   * We added an additional comparison to TimesFM and Moirai in three existing experiments. Please see our general response for more details.
>
>   * We have also modified the summary boxes and the conclusion to focus more on the design choices that induce the biases, as the reviewer suggested.
>
> * **More In-Depth Discussion of the Outlier Bias**
>
>   Given the additional page in the rebuttal revision, we have expanded our discussion of the outlier handling case study with more details of how different biases are mixed together. (See the updated discussion in Section 5). We have also added a diagnostic figure in Appendix F to highlight how the outliers affect both the embedding and encoding of TSFMs based on different design choices. Related to the point that "the real world is messy," we have extended some of our synthetic experiments to the noisy case to show that the same conclusion remains true. See our general response and Appendix E.4.
>
> * **Practical Implication of Theorem 1**
>
>   Yes, the weight matrices in a pretrained TSFM are not random Gaussian. We use Theorem 1 to mainly convey the message that *many* ReLU-activated NN embeddings send low- and high-frequency patches into nearly orthogonal spaces; therefore, they are easy to encounter during training. We have added a discussion after Theorem 1 to acknowledge that the random Gaussian assumption does not apply to trained weight matrices. Moreover, we added an experiment in Appendix C.1.4 to watch this orthogonality during training, and we observed that the low- and high-frequency patches are indeed embedded into two orthogonal subspaces at any stage of pretraining.
>
> We are happy to answer any follow-up question(s) that the reviewer may have.

---

> > ### Author Response · Authors · 2025-11-27
> >
> > Dear Reviewer SCK8,
> >
> > We sincerely appreciate your time and thoughtful evaluation of our work. Your feedback was very helpful and directly contributed to improving the manuscript, as we outlined in the rebuttal. We hope our clarifications and additional experiments have addressed your concerns. Please feel free to let us know if any questions remain. Thank you again for your time and support.
> >
> > Best regards,
> > Authors

---

### Author Response · Authors · 2025-11-18
**General Response (2/2)**

## **Real-World and Noisy Dataset**

  Experiments in our paper use both synthetic and real-world datasets. Synthetic settings allow us to isolate and precisely probe subtle differences between architectural design choices. The patterns we purposefully construct (e.g., multi-scale structure, multi-offset trends, and high-frequency components) reflect common, important, and hard-to-capture phenomena in practical time-series applications. In addition, our conclusions are not based solely on synthetic data. The results in Figures 6 and 7 use a large corpus of real-world datasets, and demonstrate that the geometric bias we identify does manifest in practice. Section 5 provides an explicit and systematic study of noise robustness, which further validates our findings.

  To make our analysis more comprehensive, we also augment our experiment in Section 4 with noise. See Appendix E.4 for the new results. Our results show that the regression-to-the-mean bias holds seamlessly for noise-free and noisy data.

## **Vocabulary Size and the Geometric Bias**

  Reviewers also mentioned the vocabulary size (when a quantization embedding is in use) as a potential factor that impacts the geometric bias. We believe this is a very interesting point and have done some additional experiments to investigate the role of the vocabulary size. (See the results in Appendix D.4.) We observe that an increasing vocabulary size makes Chronos attend more to the local content. In addition, an increasing vocabulary size equips Chronos with a stronger distance bias that makes it capable of handling the fine-scale signals better, while using a small vocabulary size nonetheless makes Chronos better at capturing fine-scale signals than Chronos-Bolt, which aligns with our discussion of the distance bias.

## **Outlier Handling Discussion**

  We appreciate the reviewers' interest in our outlier bias case study, which is an application of how these various biases can interact with each other. We agree that this is very interesting and with an additional page for the rebuttal revision, we have expanded our discussion of the outlier handling case study in Section 5 to highlight how each bias comes into play. We also added a Figure in Appendix F to reveal how TSFMs with different design choices handle these outliers.

---

### Author Response · Authors · 2025-11-18
**General Response (1/2)**

We appreciate the reviewers' valuable feedback to help improve the clarity and generality of our manuscript. We are glad that the reviewers found our work an important contribution to the field to help better understand the subtle design choices and inductive biases in TSFMs. To summarize:

* **Reviewer SCK8** highlights that "the paper addresses a critical need in the TSFM literature. [...] This is the kind of scientific inquiry that fosters deeper understanding." In addition, they state that "the authors do an excellent job of explaining not just that a bias exists, but how the specific design choice leads to it" and that our work "helps bridge the gap between theory and practice."

* **Reviewer cdYf** appreciates "the focus on clarifying the impact of design choices that are often overlooked" and "believes the paper is a solid contribution to the conference."

* **Reviewer 8DhK** highlights that the "topic and analysis are novel" and that "the paper is extremely thorough." In addition, they state that they expect our work to "highly significant for time-series modelling relying on deep learning in general and not only for research in time-series foundation models."

* **Reviewer UNdi** acknowledges that our paper "shifts the focus from performance chasing to understanding inductive biases, highly relevant for Time Series Foundation Models (TSFMs)'s robustness and transferability." They also state that our paper "did a deep analysis on each implicit bias, validating findings across several leading Time Series Foundation Models (TSFMs). It also shows practical insights on each bias's investigation."

We address the minor, reviewer-specific comments in our individual reviewer responses and the common response below. We have also updated the manuscript with the changes highlighted in red in the rebuttal version.


## **Experiments with TSFMs Beyond Chronos**

  To show the generality of our discussion of inductive biases beyond Chronos and Chronos-Bolt, we have added several additional experiments with more TSFMs in addition to the ones we already have in the paper:

  * For the geometric distance and norm biases, we have included Moirai, in addition to the existing Chronos, Chronos-Bolt, and TimesFM, in our empirical evaluations. (See the updated Figures 5-6). The results confirm that a continuous embedding preserves geometry more than a quantization embedding does as we demonstrated with Chronos-Bolt, which we used to represent continuous-embedding-based TSFMs.

  * For the regression-to-the-mean bias, we also added Moirai in our experiments in Appendix E.4, in addition to TimesFM, which we had already included for $L^2$ regression loss. The results again show that a regression-based TSFM, represented by Chronos-Bolt, regresses to the mean forecast while a classification-based TSFM, represented by Chronos, does not.

## **Practicality of Theorem 1**

  Theorem 1 concerns the setting of random Gaussian weight matrices. While the weight matrices in a pretrained model are generally not random Gaussian, our theorem shows that neural networks that embed low- and high-frequency patches into orthogonal spaces are very dense in the space of parameters. To empirically validate this orthogonality for pretrained models, we saved several checkpoints of Chronos-Bolt during training. Our new results in Appendix C.1.4 show that the conclusion of Theorem 1 holds throughout pretraining.

---

### Meta-Review · Area_Chair_6sx7 · 2025-12-25

**Summary:**

All reviewers viewed the work as an important contribution to understanding the subtle design choices and inductive biases in TSFMs. Also, each of them kept positive comments on this paper.

**Reviewer Concerns:**

Reviewers UNdi and cdYf have responded that their questions have been addressed. Reviewers SCK8 and 8DhK primarily felt that some statements in the paper were vague and that the assumptions of Theorem 1 were too strong; the authors clarified these issues in the rebuttal.

**Reviewer Scores:**

Reviewers SCK8 and 8DhK will at least maintain their current positive ratings.

---

### Decision · Program_Chairs · 2026-01-26

Accept (Poster)